# Characterizing spatiotemporal white matter hyperintensity pathophysiology in vivo to disentangle vascular and neurodegenerative contributions

Olivier Parent [1,2] ✉, Zaki Alasmar[1,2], Sophia Osborne[1], Aurélie Bussy [1,2], Manuela Costantino [1], Jérémie P. Fouquet [1], Daniela Quesada [2], Alexandre Pastor-Bernier[3,4], Alfonso Fajardo-Valdez[3], Alexa Pichet-Binette[5,6], Ann McQuarrie[7], Josefina Maranzano[8,9], Gabriel A. Devenyi [1,10], Christopher J. Steele [11,12], Sylvia Villeneuve [3,10,13], the PREVENT-AD Research Group*, the Alzheimer's Disease Neuroimaging Initiative (ADNI)*, Mahsa Dadar[1,2,10,15] ✉ & M. Mallar Chakravarty[1,2,10,14,15] ✉

White matter hyperintensities (WMHs) are neuroimaging markers widely interpreted as caused by cerebral small vessel disease, yet emerging evidence suggests that a subset may have a neurodegenerative etiology. Current imaging methods have lacked the specificity to disentangle biological processes underlying WMHs in vivo. Here, we used voxel-level normative modeling and seven microstructural MRI markers with complementary biophysical sensitivities to generate single-subject high-resolution WMH pathophysiology maps in a large cohort ($n = 32,526$). We calculated data-driven spatial patterns of similar WMHs, revealing distinct periventricular, posterior, and anterior clusters. We identified a reproducible WMH signature linked to dementia and Alzheimer's disease, characterized by a posterior predominance and a pathophysiological pattern indicative of selective fiber degeneration. Posterior WMHs connected cortical regions vulnerable to tau pathology. Our framework helps parsing vascular and neurodegenerative contributions of WMHs in vivo, which could alter the course of treatment strategies and provide nuanced interpretations of research findings.

White matter hyperintensities (WMHs) are among the most common neuroradiological findings in older adults[1,2]. They are present in most individuals over 65 years old and appear as bright regions in fluid-attenuated inversion recovery (FLAIR) images during magnetic resonance imaging (MRI) exams[3]. The presence of WMHs is widely considered a downstream consequence of cerebral small vessel disease and is typically used to detect this disorder, as neuroimaging methods

able to directly investigate small blood vessels (< 500 μm in diameter) are lacking[4]. The extent of visible WMHs is used by neurologists and radiologists to monitor the cerebrovascular disease burden and for diagnoses such as vascular dementia[5], by researchers as a quantifiable marker of cerebrovascular dysfunction[6,7], and in some clinical trials as outcome measures of vascular treatments (e.g., antihypertensives)[8,9]. Importantly, the pathogenesis of WMHs can vary, with a growing body

A full list of affiliations appears at the end of the paper. *A list of authors and their affiliations appear at the end of the paper.
✉e-mail: olivier.parent@mail.mcgill.ca; mahsa.dadar@mcgill.ca; mallar.chakravarty@mcgill.ca

of evidence suggesting that some WMHs may be caused in part by neurodegenerative processes rather than solely vascular disease[10–13]. For example, a post-mortem examination found no evidence of hypoperfusion-related damage within the WMHs of brain specimens with Alzheimer's disease (AD) pathology, and instead observed signs of Wallerian degeneration[11]. This pathological heterogeneity is currently invisible to conventional in vivo imaging and could result in misinterpretation or misdiagnosis of cerebral pathology, hindering the application of precision medicine in diagnosis and treatment, as well as complicating the interpretation of research findings. FLAIR images lack the specificity to parse the biological processes underlying WMHs, as these contain various levels of inflammation[14], edema[15,16], demyelination[11,17], and axonal degeneration[11] that cannot be distinguished with a single contrast. A deep in vivo phenotyping of WMH pathophysiology could help parse neurodegenerative from vascular etiologies.

There is evidence that some of the pathophysiological heterogeneity observed in WMHs may be dependent on their neuroanatomical location. Studies suggest that periventricular WMHs are associated to a larger extent with vascular risk factor burden and contain more edema and less demyelination relative to deep WMHs[18]. Consequently, periventricular and deep WMHs are often assessed separately using heuristic methods with criteria that can vary significantly between studies[19]. Other parcellation schemes include cortical lobes, vascular territories, or white matter tracts[20]. In contrast to these hypothesis-driven parcellations, data-driven methods can uncover more complex patterns of WMH spatial heterogeneity, and while such examinations have been done based on the co-occurrence of WMHs[12,21], these have not taken into account the underlying pathophysiological alterations. Given the sparsity of WMH prevalence at the voxel-level, such an analysis requires a very large dataset. Furthermore, the pathophysiological cascade of WMHs remains speculative, and it is unclear whether pathological events follow the same sequence in vascular and neurodegeneration-related WMHs. The prevailing hypothesis suggests that fluid infiltration from ventricles and blood vessels triggers inflammation, leading to tissue damage such as demyelination, oligodendrocyte loss, and, in more severe cases, axonal degeneration[3]; however, this is one of several pathways through which WMHs can emerge[11]. Generally, WMH pathophysiology is characterized *post-mortem* using histological and immunohistochemical stains sensitive to different aspects of cellular, molecular, and pathological features. Multi-contrast microstructural MRI, while lacking the biological specificity of these techniques, has established sensitivities to various tissue properties[22,23], providing a window into biophysical tissue alterations in the brain and allowing for the non-invasive, large-scale investigation of WMH pathophysiology. This in vivo approach could provide clinicians with supplemental information about WMH pathogenesis that could alter the course of treatment and disease prevention strategies.

Our observational study aimed to parse the pathophysiological, spatial, temporal, and ultimately etiological heterogeneity of WMHs. We generated high-resolution maps of WMH pathophysiology by performing voxel-wise normative modeling of seven MRI markers sensitive to fluid, fiber, myelin, and iron content in a large population cohort (UK Biobank; $n = 32,526$) and transferred these models to an independent cohort of subjects along the AD spectrum. We calculated spatial regions of WMHs that showed similar pathophysiological features with unsupervised clustering, allowing us to then estimate region-specific pathophysiological cascades with machine learning-based disease modeling techniques. With an improved understanding of these spatiotemporal dynamics, we identified reproducible disease-specific signatures of WMH pathophysiology in vascular and neurodegenerative disorders.

## Results

### Image processing and microstructural MRI

We used cross-sectional data from 32,526 UK Biobank (UKB) participants, a population-based prospective study of middle- to late-aged individuals. Exclusion criteria (Methods) included disrupted MRI data used in downstream analyses (missing modality, motion artifacts, or processing failures), missing demographic information, and a diagnosis of multiple sclerosis to not confound white matter lesions from WMHs (step-by-step exclusions detailed in Supplementary Fig. 1; demographic statistics detailed in Supplementary Table 1). While MRI images have already been processed by UKB[24], we updated key processing steps to ensure the highest data quality possible for our specific use case. A customized UKB template was generated[25] (available at https://github.com/CoBrALab/WMH_patho_UKB) and MRI images were aligned to this template with multispectral registration (Methods), providing highly accurate voxel-by-voxel correspondence across subjects, particularly in the periventricular white matter, which is an important area of interest. Second, to ensure accurate delineation of WMHs and normal-appearing white matter (NAWM) while leveraging multi-contrast information, we retrained the BraIn SegmentatiON (BISON) algorithm[26] with a manually defined reference.

To investigate tissue properties, the multi-shell diffusion MRI and two-echo susceptibility-weighted MRI acquisitions were used to derive seven quantitative microstructural markers characterizing important aspects of white matter biophysical composition that are altered in aging, broadly categorized as fluid-sensitive (mean diffusivity [MD] and isotropic volume fraction [ISOVF]), fiber-sensitive (fractional anisotropy [FA], intracellular volume fraction [ICVF], and orientation dispersion [OD]), and myelin- and iron-sensitive (T2* and quantitative susceptibility mapping [QSM]). However, their biological underpinnings are complex and potentially disease-dependent (summarized in Table 1). For example, diffusion tensor imaging (DTI) indices such as FA and MD are particularly difficult to interpret since they assume a main axis of diffusion within voxels, while most voxels instead contain multiple fiber populations with different orientations[27]. Different multi-compartment modeling schemes of the diffusion signal have been developed to handle the crossing-fiber problem and increase the interpretability of microstructural metrics, one of which is the Neurite Orientation Dispersion and Density Imaging (NODDI) model used here[28]. While no metric perfectly captures individual biological processes (e.g., inflammation, demyelination, edema), interpreting these markers together can help generate hypotheses about the potential biological substrates of the observed effects[23].

### Estimates of WMH pathophysiology

Pathophysiology was operationalized as the deviation in microstructural values in WMHs relative to the expected microstructure in normal-appearing white matter (NAWM). In other words, we aimed to estimate pathophysiological processes such as water accumulation/edema as opposed to microstructural properties like water density. High-resolution normative modeling of white matter microstructure was performed for the seven markers, only including individual-level voxels that were labeled as NAWM (Methods; Fig. 1A)[29]. We note that NAWM may still be pathological to a certain extent, as has been previously demonstrated[16]. However, pathological NAWM remains ill-defined, and there is currently no established framework to identify it. We thus focus on contrasting WMH from non-WMH tissue, considering that some tissue alterations are part of the NAWM variations. From these normative models, we generated age- and sex-specific maps of expected normal microstructure, which are then used to standardize WMH microstructure using a z-score procedure (example in Fig. 1B). Voxels labeled as NAWM were not included in any other analyses. The resulting seven high-resolution pathophysiology maps represent, in essence, an individual-level signature of the biophysical processes at play within WMHs. These maps have many desirable properties: (1)

                                                                                      **2**

**Table 1 | Literature review of biological sensitivities of MRI microstructural markers**

| Modality | Marker | Physical interpretation | Biological sensitivities |
|---|---|---|---|
| Diffusion tensor imaging (DTI): Eigendecomposition of the diffusion signal resulting in a tensor described by three orthogonal axes of diffusion | Mean diffusivity (MD) | Average of diffusion along the three orthogonal diffusion axes of the tensor | Extracellular free water (+)[42], inflammation (+)[43,44] |
| | Fractional anisotropy (FA) | Amplitude of diffusion in the principal direction relative to the other two axes | Intra-axonal water (+)[42], fiber organization (+)[42] |
| Neurite orientation dispersion and density imaging (NODDI): Modeling of the diffusion signal in three different tissue compartments | Isotropic volume fraction (ISOVF) | Fraction of the signal that diffuses completely isotropically | Extracellular free water (+)[28,45] |
| | Intracellular volume fraction (ICVF) | Fraction of the anisotropic signal with negligible perpendicular diffusion (modeled as a collection of sticks) | Intra-axonal water (+)[28,45] |
| | Orientation dispersion (OD) | Distribution of diffusion directions (0 when fibers are completely aligned, 1 when there is no preferred diffusion direction) | Fiber organization (-)[28,45,95] |
| Susceptibility-weighted imaging (SWI) | T2* | Exponential decay of the transverse relaxation impacted by susceptibility variations and spin-spin interactions | Myelin (-)[47,48], iron (-)[47,48], extracellular free water (+)[96] |
| | Quantitative susceptibility mapping (QSM) | Sum of paramagnetic and diamagnetic susceptibilities | Myelin (-)[47], iron (+)[47] |

The main biological sensitivities listed here are a general guide only and should not be interpreted as the only sources of signal. In other words, the metrics are sensitive but not specific to the biological sources of signal listed here. The directionality of the associations is indicated.

they isolate the effect of the pathology within WMHs on the microstructure (i.e., the WMH pathophysiology), (2) since all markers are on the same scale, they do not require further normalization, and (3) they remove the spatial microstructural contrast driven by normal anatomical variations, which is a necessary property for our spatial clustering analyses (see below). In other words, regional variations in normal white matter microstructure are removed. As an example, the single-subject raw OD map in WMHs has a high level of spatial variation, but these are not present in the z-scored map (Fig. 1B, first row compared to last row), highlighting that spatial variations were due to the anatomy and not caused by differences in WMH-related alterations.

**Spatial regions of pathophysiologically similar WMHs**
From this rich dataset, we first aimed to characterize group-level spatial patterns of WMH pathophysiology in the general population. Mean WMH pathophysiology maps were calculated by averaging all subject-level maps (excluding all voxels labeled as NAWM), resulting in one between-subject map per marker. We excluded voxels with low WMH or NAWM prevalence, which have higher noise levels (Fig. 2A; Methods; Supplementary Figs. 2 and 3), and these maps were used as inputs to a spectral clustering algorithm[30]. This data-driven approach identified spatial regions that are most similar between themselves solely in terms of WMH pathophysiology, independent of voxel location. Yet, all clusters were mostly spatially contiguous, suggesting that WMHs appearing in specific areas share pathophysiological signatures. We opted for a solution of three clusters (Fig. 2B) in the interest of parsimony since the four-cluster solution (Supplementary Fig. 4) was less spatially contiguous with similar pathophysiological differentiation across clusters. The first cluster surrounded the ventricular wall and showed low abnormality on all metrics (median z-scores: MD = 0.49, ISOVF = − 0.18, FA = −0.61, ICVF = − 1.26, OD = − 0.3, T2* = 0.63, QSM = 0.06). These voxels may be contaminated by partial volume effects (i.e., ventricle signal) and include small caps WMHs that are likely benign[31]. The second cluster was located in periventricular-posterior regions and showed abnormality related to fluid accumulation, fiber alterations, and myelin and iron loss (see Table 1 for more complex biological interpretations; median z-scores: MD = 4.01, ISOVF = 1.07, FA = −1.56, ICVF = − 3.22, OD = − 0.3, T2* = 1.47, QSM = 0.23). The third cluster was located in deep-anterior regions and showed a similar pathophysiological pattern to the posterior cluster but with higher absolute abnormality on most metrics (median z-scores: MD = 5.89, ISOVF = 1.47, FA = −1.99, ICVF = − 4.37, OD = −0.34,

T2* = 1.6, QSM = 0.29). We also observed that the clustering solution explained most of the variance in the average pathophysiology spatial maps of MD, ICVF, and ISOVF ($R^2$ of 0.81, 0.78, and 0.63, respectively; Supplementary Fig. 5), a moderate amount for FA and T2* ($R^2$ of 0.39 and 0.29, respectively), and almost none for QSM and OD ($R^2$ of 0.04 and 0.00, respectively). The posterior and anterior clusters mostly differed in the severity of pathophysiological markers, but still revealed distinct spatial signatures. Further characterization of specific pathophysiological alterations is needed to assess potential etiological signatures.

To quantify and contextualize the localization of each spatial cluster, we compared the overlap of our custom parcellation with other heuristic parcellations used in the field: periventricular/deep, lobar regions, cerebral artery territories as estimated from a probabilistic atlas[32], and fiber type (Fig. 2C). We observed that the first cluster was more periventricular (Dice similarity coefficient [DSC] = 0.37), located in temporal white matter (DSC = 0.31) and in commissural fibers (DSC = 0.21), and supplied by the posterior cerebral artery (DSC = 0.27). This limited overlap between cluster 1 and the periventricular a priori region is due to the much broader coverage of the a priori region (<8 mm from the ventricle, as is typically done)[19], as compared to the data-driven cluster. The second cluster was more periventricular (DSC = 0.54), located in parietal white matter (DSC = 0.37) and commissural fibers (DSC = 0.45), and supplied by the middle cerebral artery (DSC = 0.37), although it also showed overlap with the posterior and anterior cerebral artery territories (DSC of 0.30 and 0.26, respectively). The third cluster was mostly located in deep and frontal white matter (DSC of 0.76 and 0.66, respectively) as well as association and projection fibers (DSC of 0.56 and 0.41, respectively), and largely supplied by the middle cerebral artery (DSC = 0.68). The lack of consistent agreement with often-used WMH parcellations suggests an improved pathophysiological specificity in these data-driven clusters that is not captured with the more heuristic approaches. For the rest of the manuscript, these spatial clusters will be referred to as periventricular (cluster 1), posterior (cluster 2), and anterior (cluster 3).

To investigate if these patterns simply reflected severity stages (e.g., spreading of WMHs from periventricular to posterior to anterior regions), we separated our sample into three groups based on total WMH volume and ran the clustering analysis within each group (Supplementary Fig. 6A). Remarkably, the spatial overlaps between the subgroup clusters and the original clustering solution in the complete

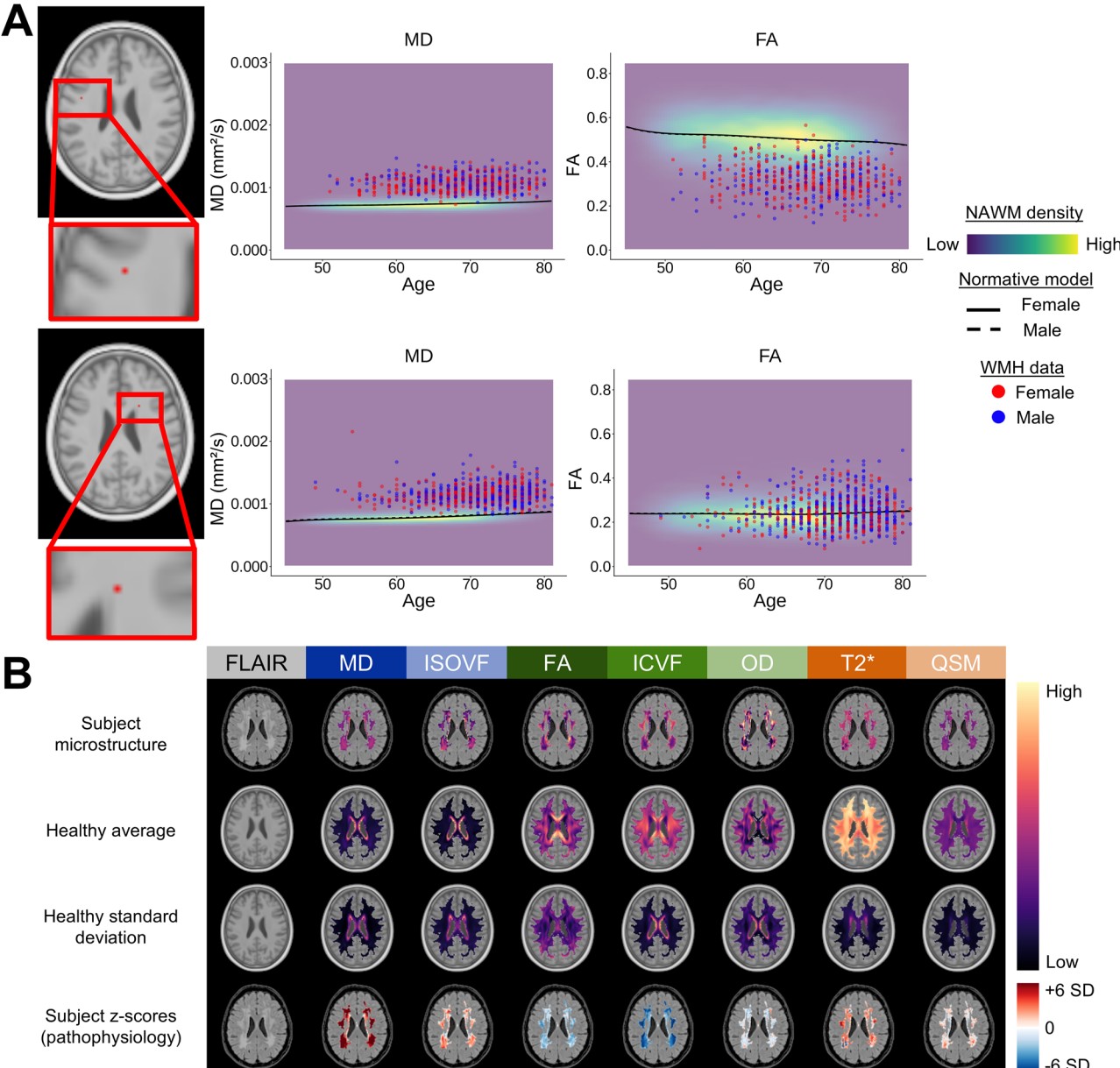

**Fig. 1 | High-resolution estimates of WMH pathophysiology. A** Normative modeling of NAWM microstructure was performed at each white matter voxel for the seven microstructural markers. The process is exemplified at two voxel locations for the MD and FA markers. The density of NAWM microstructural values, fitted normative models, and individual WMH microstructural data are shown. **B** The procedure to derive WMH pathophysiological estimates is shown. First row: raw WMH microstructural maps of one subject in UKB space. Second and third row: average and standard deviation maps of the expected NAWM microstructure, matching the age and sex of the subject. Fourth row: abnormality z-scored maps

(i.e., WMH pathophysiological estimates). Each microstructural marker is color-coded by its primary biological sensitivity: fluid-sensitive in shades of blue, fiber-sensitive in shades of green, and myelin and iron-sensitive in shades of orange. This color coding is kept throughout the manuscript. Source data are provided as a Source Data file. *WMH* white matter hyperintensity; *NAWM* normal-appearing white matter; *FLAIR* fluid-attenuated inversion recovery; *SD* standard deviation; *MD* mean diffusivity; *FA* fractional anisotropy; *ISOVF* isotropic volume fraction; *ICVF* intra-cellular volume fraction; *OD* orientation dispersion; *QSM* quantitative suscept-ibility mapping.

dataset were very high (DSCs: Periventricular > 0.87; Posterior > 0.78; Anterior > 0.88; Supplementary Fig. 6B) and the associated patho-physiological patterns were highly congruent. We also found con-gruent spatial and pathophysiological patterns between males and females (DSCs: Periventricular > 0.95; Posterior > 0.91; Anterior > 0.96; Supplementary Fig. 7). This demonstrated that the observed spatial variations in WMH pathophysiology are robust to the severity of WMH burden and are consistent across sexes.

Finally, voxels that were previously excluded from the clustering were assigned to the closest spatial cluster using a search area strategy (Methods; Supplementary Fig. 8A). This yielded a parcellation that was

comparable but more spatially contiguous relative to a clustering solution where voxels were not excluded based on WMH prevalence (Supplementary Fig. 8B). This parcellation enabled a pathophysiology-informed dimensionality reduction of the high-resolution WMH maps, as we extracted subject-wise region-of-interest (ROI) measures by sampling the median pathophysiological estimates within WMHs in each spatial cluster (Methods). This reduced the complex voxel information into a set of 24 values (8 measures * 3 regions) for each subject. All further analyses are made at the ROI level and not at the voxel level. To simplify the interpretation, we inverted the values of markers that showed negative effects in WMHs (FA, ICVF, and OD) so

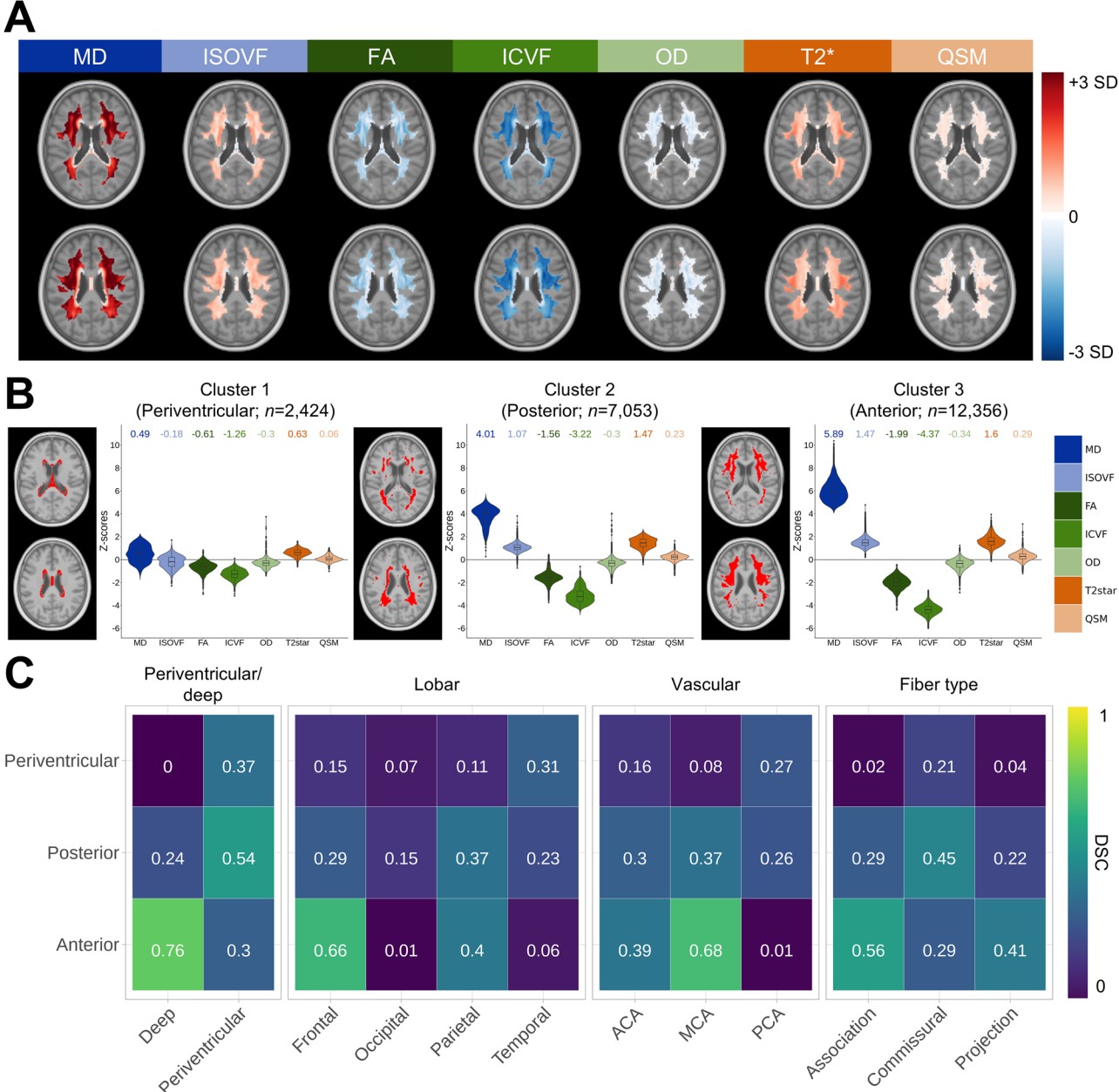

**Fig. 2 | Spatial regions of pathophysiologically similar WMHs. A** Maps of between-subject averaged WMH pathophysiological estimates for each marker. Voxels with a low prevalence of WMH labels (< 30) or NAWM labels (< 5000) were excluded. These maps were used as inputs to a spectral clustering algorithm. **B** Spatial clusters are shown along with the number of voxels within each cluster. Left: voxels included in the cluster are indicated in red. Right: Pathophysiological distributions of the voxels included in the cluster. Medians, first and third quartiles are indicated with box plots. Median values of those distributions are shown at the top. **C** Dice similarity coefficients (DSCs) between our derived spatial clusters (y-axis) and WMH regions from other parcellations (x-axis). Only clustered voxels are included in the calculation. Source data are provided as a Source Data file. *ACA* Anterior cerebral artery; *MCA* middle cerebral artery; *PCA* posterior cerebral artery.

that an increase in z-scores is consistently interpreted as worsening WMH pathophysiology for all markers.

**Defining temporal pathophysiological cascades**

Next, we investigated the temporal heterogeneity of WMH pathophysiology by modeling pathophysiological cascades in our three derived spatial regions using the Subtype and Stage Inference (SuStaIn) technique[33]. SuStaIn is an event-based unsupervised machine learning technique that infers temporal progression patterns from cross-sectional data and is further able to find progression subtypes, essentially sorting abnormality events in the order in which they tend to appear across different clusters of subjects[33]. Pathophysiological

events were determined empirically on a per-marker basis (at least 1% of participants needed to reach the threshold; Methods; Supplementary Fig. 9).

Interestingly, we did not find evidence for sub-trajectories of WMH pathophysiological progression in any of the regions, as is evidenced by nearly identical biomarker progression curves even when investigating cross-validated three-trajectory solutions (Supplementary Fig. 10). We interpret this finding as evidence that most of the pathophysiological heterogeneity is spatial and not temporal, and was adequately accounted for in our spatial clustering step. We thus describe here the pathophysiological cascades shared across all subjects within each WMH region (Fig. 3A). In periventricular WMHs, we

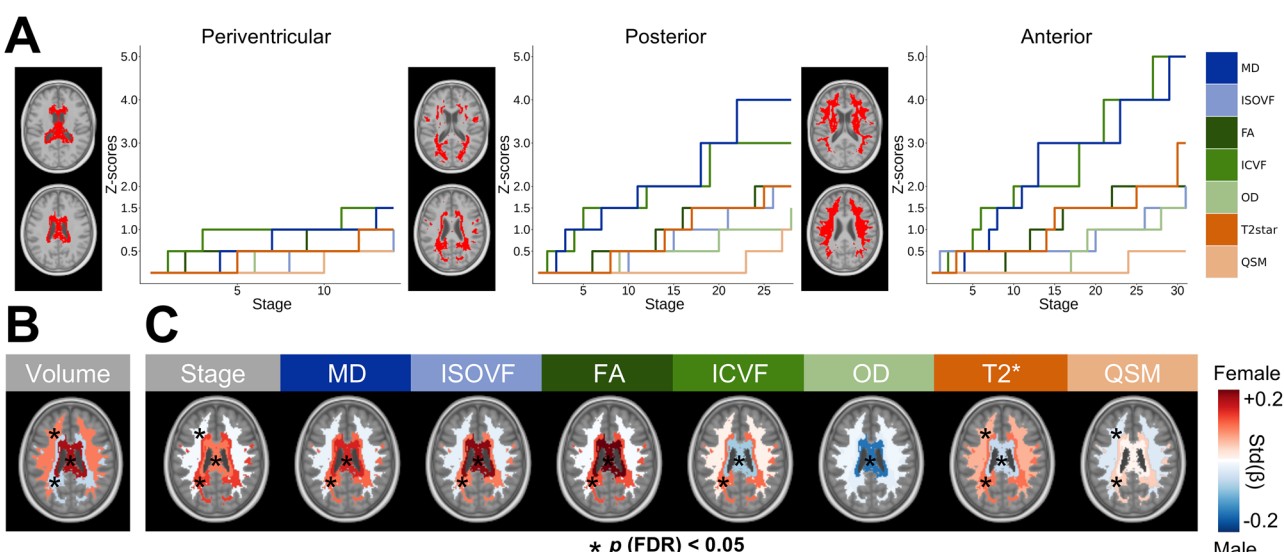

**Fig. 3 | Pathophysiological cascades and sex differences. A** In each WMH spatial region, we modeled the temporal pathophysiological cascade using SuStaIn. Shown are the winner-take-all trajectories, with the x-axis representing a data-driven temporal axis of pathophysiological events (stages) and the y-axis representing the abnormality z-score thresholds. **B** Sex differences in WMH volume. **C** Sex differences in WMH pathophysiology, controlling for the corresponding regional WMH volume. Differences are expressed as standardized betas. Red colors indicate a higher effect in females, and blue colors indicate a higher effect in males. Significant effects after performing false discovery rate (FDR) correction ($p < 0.05$) are shown with asterisks. Source data are provided as a Source Data file. $Std(\beta)$ standardized beta.

observed low abnormality in most markers, with the maximum z-score thresholds being set to 1.5 SD (for MD and ICVF) according to our heuristic. ICVF was the earliest marker to be altered. Posterior WMHs displayed early alterations of the ICVF and MD markers, followed by FA, T2*, and ISOVF in the intermediate stages, and OD and QSM in the later stages. Anterior WMHs showed a similar progression pattern, but with most markers reaching higher abnormality. Positional variance diagrams showed very low stage uncertainty, demonstrating the stability of these temporal trajectories (Supplementary Fig. 11). Using WMH volume as a simpler empirical definition of temporal disease severity, we recapitulate results from our data-driven findings of disease progression (Supplementary Fig. 12A). SuStaIn stages were correlated with regional WMH volumes (Spearman $\rho$: Periventricular = 0.41, Posterior = 0.66, Anterior = 0.85; all $p < 0.001$; Supplementary Fig. 12B) but still showed substantial variation not explained by WMH volumes. The subject-level stage from SuStaIn provided a summary measure of the temporal progression of WMH pathophysiology and is included in further analyses.

## Sex differences in WMH pathophysiology

Previous studies have reported higher WMH burdens in females that manifest after midlife, which is hypothesized to be due to the menopausal transition and the consequent reduction in the neuroprotective effects of estrogen[34,35]. We investigated whether a specific signature of WMH pathophysiology exists in females, providing standardized betas and false discovery rate (FDR)-corrected $p$-values[36]. Interestingly, we found higher WMH volumes in the periventricular ($\beta = 0.17$, $p$ (FDR) < 0.001) and anterior ($\beta = 0.09$, $p$ (FDR) < 0.001) regions in females, but higher WMH volumes in posterior ($\beta = -0.05$, $p$ (FDR) < 0.001) regions in males (Fig. 3B). Investigating differences in WMH pathophysiology when males and females have equivalent WMH volumes (Fig. 3C), we observed a clear pattern of strong effects in females restricted to periventricular and posterior WMHs for most pathophysiological markers. This was not the case for the OD marker, which showed a significantly higher effect in males in the periventricular region but no significant effects in other regions. Taken together, our results show nuanced sex-specific effects in WMHs. There are clear spatial differences, with females having more WMHs but similar pathophysiological

effects in anterior regions, while males show higher WMH volumes but lower pathophysiological effects in posterior regions.

## WMH pathophysiology in vascular and neurodegenerative disorders

With a deeper understanding of WMH spatiotemporal dynamics, we were now well-positioned to elucidate complex disease-specific phenotypes in vascular and neurodegenerative disorders. We first compared WMHs across different diagnostic categories with a case-control design using UKB lifetime diagnosis data (i.e., prevalent and incident cases; Fig. 4A). Control subjects had no endocrine, circulatory, behavioral, nervous, or metabolic disease-related lifetime diagnoses ($n = 8528$; "Methods"). Case subjects were defined by groupings of ICD-10 codes without comorbidities among the diagnoses analyzed (groups described in Supplementary Fig. 13).

In individuals with ischemic heart disease ($n = 2283$), WMH alterations were small but significant, with stronger effects in anterior regions (average $\beta = 0.17$; $p$ (FDR) < 0.05 for all markers) than posterior regions (average $\beta = 0.12$; $p$ (FDR) < 0.05 for all markers except OD and T2*). The stroke group ($n = 526$) showed moderate-to-large effects, particularly in anterior WMHs (average $\beta = 0.47$; $p$ (FDR) < 0.05 for all markers) relative to posterior WMHs (average $\beta = 0.31$; $p$ (FDR) < 0.05 for all markers). In contrast, the dementia group excluding vascular dementia ($n = 31$) showed moderate effects, particularly in posterior WMHs (average $\beta = 0.24$; $p$ (FDR) < 0.05 for WMH volume, MD, ISOVF, OD) relative to anterior WMHs (average $\beta = 0.09$; $p$ (FDR) < 0.05 for OD). Notably, FA effects were more pronounced in stroke, whereas OD effects were more pronounced in dementia. This suggests a signature of neurodegenerative WMHs indicative of specific disruptions to the fiber organization captured by the OD marker. However, this analysis is limited by the uncertainty in dementia type ($n = 23$ were diagnosed with unspecified dementia).

To better isolate disease-specific effects, we investigated polygenic risk scores (PRS) indexing the genetic susceptibility to a certain disease[37], comparing high genetic risk (top 1%) and low genetic risk (bottom 50%) individuals (Fig. 4B). For cardiovascular disease and ischemic stroke, WMHs showed more severe alterations in high PRS individuals in anterior regions (average $\beta = 0.12$ and 0.18, respectively;

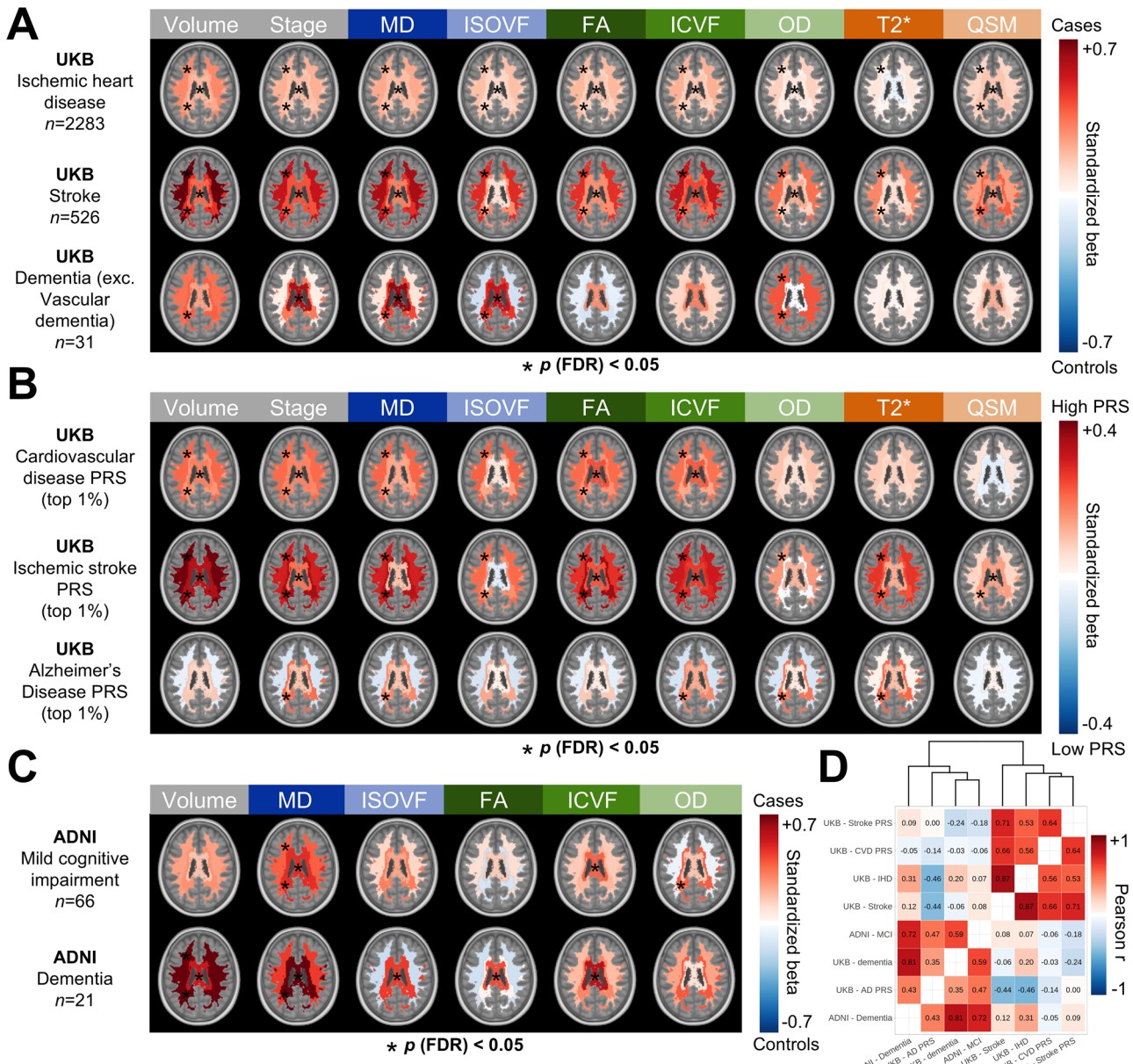

**Fig. 4 | Differences in WMH pathophysiology across neurodegenerative and vascular diseases.** Effect size patterns of spatiotemporal WMHs pathophysiology comparing cases and controls in (**A**) UK Biobank (UKB) participants according to ICD-10 code groupings of ischemic heart disease, stroke, and dementia, (**B**) UKB participants at high genetic risk of cardiovascular disease, ischemic stroke, and Alzheimer's disease, and (**C**) ADNI participants with mild cognitive impairment and dementia. Red colors indicate higher WMH burden in cases, and blue colors indicate higher WMH burden in controls. Significant effects at the FDR-corrected $p < 0.05$ level are indicated with black asterisks. **D** Effect size patterns were correlated and clustered with hierarchical clustering. Source data are provided as a Source Data file. *FDR* false discovery rate; *PRS* polygenic risk score.

$p$ (FDR) < 0.05 for all markers except OD, T2*, QSM, and for all markers except QSM, respectively) than posterior regions (average $\beta = 0.09$ and 0.17, respectively; $p$ (FDR) < 0.05 for WMH volume, stage, MD, FA, and for all markers except OD, respectively). For AD, WMHs showed more severe alterations in high PRS individuals in posterior regions (average $\beta = 0.08$; $p$ (FDR) < 0.05 for stage, MD, ICVF, OD, T2*) relative to anterior regions (average $\beta = 0.01$; no significant associations at $p$ (FDR) < 0.05). The inverse pattern of FA and OD effects in stroke and Alzheimer's PRS echoed findings from the diagnosis analysis. This analysis was replicated using different cut-off values to determine high genetic risk individuals (top 3% and top 5%; Supplementary Fig. 14).

We replicated the Alzheimer's-specific findings using ADNI3 data (Fig. 4C), which included fiber- and fluid-sensitive metrics but not myelin- and iron-sensitive metrics. Normative models were transferred

to compute WMH pathophysiology maps (Methods). Among individuals with amyloid positron emission tomography (PET) data, 28% of cognitively normal individuals, 53% of people with mild cognitive impairment, and 94% of people with dementia were amyloid-positive (demographic statistics detailed in Supplementary Table 2). Compared with cognitively normal control participants ($n = 125$), individuals with mild cognitive impairment (MCI; $n = 66$) and individuals with dementia ($n = 21$) showed a similar pattern of higher effect sizes in posterior WMHs, lower effect sizes for FA, and higher effect sizes for OD. In the dementia group, some effects did not reach significance, likely due to the lower sample size. For example, the posterior WMH OD effect size was almost identical in the MCI group ($\beta = 0.37$; $p = 0.01$) and dementia group ($\beta = 0.38$; $p = 0.08$), hence the difference in significance likely stems from differences in sample sizes.

We investigated sex by group interactions to investigate if some effects were higher in males or females, but no effect was significant at the $p$ (FDR) < 0.05 level (Supplementary Fig. 15).

We quantified the replicability across the eight reported effect size patterns (Fig. 4D), observing high correlations within vascular and neurodegenerative groupings, which clustered together (Fig. 4D). For example, the UK Biobank dementia signature was highly correlated with the ADNI dementia signature ($r = 0.81$). We additionally compared amyloid-positive MCI and dementia subjects ($n = 42$) to amyloid-negative cognitively normal subjects from ADNI (Supplementary Fig. 16A) in order to isolate biomarker-confirmed cases. Effects were similar to the other ADNI patterns and showed higher correlations with effect size patterns in the neurodegenerative grouping (average $r = 0.63$) compared to the vascular grouping (average $r = 0.32$; Supplementary Fig. 16B). These findings were also reproduced using a parcellation that was made without including rare WMH voxels using our search area strategy (Supplementary Fig. 17), with minimal differences that do not alter our conclusions. This demonstrated the reproducibility of the disease-specific WMH pathophysiological signatures, which clearly differed in vascular and neurodegenerative pathologies. However, these results remain correlational and require validation using gold-standard ex vivo methods.

As a proof of concept, we investigated whether a machine learning model could differentiate vascular from neurodegenerative pathologies based on our derived spatial and pathophysiological WMH data. We trained Lasso logistic regression models to classify individuals with stroke diagnoses ($n = 642$) from individuals with AD-related dementia or MCI diagnoses ($n = 127$), combining data from UKB and ADNI. WMH variables were additionally harmonized across datasets using ComBat (see Methods)[38]. Models were trained using 5-fold cross-validation repeated 100 times. We used 6 sets of predictors: WMH volumes only or WMH volumes and microstructure, either unparcellated, following the periventricular/deep parcellation, or using our custom clusters (distributions of area under the receiver operating characteristic curve (AUROC), balanced accuracy, and F1 score are available in Supplementary Fig. 18). Performance was compared using paired Wilcoxon tests. Unparcellated WMH volumes performed poorly in differentiating stroke from cognitive impairment (AUROC mean[SD]: 0.562[0.001]) and improved using clustered volumes (periventricular/deep: 0.630[0.005]; custom clusters: 0.666[0.006]). Performance was noticeably elevated when adding microstructural data (unparcellated: 0.653[0.007]; periventricular/deep: 0.697[0.009]; custom clusters: 0.706[0.011]). All performance differences were significant at $p$(Bonferroni) < 0.001. Results were similar for balanced accuracy and F1 score. Thus, we observed that (1) WMH microstructural information adds meaningful predictive value for disentangling pathologies and (2) our custom clusters performed better than the traditional periventricular/deep parcellation on this task.

### Posterior WMHs connect cortical regions prone to tau pathology

Next, we investigated the gray matter correlates of our spatial WMH clusters, with the hypothesis that Wallerian degeneration stemming from gray matter Alzheimer's pathology would affect the associated white matter fibers, which could be linked to our observed pattern of fiber disorganization in posterior WMHs in neurodegenerative disorders. Using a high-resolution tractogram derived from 1001 subjects from the Human Connectome Project[39], we extracted streamlines passing through each white matter spatial cluster and calculated normalized cortical connectivity profiles (Fig. 5A; Methods). Since larger ROIs will have more streamlines, and thus higher raw connectivity metrics, we further normalized the values by dividing by the number of streamlines of each ROI across all three clusters. This analysis showed that the posterior WMH region disproportionately connects inferior

temporal and occipital regions, while the anterior WMH region connects dorsal frontal regions.

The cortical patterns were compared with average maps of amyloid-beta (Aβ) and tau positron emission tomography (PET) standardized uptake value ratio (SUVR) in Aβ-positive individuals from the PREVENT-AD cohort (Fig. 5B)[40]. This dataset was chosen due to the high representation of individuals at early stages in the AD spectrum. We observed high correlations between cortical connectivity profiles of the posterior WMH region and the tau PET distribution ($r = 0.65$, $p$ (Bonferroni) = 0.011), which was significantly different from a spatial permutation-based null (Fig. 5C)[41]. No significant associations were observed for Aβ PET. These results demonstrate that the posterior WMH region structurally connects cortical regions prone to early tau pathology, but not amyloid pathology.

## Discussion

The interpretation and clinical management of WMHs are confounded by their differing vascular or neurodegenerative pathologies. This nuance is rarely taken into account, as there is no established framework or guidelines to distinguish these two processes in vivo. We hypothesized that the underlying pathological contributions to WMHs would be revealed by diverging spatiotemporal pathophysiological signatures captured by fluid-, fiber-, and myelin- and iron-sensitive microstructural MRI metrics. Leveraging a large sample size ($n = 32,526$), we estimated subject-level WMH pathophysiology and group-level spatiotemporal patterns, as well as derived disease-specific signatures in vascular and neurodegenerative disorders.

Our data-driven approach adds important nuance to the often-used periventricular/deep parcellation and revealed that deep-anterior WMHs had more severe pathophysiology than periventricular-posterior WMHs, which does not seem to be caused by spreading patterns. These spatial patterns were mostly influenced by the MD, ICVF, and ISOVF markers, and were a mix of different heuristic parcellations since they incorporate both the medial-to-lateral axis (similar to periventricular/deep) and the anterior-to-posterior axis (represented in the lobar parcellation). This anterior-to-posterior differentiation is consistent with other data-driven evidence based on WMH co-occurrence[12,21]. The investigation by Phuah et al.[12] used spectral clustering on WMH probability maps from FLAIR images, resulting in a 5-cluster solution and found that deep frontal WMHs were associated with hypertension and diabetes, while parietal WMHs were associated with amyloid SUVR. Habes et al.[21] used non-negative matrix factorization to uncover clusters of covarying WMH volumes and found that high blood pressure and female sex were associated with higher frontal WMH load, while AD genetic risk was associated with higher dorsal WMH load. Our approach to parsing WMH spatial heterogeneity extends this emerging data-driven literature by anchoring these spatial patterns in the underlying WMH pathophysiology. We note that our clustering solution was based on group averages to increase robustness, but future studies should explore clustering methods taking into account subject-level variance.

Temporally, WMHs followed a consistent pathophysiological cascade across individuals. In agreement with the prevailing hypothesis, water infiltration and inflammation seem to be early events, as reflected in the early MD abnormality[4,42–44]. While the early reduction in ICVF could point to axonal loss[28,45], other observations are most consistent with inflammatory swelling of cells, which could increase the extracellular signal fraction and thus decrease the intracellular signal fraction, resulting in the observed effect. The combination of positive T2* abnormality, consistent with our previous work[46], and very low QSM positive abnormality potentially indicates myelin and iron loss, as these two processes would have an additive positive impact on T2* but would cancel each other out on QSM[47,48]. While iron accumulation occurs in aging, especially in subcortical gray matter regions[49], iron

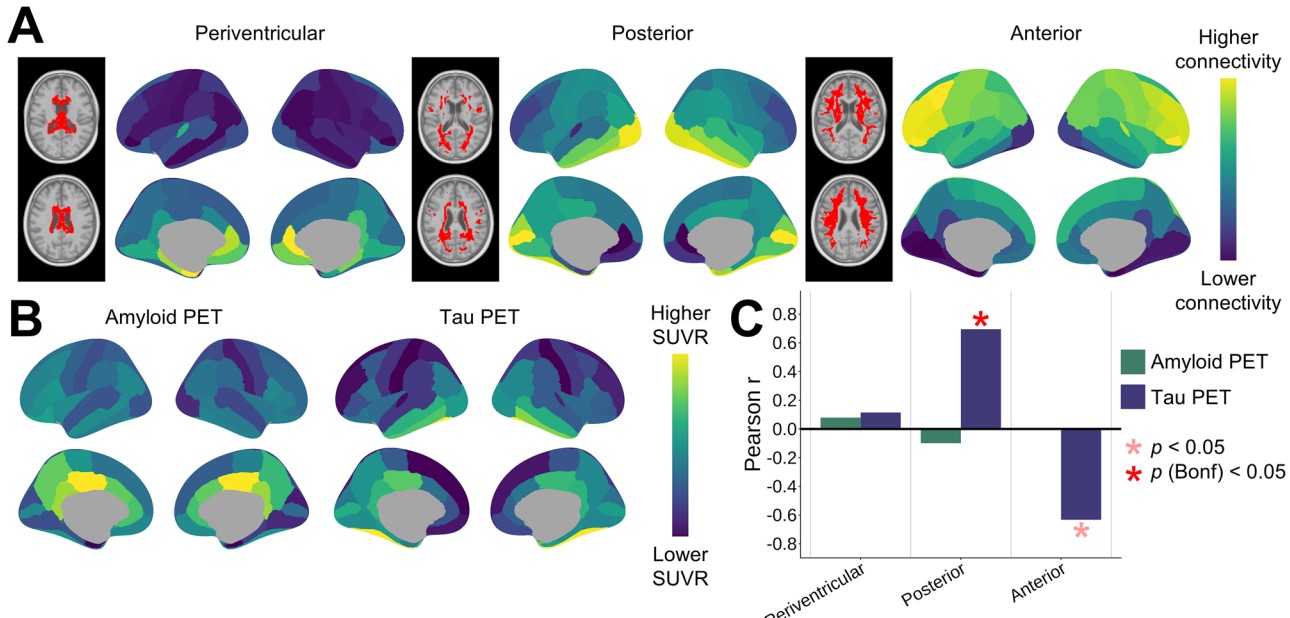

**Fig. 5 | Cortical connectivity profiles of WMH clusters and their associations with amyloid and tau PET distributions. A** The total amount of streamlines passing through the WMH spatial clusters and connecting each cortical gray matter region were calculated and normalized. Yellow regions indicate a disproportionately higher connectivity relative to the other clusters. **B** Group averages of amyloid and tau PET standardized uptake value ratio (SUVR) distributions in amyloid-positive individuals from the PREVENT-AD cohort. **C** Spatial Pearson correlations between cortical WMH patterns and PET averages. *p*-values were calculated from two-sided spin test null models preserving the spatial auto-correlation of cortical maps. Significant effects at the uncorrected and Bonferroni-corrected *p* < 0.05 levels are indicated with asterisks in shades of red (Posterior WMHs · Tau PET: *p* (Bonferroni) = 0.011; Anterior WMHs · Tau PET: *p* (uncorrected) = 0.043). Source data are provided as a Source Data file.

loss in white matter could be caused by the death of oligodendrocytes, which contain large amounts of paramagnetic iron[50]. As with all in vivo MRI studies, biological interpretations remain tentative and will need to be confirmed with gold standard ex vivo methods such as histology. Furthermore, we hypothesize that SuStaIn did not uncover subtrajectories because the subgroup of individuals with neurodegeneration-related WMHs could be very small and vastly overweighted by the subgroup with vascular-related WMHs (e.g., a few hundred versus 10 s of thousands). Another limitation of SuStaIn is that it depends on the sensitivity of the markers. It is possible that small deviations in fiber-sensitive metrics like OD are biologically meaningful (as evidenced by the differential OD effects in vascular and neurodegenerative diseases) but were overshadowed by less disease-specific markers that show very high abnormality (like the MD marker). Further analyses are needed to establish if our results are due to modeling limitations.

We uncovered a signature of neurodegeneration-related WMHs in individuals with dementia and AD spectrum disorders, showing a posterior dominance and a reproducible pattern of specific effects on fiber-sensitive metrics. This finding adds to the body of literature linking posterior WMHs to AD[12,13,51,52]. The pathophysiological signature suggests selective degeneration of fibers, since such processes would reduce the complexity of diffusion orientations as water would diffuse relatively more strongly in the direction of the spared fibers, explaining the observed lower OD and relatively preserved FA. This interpretation is further supported by our finding that posterior WMH regions structurally connect cortical regions prone to early tau accumulation. We posit that our observations point to a link between tau pathology in inferior temporal regions, degeneration of the associated fiber populations, and the appearance of posterior FLAIR hyperintensities. Cortical regions connecting through WMHs have been shown to be disproportionately affected relative to unconnected regions[53,54]. Further research is needed to test the causality of this hypothesis. Our prediction framework, where we classified individuals

with MCI and dementia from individuals with stroke, demonstrated that our approach using customized spatial regions and microstructural WMH information meaningfully outperformed the traditional approach of only using WMH volumes following the periventricular/deep parcellation (mean AUROC of 0.706 for the former and 0.630 for the latter). Still, the highest accuracy remained moderate, highlighting the inherent difficulty of the task, driven in part by the challenge of obtaining pure groups (i.e., a group with only AD pathology and no cerebrovascular pathology, and vice-versa) given the high degree of co-presence of the two disorders[55]. However, this analysis should be considered as a proof-of-concept since its primary goal was to demonstrate that WMH characteristics between the two groups systematically differ to the degree that this information can be used to classify the two disorders. In other words, we wanted to investigate this classification purely based on WMH characteristics, and therefore did not use more disease-specific markers (e.g., blood pressure, amyloid PET, whole brain features) that would undoubtedly improve the accuracy for practical applications.

The question remains as to why neurodegeneration would cause hyperintense signals on FLAIR images, which are mostly sensitive to increased fluid content. Our observations allow for a more informed hypothesis. In vascular-related WMHs, hypoxia/ischemia, potentially leading to blood-brain barrier dysfunction, may result in fluid leakage from the ventricles or the blood vessels, as evidenced by recent works[15,56]. In neurodegeneration-related WMHs, the death of axons and myelin (captured by the altered OD marker) could leave more space to be filled by extracellular fluid content or inflammatory cells, but without important fluid leakage from ventricles or blood vessels, which could explain the lower MD abnormality in posterior WMHs versus anterior WMHs. More hypotheses are discussed in a recent review[57].

While we cannot exclude the possibility that the neurodegenerative WMH pattern could be due to vascular dysfunction specific to AD, our interpretation is supported by both in vivo and ex vivo studies

showing the absence of ischemic cerebrovascular pathology in posterior WMHs in AD[11,12,58,59] (see reviews)[10,57]. Furthermore, we observe the same pattern in individuals at high genetic risk of AD, where vascular risk factors are likely similarly distributed relative to low genetic risk-individuals, and our findings are consistent with the reported early emergence of WMHs in autosomal-dominant AD[51,59]. Anterior WMHs have been associated with both neurodegeneration and small vessel disease[60], pointing to an overlap in pathology that cannot purely be distinguished by neuroanatomical location and supporting the need for WMH pathophysiological assessments in vivo, since both vascular and Alzheimer's pathology are frequently comorbid in the general population[55]. This is supported by our finding that WMH volume was significantly elevated relative to controls across all three spatial regions in both cerebrovascular disorders and dementia, highlighting the need for a deeper phenotyping of WMH pathophysiology to parse etiologies. We still encourage other groups to use our publicly available parcellation scheme (https://github.com/CoBrALab/WMH_patho_UKB), but note that this is informative but not sufficient to disentangle vascular from neurodegenerative contributions. Based on our observations, metrics derived from multi-shell diffusion MRI acquisitions in particular show promise for identifying WMH pathogenesis.

Investigating sex differences, we observed that females had lower WMH volumes but higher pathophysiological effects in the posterior region, and higher WMH volume but similar pathophysiological effects in the anterior region relative to males. This finding adds nuance to the reported higher WMH load in women[61], further highlighting the relevance of differentiating posterior and anterior WMHs. While this pattern could be due to the menopausal transition and the reduction in estrogen's neuroprotective effects[35], a recent investigation from our group found no differences in WMH volumes in women before and after the menopausal transition when accounting for the effect of age in UK Biobank[62], although examining region-specific effects using our custom parcellation would be an interesting follow-up. The observed higher WMH volumes in females in the more vascular-related anterior region has been previously reported by other groups[63]. While this is counterintuitive, given that cerebrovascular diseases are more prevalent in males, this could be due to the higher frequency of arterial stiffness in females[64]. The pattern of higher WMH volumes in males but worse pathophysiology in females in the more neurodegeneration-related posterior region is intriguing, and may be related to female-specific AD dynamics. For example, inflammatory microglial activation has been reported to play a more important role in the AD pathological cascade in females versus males, which could also impact WMH pathophysiology[65].

While our findings will require further validation before being integrated into clinical settings, using in vivo MRI-based pathophysiological markers to disentangle neurodegenerative and vascular contributions to WMHs could have broad implications for the clinical management of patients and the interpretation of research findings and clinical trials. Hypothetically, aggressively treating hypertension following a clinical radiological finding of large WMHs may not be warranted if these are mostly related to neurodegenerative processes and increase the risk of important side effects, including insomnia and vision problems[66]. Second, the use of amyloid-clearing drugs like lecanemab is currently restricted to individuals without comorbid cerebrovascular pathology, including a criteria of no significant white matter hyperintensities[67], since they are at higher risk of developing potentially severe amyloid-related imaging abnormalities (ARIA)[67]. Clarifying WMH pathogenesis could improve the selection of patients, potentially including people with neurodegeneration-related WMHs who could benefit most from these treatments. Third, identifying neurodegenerative WMHs could complement other AD-specific biomarkers, such as amyloid and tau protein markers, and help detect the disease at earlier stages, when treatments are more likely to be effective. Lastly, our framework could help with the differential diagnosis of vascular and Alzheimer's disease dementia.

While our findings are robust due to the large sample size and multiple sensitivity and replication analyses, our work also has some limitations. First, the interpretation of MRI microstructural metrics is complex and may not hold in different disease contexts[68]. In other words, there is no one-to-one mapping of any MRI metric to any specific pathophysiological process. Our multimodal approach combined metrics with different specificities and sensitivities to provide better-informed inferences of biological processes than any metric individually. In future work, we aim to confirm these biological interpretations by correlating microstructural MRI values within WMHs with histology and immunohistochemistry measures, leveraging the Douglas-Bell Canada Brain Bank, which contains over 3000 brain specimens[69]. Second, it remains unclear if and how Alzheimer's cortical pathology and our derived signature of neurodegeneration-related WMHs are causally linked. It will be critical to determine the direction of this association, with the possibilities of a white matter-first process (i.e., retrograde degeneration), gray matter-first process (i.e., anterograde Wallerian degeneration), or a mix of both.

In conclusion, this work demonstrates that moving beyond volumetric summaries of WMHs and incorporating microstructural MRI markers enables more specific separation of cerebrovascular and neurodegenerative pathologies. By modeling spatial WMH heterogeneity in a data-driven manner based on pathophysiology, we identified a posterior WMH signature in individuals on the AD spectrum, characterized by selective fiber alterations and anatomical connectivity to cortical regions vulnerable to early tau accumulation. While in vivo MRI cannot provide a definitive biological ground truth, the present results establish concrete, reproducible pathophysiological distinctions that extend beyond existing periventricular/deep classifications. This framework for measuring WMH pathophysiology using clinically translatable imaging tools provides a principled foundation for future mechanistic and clinical studies.

## Methods

### Study design and participants

The UK Biobank (UKB) dataset is a cohort study comprised of over 500,000 individuals from midlife to advanced age (application #45551). The study was approved by the North West Multicenter Research Ethics Committee (United Kingdom). A subset of ~40,000 participants underwent a comprehensive brain MRI protocol. Data from the UK Biobank is publicly available via material transfer agreements (https://www.ukbiobank.ac.uk/enable-your-research/register). The Alzheimer's Disease Neuroimaging Initiative (ADNI) dataset is a longitudinal observational study composed of participants along the Alzheimer's disease (AD) spectrum (http://adni-info.org/) and was approved by the institutional review board at each participating institution. We focused on the ADNI3 dataset since it included the multi-shell diffusion MRI acquisition necessary to derive NODDI metrics. Data from ADNI is publicly available via a data-sharing application (https://adni.loni.usc.edu/data-samples/adni-data/#AccessData). The Pre-symptomatic Evaluation of Experimental or Novel Treatments for Alzheimer Disease (PREVENT-AD) is a longitudinal study comprised of individuals who have parents or siblings diagnosed with AD and were cognitively normal at recruitment[40]. It was approved by the institutional review board at McGill University. Data from PREVENT-AD is publicly available via a data user agreement (https://openpreventad.loris.ca/). All participants gave written informed consent prior to participation in the studies. Sex was self-reported.

### Image processing

MRI acquisition parameters for UKB and ADNI3 are available in Supplementary Methods 1. Microstructural maps were processed by the

UKB team[24,70], and we processed ADNI microstructural maps ourselves, matching the UKB processing steps when possible. Processing steps to derive microstructural metrics from diffusion and susceptibility-weighted imaging are detailed in Supplementary Methods 2.

Since voxel-by-voxel correspondence and accurate segmentation of WMHs were crucial for our application, we updated key processing steps from UKB and applied those same steps to the ADNI dataset. T1w and FLAIR images were denoised[71], corrected for N3 inhomogeneities[72], and intensity normalized. Brain masks were calculated using the BEaST algorithm[73]. A UKB study-specific template space was generated using the processed T1w images of 200 subjects with representative age and sex distributions (available at https://github.com/CoBrALab/WMH_patho_UKB), and an ADNI template was generated similarly from all cross-sectional T1w images[25]. All modalities were then aligned at the subject level by performing rigid registration of other modalities (FLAIR, diffusion, and susceptibility) to the T1w subject space using the Advanced Normalization Tools software[74]. We then performed multispectral non-linear registration to the study-specific templates using preprocessed T1w images and fractional anisotropy maps supersampled to 1 mm isotropic[75] as inputs[74]. The ADNI template was non-linearly registered to the UKB template. All microstructural maps were transformed in UKB space using 4th-order B-spline interpolation at a 2 mm isotropic resolution to match the diffusion-weighted imaging resolution (of note, the susceptibility-weighted imaging acquisition had an original resolution of $0.8 \times 0.8 \times 3$ mm). The rich spatial contrast of fractional anisotropy maps allowed for increased registration accuracy of white matter tracts[76], a crucial aspect for our application. WMHs were segmented in native space with the Brain tIssue SegmentatiON (BISON) algorithm, a multi-contrast random forest classifier[15] that was retrained on 60 manually-labeled WMH masks on UKB data using both T1-weighted and FLAIR images as inputs.

## Quality control

The MRI data underwent multiple steps of quality control. We visually inspected T1w and FLAIR images according to guidelines established by our group and excluded scans showing significant motion artifacts (https://github.com/CoBrALab/documentation/wiki/Motion-Quality-Control-(QC)-Manual). Motion during diffusion MRI scanning was quantified using an automated tool[77] and scans with a mean absolute head motion (UKB field 24450) above 3 mm were excluded. Participants were further excluded if they failed any image processing steps or had a multiple sclerosis diagnosis according to the first-occurrence field (UKB field 131042). The final UKB sample was composed of 17,382 females (53.4%) and 15,144 males (46.6%) aged 45 to 81 years old (mean [SD] = 63.52 [7.49]; Supplementary Table 1). The final ADNI sample was composed of 123 females (58%) and 89 males (42%) aged 51 to 93 years old (mean [SD] = 73.41 [8.44]; Supplementary Table 2).

## Normative modeling and WMH pathophysiology

Normative modeling of NAWM was performed on a voxel-wise basis in the space of our UKB template for each of the seven markers only using UKB microstructural data (Fig. 1A). In other words, only individual-level voxels that were labeled as NAWM by the BISON algorithm were included in the normative models. We used Bayesian linear regression from the PCN toolkit/v0.35[29] with the covariates of age (modeled with 3rd-order B-splines with 4 knots) and sex. Age- and sex-specific maps of average and standard deviations of NAWM microstructure were then generated and used to z-score the microstructure of individual-level voxels labeled as WMH (example in Fig. 1B). This process crucially isolates the impact of the presence of a WMH on the microstructure and removes the spatial microstructural contrast driven by normal anatomical variations.

The normative model hyperparameters were then transferred to ADNI data in order to account for site and scanner variations. The adaptation sample to estimate the site effect was comprised of the first timepoint available of cognitively normal individuals within the UKB age range (45–81 years old; $n = 114$), which is deemed a large enough sample for accurate transferring of normative model hyperparameters[78]. For the ADNI subjects older than 81 years old ($n = 35$), pathophysiology maps were calculated from normative values of 81-year-olds in order to not extrapolate the models.

## Analyses

All linear models were processed in R/v4.1.2 using the lm function, scaling all continuous variables to obtain standardized beta coefficients. $p$-values were corrected using the false discovery rate (FDR)[36] on a per-analysis basis unless otherwise indicated. For example, in Fig. 4A, FDR correction is applied on 81 $p$-values (9 measures * 3 regions * 3 diagnoses).

## Spatial clustering

Between-subject averages of WMH pathophysiology for each of the seven microstructural markers were computed (Fig. 2A). Voxels with a low prevalence of WMH ( < 30) or NAWM ( < 5000) labels were excluded (32,424 excluded voxels out of 54,257 total voxels). This was necessary to (1) accurately model the NAWM microstructure trends using our Bayesian linear regression approach, and (2) obtain robust estimates of the WMH average pathophysiology. These thresholds were informed by prevalence by standardized mean error (SEM) plots (Supplementary Fig. 3), since we observed SEM plateaus starting at these cutoff values. While excluding voxels with low NAWM prevalence caused by the almost ubiquitous presence of WMH in older adults could induce a bias, WMHs in these areas (which are primarily in immediately periventricular regions, as seen in Supplementary Fig. 2) are likely benign and would not be informative for our goal of parsing heterogeneity. Included voxels were thus stable, robust estimates of the average WMH pathophysiology and still covered large portions of the white matter, permitted by the very large sample size. These maps were used as inputs to a spectral clustering algorithm using the Spectrum/v1.1 package in R[30] resulting in spatial clusters of pathophysiologically similar WMHs (Fig. 2B). The eigengap method[30] determined that the optimal number of clusters was 4 (Supplementary Fig. 4A); however, the three-cluster solution resulted in more spatially contiguous clusters with similar pathophysiological differentiation across clusters (Supplementary Fig. 4B). We calculated the variance explained ($R^2$) in the maps of between-subject averaged WMH pathophysiology (Fig. 2A) by the spatial clustering solution (Fig. 2B) for each microstructural metric using ANOVAs (Supplementary Fig. 5). We calculated the Dice similarity coefficient (DSC) overlaps between our derived spatial cluster and other often-used parcellations (Fig. 2C): periventricular/deep (derived manually by defining a distance from the ventricles of > 8 mm as deep), lobar[79,80], cerebral arterial territory[32], and an atlas of fiber type. We only included clustered voxels in the overlap calculations. The fiber atlas was derived using the high-resolution population-averaged streamline tractography obtained from the Fiber Data Hub, comprising 34 association, 2 commissural, and 34 projection fiber bundles derived from 1065 Human Connectome Project participants and processed using DSI Studio[81,82]. For each fiber bundle, a voxelwise tract density map was computed by calculating the tract density index (TDI) as the total number of streamlines in the bundle passing through each voxel. Within each fiber class (association, commissural, projection), voxelwise TDIs were aggregated by taking the maximum TDI across all constituent fiber bundles, yielding one class-specific density map per class. A three-class fiber atlas was then generated using a winner-take-all voting approach.

The clustering analysis was repeated in different subgroups: based on WMH volume (low: ≤ 5000 mm³, $n = 27,511$; medium: > 5000 and ≤ 10000 mm³, $n = 3103$; high: > 10000 mm³, $n = 1192$), excluding voxels with prevalence of WMH < 10 (Supplementary Fig. 6A), and

across sexes, excluding voxels with prevalence of WMH < 15 (Supplementary Fig. 7A). DSCs between the original clusters and subgroup clusters were calculated, only including voxels that were present in both clustering solutions in the calculation (Supplementary Fig. 6B and Supplementary Fig. 7B).

To derive a final parcellation covering all possible WMH locations (i.e., including voxels that were previously excluded based on low WMH or NAWM prevalence), we used a search area strategy: for each excluded voxel, we sampled the assigned clusters of the surrounding voxels and calculated the most prevalent cluster assignment. If the excluded voxel was not immediately surrounded by voxels with an assigned cluster, we extended the search area iteratively by steps of one voxel until there were voxels with an assigned cluster in the search area. This final parcellation covering the whole white matter is shown in Supplementary Fig. 8A and is made publicly available in both our custom UKB template space and in standard MNI ICBM152 09c symmetric space (https://github.com/CoBrALab/WMH_patho_UKB).

We then calculated region-of-interest measures of WMH pathophysiology by computing the median pathophysiological value for each metric within WMHs in each spatial cluster. In cases with fewer than 5 WMH voxels within a spatial cluster, data was imputed as 0, indicating no WMH-related microstructural abnormality. Region-wise WMH volumes were calculated in common space (inherently normalized for brain size) and log-transformed, as recommended[83]. To simplify the interpretation, we inverted the values of markers that showed negative effects in WMHs (FA, ICVF, and OD) so that an increase in z-scores is consistently interpreted as worsening WMH pathophysiology for all markers.

### Temporal modeling

To model the temporal progression of WMH pathophysiological events, we used the Subtype and Stage Inference (SuStaIn) technique[33,84], specifically the piecewise linear z-score model since our data was expressed as abnormality z-scores (Fig. 3A). Pathophysiological events were determined based on z-score thresholds ($Z = 0.5, 1, 1.5, 2, 3, 4, 5$) and the maximum z-score was determined empirically on a per-marker basis (at least 1% of participants needed to reach the threshold; Supplementary Fig. 9). We note that subjects without WMHs in a cluster (< 5 voxels) will have all 0 values and will be assigned the stage and subtype of 0, thereby not contributing to the variance in SuStaIn stages used in subsequent analyses. SuStaIn was performed using 10-fold cross-validation with 15 startpoints and 10,000 Monte Carlo Markov Chain resamples at each fold using pySuStaIn. We compared the one-subtype solutions to the three-subtype solutions and concluded that there was no evidence of sub-trajectories (Supplementary Fig. 10). We also modeled the relationships between WMH pathophysiology and volume (i.e., using WMH volume as an empirical proxy of WMH temporal progression). Non-linear relationships with log-transformed WMH volume were modeled with 4th-order B-splines (Supplementary Fig. 12A). Spearman correlations were calculated for associations between SuStaIn stages and regional WMH volumes (Supplementary Fig. 12B).

### Sex differences

Sex differences in WMH measures were investigated using linear models 1) for regional WMH volumes, correcting for age (to investigate simple sex differences; Fig. 3B), and 2) for WMH pathophysiological estimates, additionally correcting for region-specific WMH volume (to investigate differences in WMH pathophysiology when males and females have equivalent extents of WMHs; Fig. 3C).

### Associations with vascular and neurodegenerative disorders

UK Biobank diagnosis data was used through the first occurrences field (category #1712), which mapped data from self-reported medical conditions, primary care, hospital inpatient, and death register records

to ICD-10 codes together with a date of diagnosis. Control subjects were defined as subjects with no endocrine, circulatory, behavioral, nervous, or metabolic disease-related lifetime diagnoses ($n = 8528$). Case subjects were defined by groupings of ICD-10 codes (detailed in Supplementary Fig. 13) regardless of the timing relative to the MRI visit (i.e., prevalent and incident cases). We excluded subjects with comorbidities on the studied diagnoses (e.g., for dementia, we excluded individuals with a diagnosis of stroke or ischemic heart disease). We then compared WMHs between cases and controls using linear models while covarying for age and sex (Fig. 4A).

We investigated the associations between WMH pathophysiology and genetic susceptibility to vascular and neurodegenerative diseases, as indexed with polygenic risk scores (PRS) computed by UK Biobank[37]. We used the standard PRS calculated on all UK Biobank participants. Briefly, the authors used genome-wide association study (GWAS) meta-analysis results external to the UK Biobank to train trait-specific PRS algorithms and then calculated subject-wise PRS values as the sum of the per-variant effect size multiplied by allele dosage. The first 5 genetic principal components were used to obtain distributions with approximately zero means and unit variances per ancestry group. We used a similar case-control design as the diagnosis analysis, where controls were defined as individuals in the bottom 50% of a PRS and cases as individuals above the top 1% of a PRS. Groups were compared using linear models with no covariates (Fig. 4B). Results with other cutoff values for the case group (top 5% and top 3%) are presented in Supplementary Fig. 14.

For the ADNI replication analysis, our sample was composed of cognitively normal participants ($n = 125$), participants with a diagnosis of mild cognitive impairment ($n = 66$), and participants with a diagnosis of dementia ($n = 21$), excluding participants where the cause was determined to be due to non-AD factors. We kept the last timepoint available with the required MRI acquisitions for each participant in order to maximize the number of individuals diagnosed with mild cognitive impairment or dementia. Groups were compared with linear models controlling for age and sex (Fig. 4C). Amyloid and tau PET data availability in ADNI was insufficient to investigate associations with WMH markers with adequate statistical power separately by cognitive group (Amyloid PET: 127 controls, 75 MCI, 33 dementia; tau PET: 43 controls, 20 MCI, 9 dementia). However, we compared amyloid-positive MCI and dementia subjects combined ($n = 42$) to amyloid-negative cognitively normal subjects from ADNI ($n = 73$; Supplementary Fig. 16). Amyloid positivity was determined from PET data ([18 F] florbetaben (FBB) or [18 F] florbetapir (FBP)) using tracer-specific cerebellum-normalized cortical uptake SUVR thresholds[85].

Each analysis was repeated with a group by sex interaction term to investigate sex differences in the reported effects (Supplementary Fig. 15).

Each effect size pattern was correlated to all other patterns with Pearson's correlations, excluding modalities that were not present for the correlations with ADNI patterns. Hierarchical clustering was performed on the resulting correlation matrix using the *hclust* function in R (Fig. 4D).

We then used a machine learning classification framework to classify individuals with stroke from individuals with MCI or dementia purely based on their WMH characteristics (Supplementary Fig. 18). The input data was region-of-interest measures following three parcellations (unparcellated, periventricular/deep, custom clusters) derived as specified in section Spatial clustering, with either volumes only or volumes and microstructural variables. From UKB, we selected individuals with stroke or dementia diagnoses, excluding vascular dementia, as in the previous analyses, and removed overlapping participants. From ADNI, we selected participants with a clinical diagnosis of MCI or dementia, excluding participants where the cause was determined to be due to non-AD factors, as in the previous analysis. The WMH data (volumes and microstructure) were harmonized across

ADNI and UKB using the ComBat method with the neuroHarmonize/v2.4.5 python package[86]. Specifically, controls from UKB and ADNI were used to calculate site effects with the covariates of age and sex, and corrections were applied to the diseased groups. Lasso logistic regression models were fit using the LogisticRegression function from sklearn/v1.7.2 in Python (parameters: L1 penalty, C = 1, saga solver, balanced class weights (to address class imbalances), and max_iter = 1000). We performed 5-fold cross-validation, calculated performance metrics using all out-of-fold predictions, and repeated the process 100 times. Input data was normalized using a scaler fit on the train set and predicted on the test set at each fold. We did six runs across six sets of predictors as specified above and compared their performance using paired Wilcoxon tests using the rstatix R package, adjusting $p$-values with Bonferroni.

### Cortical gray matter connectivity profiles

To investigate the cortical regions that were structurally connecting through our three WMH spatial clusters, we used a high-resolution tractogram of 10 million streamlines derived from the Human Connectome Project dataset with diffusion MRI-based tractography[39,87]. We extracted the streamlines passing through each cluster using mrtrix3's tckedit command[88]. Using the CerebrA atlas (a precisely-registered and manually-corrected version of the MindBoggle101 atlas in MNI ICBM152 space)[89], connectivity matrices were constructed and summed (to calculate the total connectivity of each ROI) for each of the three streamline sets using the tck2connectome command. Since larger ROIs will have more streamlines, and thus higher raw connectivity metrics, we further normalized the values by dividing by the number of streamlines of each ROI across all three clusters (Fig. 5A).

To calculate AD pathology maps, we used the PREVENT-AD dataset, a longitudinal cohort of individuals with a parental history of AD who were cognitively unimpaired at recruitment[40], and calculated average maps of amyloid-beta (Aβ) and tau positron emission tomography (PET) standardized uptake value ratio (SUVR) in Aβ-positive individuals using cross-sectional data from the last available timepoint ($n = 35$). Aβ PET imaging was performed with the [$^{18}$F]NAV4694 (NAV) tracer, and tau PET was performed with the [$^{18}$F]AV1451 (flortaucipir) tracer. Processing details are available in McSweeney et al.[90]. Briefly, PET images were registered to each subject's T1w scan, masked to exclude CSF signal, and smoothed with a 6 mm³ Gaussian kernel. Standardized uptake value ratios (SUVRs) were normalized with cerebellum gray matter values for Aβ PET[91] and inferior cerebellum gray matter values for tau PET[92]. Aβ positivity was determined based on global Aβ SUVR in a data-driven fashion using Gaussian mixture modeling with two components[91]. PET volumetric maps were registered to MNI space and averaged. ROI values were extracted using the CerebrA atlas (Fig. 5B)[89].

We computed Pearson correlations between the cortical connectivity maps and averaged PET maps, performed spin tests to calculate 10,000 permutation-based nulls preserving the spatial autocorrelation[41], and derived $p$-values using the neuromaps package in Python (Fig. 5C)[93]. $p$-values were corrected with Bonferroni instead of FDR due to the lower amount of comparisons ($n = 6$).

### Reporting summary

Further information on research design is available in the Nature Portfolio Reporting Summary linked to this article.

## Data availability

Data from the UK Biobank is publicly available via material transfer agreements (https://www.ukbiobank.ac.uk/enable-your-research/register). Data from ADNI is publicly available via a data-sharing application (https://adni.loni.usc.edu/data-samples/adni-data/#AccessData). Data from PREVENT-AD is publicly available via a data user agreement (https://openpreventad.loris.ca/). Source data are provided in this paper.

## Code availability

We share all analysis code, results, and figures in raw form, and our derived UKB template space and WMH parcellation based on pathophysiology on GitHub (https://github.com/CoBrALab/WMH_patho_UKB)[94].

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

## Acknowledgements

We wish to thank all UK Biobank, PREVENT-AD, and ADNI participants and staff for their invaluable contributions. This research has been conducted using the UK Biobank Resource under Application Number 45551. This research used the NeuroHub infrastructure and was undertaken thanks in part to funding from the Canada First Research Excellence Fund, awarded through the Healthy Brains, Healthy Lives initiative at McGill University. The authors used the Digital Research Alliance of Canada (https://alliancecan.ca/en) computational resources in this work. O.P. is funded by the Alzheimer Society of Canada and the Fonds de Recherche du Québec - Santé (FRQS). MMC receives salary support from the Fonds de Recherche Québec – Santé and from a James McGill Professorship. MMC also receives research support from the Canadian Institutes of Health Research, Natural Sciences and Engineering Research Council – Canada, McGill University's Health Brains for Health Lives (a Canada Research Excellence Fund Initiative), and TRIDENT (a New Frontiers in Research Fund program). M.D. receives salary support from the Fonds de Recherche Québec – Santé and research support from the Canadian Institutes of Health Research, Natural Sciences and Engineering Research Council – Canada, and Brain Canada. Z.A. is funded by the Canada Brain Research Fund (CBRF) Rising Star program. MC is funded by the Fonds de Recherche du Québec - Santé (FRQS). AB received support from the Alzheimer Society of Canada and Healthy Brains Healthy Lives. The PREVENT-AD program was launched in 2011 as a $13.5 million, 7-year public-private partnership using funds provided by McGill University, the Fonds de Recherche du Québec – Santé (FRQ-

356162), an unrestricted research grant from Pfizer Canada, the J.L. Levesque Foundation, the Douglas Hospital Research Center and Foundation, the Government of Canada, the Canada Fund for Innovation, the Canadian Institutes of Health Research (SV: 178385, JP: 153287, 178210, LC: 165921, TS: 175328,) the Alzheimer Society of Canada, the National Institutes of Health of the United States (NS: NIH AG068563), the Alzheimer Association (SB: AARG-NTF 926696) and Brain Canada Foundation. A complete listing of ADNI investigators can be found at: http://adni.loni.usc.edu/wp-content/uploads/how_to_apply/ADNI_ Acknowledgement_List.pdf. A complete list of PREVENT-AD investigators can be found at: https://preventad.loris.ca/acknowledgements/ acknowledgements.php?DR=8.0&authors. Data used in preparation of this article were obtained, in part, from the Alzheimer's Disease Neuroimaging Initiative (ADNI) database (adni.loni.usc.edu) and the Presymptomatic Evaluation of Novel or Experimental Treatments (PREVENT-AD) program (https://www.centre-stopad.com/). As such, the investigators within the ADNI and PREVENT-AD contributed to the design and implementation of ADNI/PREVENT-AD and/or provided data, but most investigators did not participate in the analysis or writing of this report.

## Author contributions

O.P.: conceptualization, data curation, formal analysis, investigation, methodology, resources, software, visualization, writing – original draft preparation, writing – original draft preparation, writing – review & editing. Z.A.: formal analysis, methodology, software, visualization, writing – review & editing. S.O.: formal analysis, software, visualization, writing – review & editing. A.B.: methodology, writing – review & editing. M.C.: data curation, writing – review & editing. J.P.F.: methodology, writing – review & editing. D.Q.: data curation, writing – review & editing. A. Pastor-Bernier: methodology, writing – review & editing. A.F.V.: methodology, writing – review & editing. A. Pichet-Binette: conceptualization, writing – review & editing. A.M.: methodology, writing – review & editing. J.M.: methodology, writing – review & editing. G.A.D.: methodology, resources, writing – review & editing. C.J.S.: methodology, writing – review & editing. S.V.: methodology, writing – review & editing. M.D.: conceptualization, project administration, resources, software, supervision, writing – review & editing. M.M.C.: conceptualization, funding acquisition, project administration, resources, supervision, writing – review & editing.

## Competing interests

The authors declare no competing interests

## Additional information

[1]Cerebral Imaging Centre, Douglas Mental Health University Institute, Montreal, Canada. [2]Integrated Program in Neuroscience, McGill University, Montreal, Canada. [3]Center for the Studies in the Prevention of Alzheimer's Disease, Douglas Mental Health University Institute, Montreal, Canada. [4]Centre d'études avancées en médecine du sommeil, Université de Montréal, Montreal, Canada. [5]Département de pharmacologie et physiologie, Université de Montréal, Montreal, Canada. [6]Centre de recherche de l'Institut universitaire de gériatrie de Montréal, Montreal, Canada. [7]Department of Anatomy and Cell Biology, McGill University, Montreal, Canada. [8]Département d'Anatomie, Université du Québec à Trois-Rivières, Trois-Rivières, Canada. [9]Department of Neurology and Neurosurgery, McGill University, Montreal, Canada. [10]Department of Psychiatry, McGill University, Montreal, Canada. [11]Department of Psychology, Concordia University, Montreal, Canada. [12]Department of Neurology, Max Planck Institute for Human Cognitive and Brain Sciences, Leipzig, Germany. [13]McConnell Brain Imaging Centre, Montreal Neurological Institute, Montreal, Canada. [14]Department of Biomedical Engineering, McGill University, Montreal, Canada. [15]These authors contributed equally: Mahsa Dadar, M. Mallar Chakravarty. ✉e-mail: olivier.parent@mail.mcgill.ca; mahsa.dadar@mcgill.ca; mallar.chakravarty@mcgill.ca

## the PREVENT-AD Research Group

Alexa Pichet-Binette[5,6], Sylvia Villeneuve ⓘ [3,10,13], Mahsa Dadar[1,2,10,15] ✉ & M. Mallar Chakravarty[1,2,10,14,15] ✉

