## [Transparent Peer Review file · Nature Communications]

Characterizing spatiotemporal white matter hyperintensity pathophysiology in vivo to disentangle vascular and neurodegenerative contributions

Corresponding Author: Mr Olivier Parent

Version 0:

Reviewer comments:

Reviewer #1

(Remarks to the Author)

Parente and colleagues assessed multiple MRI parameters within areas of WMH in a large sample of participants from the UK Biobank ($n > 300,000$) to test whether WMH cluster into spatially and microstructurally heterogeneous subtypes. The rationale was that combining multiple quantitative MRI measures (diffusion and susceptibility measures) may reveal the underlying biological substrates of WMH and thus detect different subtypes, such as those more characteristic of vascular vs. neurodegenerative etiology. The authors employed spectral clustering to derive WMH clusters and plotted the z-values of abnormality for each MRI measure. Pseudotemporal evolution was estimated using event-based modeling. MRI measures were subsequently compared between control subjects and different diagnostic groups in the UK Biobank, as well as across levels of PRS scores for cardiovascular/stroke and AD. Furthermore, MR measures were compared between AD and control groups in an external dataset from the ADNI study. Spatial associations between WMH areas of each cluster and typical tau deposition were assessed based on group-level connectivity analyses. In addition, sex differences were investigated. Results showed three clusters of WMH: periventricular, posterior periventricular, and anterior-deep white matter types. Each cluster was characterized by distinct z-score profiles across MRI parameters, suggesting microstructural alterations specific to different WMH types. Event-based modeling with SuStaln indicated a similar temporal ordering of the development of abnormalities among the different MR measures, with mean diffusivity and intracellular volume fraction (ICVF) appearing first. Comparisons of MR parameters across groups for each MRI measure revealed disease-dependent differences. Overall, this is a comprehensive study involving multi-parameter characterization of microstructural alterations in WMH. The approach is innovative is a step forward from simply mapping the spatial locations to different diseases, given that the underlying microstructural changes can be heterogeneous as revealed in prior post-mortem studies. Yet the conclusiveness of the findings remains somewhat unclear.

Comments

1. A concern is that the MRI measures assessed in the current study cannot be unambiguously linked to the pathophysiological events underlying WMH, and the chosen approach may not resolve these uncertainties. In the introduction, the authors state that “fluid infiltration from ventricles and blood vessels triggers inflammation, leading to tissue damage such as demyelination, oligodendrocyte loss, and, in more severe cases, axonal degeneration,” which may lead to WMH. They also suggest that multiparametric MRI provides homologs of post-mortem brain stains and can therefore reveal WMH pathophysiology. However, MR measures such as FA, MD, OD, ICVF, ISOVF, T2*, and QSM are ambiguous in terms of the underlying substrate when considered individually. It remains unclear how combining these measures resolves such ambiguities, especially when nearly all indices show alterations, as observed in this study. For example, free water (edema), fiber loss, macromolecules (inflammation), or myelin loss may all affect MD, FA, or OD. The authors are advised to critically discuss these issues and moderate their conclusions.
2. Visual inspection of cluster 1 suggests that it covers a small rim around the ventricles, an area prone to partial volume effects (PVE), which may result from inadequate segmentation and spatial normalization. Additionally, for this cluster, z-scores for most MR measures are close to zero, making interpretation difficult. Did the authors exclude voxels directly adjacent to ventricles to minimize PVE or took other measures to reduce PVE?
3. Clusters two and three show the same pattern of alterations in MRI measures but differ quantitatively in z-scores and spatial location. Do the authors conclude that these two signatures reflect similar etiology or underlying pathophysiology?
4. The analysis of group differences and associations with PRS scores remains unclear.
 - a. Conceptually, what was the rationale for comparing groups across each MRI measure separately after identifying cluster-

specific MRI signatures? Why not test whether patients with a particular disease are more strongly associated with specific WMH clusters? Such an analysis could clarify whether WMH clusters and their microstructural alterations are disease-specific.

b. Technically, it is unclear how statistical analyses and corresponding FDR corrections were performed. For example, for ischemic heart disease the authors report: “WMH alterations were small but significant, with stronger effects in anterior regions (average beta = 0.19; p (FDR) < 0.05 for all markers) than posterior regions (average beta = 0.13; p (FDR) < 0.05 for all markers except OD and T2*.)” Does that mean a voxel-based analysis was performed for each MRI marker, with beta coefficients subsequently averaged across all markers within each cluster mask? What exactly does the FDR correction refer to?

c. For controls, several diseases were defined as exclusion criteria. Were the same criteria applied to disease cases (apart from the disease of interest)?

5. Related to point 4: Figure 4: What determined the location of the asterisks?

6. The authors highlight OD differences in MCI/AD, yet alterations in other indices were observed as well. Furthermore, OD differences were seen in MCI but not in AD dementia. How do the authors interpret these findings?

7. Why dichotomize PRS when it is inherently a continuous measure? Dichotomization introduces arbitrary cut-offs, reduces statistical power, and, as in the present study, results in exclusion of a substantial amount of data.

8. Why were no covariates controlled for when comparing AD vs. controls in the ADNI sample? Were the MCI subjects in ADNI amyloid-positive?

9. Overall, the number of AD/dementia cases is small, and in the UK Biobank includes mixed dementia types. The results should be interpreted with caution.

10. How do the authors interpret the observed sex effects? Were sex differences in MRI measures evident across diagnostic groups?

(Remarks on code availability)

Reviewer #2

(Remarks to the Author)

I thank the authors for the considerable effort invested in this manuscript. From the very first read, it was clear to me that this work is the product of years of dedicated research. I am thus sincerely grateful to both the authors and the editors for the opportunity and privilege to review this wonderful piece of work. To match the authors' dedication, I have sought to provide feedback that was both thorough and (hopefully) constructively critical.

Line of argumentation. The authors argue that attributing white matter hyperintensities (WMH) to a vascular origin, as the expression “of presumed vascular origin” does, neglects their heterogeneous origins. While I partly agree with the authors' line of argumentation, I would also contend that the qualifier itself is not the core problem. The expression “of presumed vascular origin” does not categorically assert a vascular cause for all WMH, but rather implies that some—though not necessarily all—may arise from vascular mechanisms. The authors of the STRIVE criteria themselves acknowledged that, although the vascular hypothesis provided a plausible explanation for the origin of WMH, WMH were possibly multifactorial and emphasised that the qualifier “of presumed vascular origin” was a way to distinguish CSVD-related WMH from similar lesions in disorders such as multiple sclerosis and leukodystrophies (see page 826 of Wardlaw et al. (2013) for exact phrasing). Instead, I would argue that the core limitation of STRIVE's definition lies in its broad characterisation of WMH of presumed vascular origin as T2-weighted MRI hyperintense white matter regions, without providing additional criteria to reliably differentiate vascular from non-vascular aetiologies. The leap from “WMH of presumed vascular origin” to assuming that all WMH fall under this vascular category is another issue entirely. My main recommendation for the manuscript is to move from this initial line of argumentation and place greater emphasis on the subtyping of WMH per se. Sure, WMH are multifactorial, but how can we meaningfully differentiate between their underlying causes in vivo and develop robust methods to quantify them accordingly? This is exactly what the authors aimed to accomplish: to define WMH subtypes using multimodal data, assess their clinical relevance, and try to determine how the underlying processes unfold—and whether these differ across the identified subtypes.

Acknowledging previous work. The second paragraph of the introduction provides context by highlighting recent findings suggesting that the observed heterogeneity may, to some extent, be connected to the anatomical location of WMH within the brain. There might be a limit on the number of references, but several claims in this section would really benefit from a more comprehensive acknowledgment of prior work and accurate phrasing. To the best of my knowledge, the earliest work to distinguish WMH based on their anatomical location is the same study that introduced what is now known as the Fazekas scale (Fazekas et al., 1987). Although based on a small sample, this visual rating offered some of the earliest neuroimaging evidence suggesting that anatomical location relates to the heterogeneity of WMH. However, as later discussed by Kim et al. (Kim et al., 2008), this distinction requires “complementary rules” to effectively classify periventricular and deep WMH, as both the terminology, definitions, and implementations used were inconsistently—the limitation the authors are also highlighting. While I agree with this point, I would like to point out that this is not the only approach to study WMH; other approaches include, for instance, the increasingly popular assessment of the so-called multispot WMH pattern for CAA diagnosis (Charidimou et al., 2022; Charidimou & Baron, 2025) as well as lobe-wise and voxel-wise analyses (see (Botz et al., 2023) for a systematic revision of the literature up to 2023). To ensure accuracy, I kindly encourage the authors to broaden their current literature review to acknowledge prior work in the field.

Novelty claims. I find it difficult to understand why the authors claim that “no study has used unbiased, data-driven methods to systematically investigate spatial patterns of WMH pathophysiology”. In recent years, we have seen a couple of papers using data-driven clustering methods for examining spatial WMH patterns, e.g., (Bachmann et al., 2024; Phuah et al., 2022; Rosbergen et al., 2024). I understand that spatial patterns and “pathophysiology” are not synonymous—the former relates to spatial distribution, while the latter concerns microstructural integrity. Nonetheless, there is already a paper published last

year that incorporates microstructural measurements into the analysis (Rosbergen et al., 2024). One can say that this one has more metrics, but I wonder whether that would be enough to claim novelty. The term “unbiased” should not be used lightly here—it represents a substantial claim that, to the best of my knowledge (and apologies if I have overlooked something), has not been adequately verified. Since the methods are trained on specific datasets acquired using particular protocols and targeting certain populations, inherent biases are inevitable. Using a large cohort for training does not, in itself, guarantee the absence of bias. One thing is to show that patterns generalise and a completely different one that they are unbiased. The use of the term “systematic” is also problematic and may work against the authors, as it requires strong experimental justification. In this context, “systematic” would imply that the authors, for instance, investigated the minimal set of sequences needed to achieve the observed subtyping (i.e., whether all sequences are essential or if some can be omitted) and assessed the sensitivity of the subtyping to each subset of images (e.g., number of groups). However, I could not find any evidence supporting this “systematic” aspect. Apologies if I have overlooked it. Overall, I believe the current claims of novelty are not sufficiently substantiated and would benefit from revision. To be clear, I am not questioning the quality of the work—it is undoubtedly a strong contribution—but rather pointing out that the way it is currently described lacks precision and does not fully reflect the actual work presented.

MRI vs post-mortem. In the introduction, the authors suggest that multi-contrast microstructural MRI sequences can serve as “homologs” of post-mortem stains. I’m not entirely convinced by this claim. While microstructural measurements undoubtedly bring us closer to understanding the underlying nature of these lesions, but they are not true homologs in the strict biological or pathological sense. MRI signals are indirect, sensitive to a mix of tissue properties (e.g., water content, diffusion, susceptibility), and their interpretation is often ambiguous and context-dependent (Both table 1 and the discussion actually talk about this; see also (Figley et al., 2022) for more references and further discussion on the topic). It would be more accurate (and cautious) for authors to say that multi-contrast microstructural MRI can approximate or reflect some aspects of histopathology—rather than calling them “homologs” which implies a much stronger equivalence than is currently supported by evidence—which is not the aim of the study.

Estimation of WMH pathophysiology. The authors generate age- and sex-specific normative maps and compare each individual image against them to produce personalised abnormality maps. I find this approach interesting, as it directs the clustering towards individual abnormalities rather than shared features that might be more closely linked to ageing. My main concern, however, lies in how this is operationalised: the authors appear to assume that voxels labelled as normal-appearing white matter (NAWM) represent truly healthy measurements. This is a strong assumption, considering that the distinction between normal and abnormal is highly context-dependent and often lacks a clear boundary. In segmentation algorithms, for instance, the choice of threshold has effects on the segmentation of WMH and, by extension, on NAWM. A more stringent threshold tends to result in undersegmentation of WMH, while a more lenient one leads to oversegmentation. As a consequence, the definition of NAWM can vary depending on the selected threshold. At the same time, I believe the authors would agree that the abnormality associated with WMH extends beyond the visibly affected tissue, esp. in microstructural sequences (Duering et al., 2023; Ter Telgte et al., 2018). I thus truly wonder whether the assumption is correct. Could the authors please elaborate further on the correctness of their assumptions?

Assumptions in spectral clustering. The spectral clustering results reveal regions that align with previous findings in the literature, with anterior WMH more closely associated with vascular problems and posterior WMH more linked to neurodegenerative processes. This is a valuable contribution to the literature, as it provides large-scale validation of previously observed patterns. Interestingly, even the authors appeared somewhat surprised that, despite using multiple imaging modalities, the clusters emerged in just three specific brain regions. Given that these patterns largely mirror those identified in earlier unimodal studies, I wonder whether the inclusion of multiple modalities—which is one of the key points of the work—offered any substantial value. This ties back to my earlier point about the need for a systematic evaluation of the proposed approach. It would be highly beneficial for the authors to thoroughly assess the individual contribution of each modality to the clustering itself. The spatial layout of the clusters is to some extent unsurprising. First, the authors selected a three-cluster solution rather than four, prioritising a pattern in which the clusters remained more spatially coherent. Second, it is well known that white-matter microstructure differs across brain regions (Brander et al., 2010; Lawrence et al., 2021), because local fibre architecture varies—from highly aligned, single-direction bundles to regions with dense fibre crossings. Could this very anatomical structure, by design, influence how the clustering unfolds? It is also interesting to see that apparently there are no differences in the “temporal dynamics”. If they unfold in the same manner, what distinguishes them from each other?

Note: While I talk about many points raised in the introduction, it is clear that many of them apply equally to the discussion.

Recommendations. Assuming these patterns are indeed valid, the pertinent issue becomes their practical implementation. Must we acquire every imaging modality used in the study, or would a carefully selected subset—or simply applying the resulting cluster masks—be adequate? The authors argue that the prevailing misinterpretation of WMH as being of vascular origin poses a significant obstacle to implementing precision medicine in both diagnosis and treatment. I share this view—if a patient’s WMH do not stem from a vascular cause, prioritising the management of cardiovascular risk factors may not be a rational approach. Since this is a recurring theme throughout the manuscript, I would encourage the authors to verbalise it further. What are the concrete benefits of understanding the aetiology of WMH?

Minor

- Introduction – “[...] For example, ... Wallerian degeneration [...]”: it is clear that not all WMH are a product of hypoperfusion, they are multifactorial. Could the authors please elaborate on the importance of this example?
- I wonder whether it is necessary to reference some of the hypotheses mentioned in lines 119 to 121, as it is already clear at that point that these represent only one of several possible pathways through which WMH may develop.
- Introduction - “[...] most common brain radiological findings [...]”: brain radiological -> neuroradiological
- Introduction - “[...] They appear in most [...]”: perhaps “are present” instead.
- Introduction - “[...] are typically detected [...]”: “appear” perhaps
- Discussion – The account of how the process unfolds still needs sharper focus, subtler reasoning, and a well-curated thesis. With so much work already completed, what new understanding can we add about the breakdown of white-matter

integrity that has not yet been articulated?

References

- Bachmann, D., von Rickenbach, B., Buchmann, A., Hüllner, M., Zuber, I., Studer, S., Saake, A., Rauen, K., Gruber, E., Nitsch, R. M., Hock, C., Treyer, V., & Gietl, A. (2024). White matter hyperintensity patterns: associations with comorbidities, amyloid, and cognition. *Alzheimer's Research and Therapy*, 16(1). <https://doi.org/10.1186/s13195-024-01435-6>
- Botz, J., Lohner, V., & Schirmer, M. D. (2023). Spatial Patterns of White Matter Hyperintensities: A Systematic Review. May, 1–13. <https://doi.org/10.3389/fnagi.2023.1165324>
- Brander, A., Kataja, A., Saastamoinen, A., Ryymin, P., Huhtala, H., Öhman, J., Soimakallio, S., & Dastidar, P. (2010). Diffusion tensor imaging of the brain in a healthy adult population: Normative values and measurement reproducibility at 3 T and 1.5 T. *Acta Radiologica*, 51(7), 800–807. <https://doi.org/10.3109/02841851.2010.495351>
- Charidimou, A., & Baron, J. C. (2025). White Matter Hyperintensity Multispot Pattern Lesions and Cerebrovascular Amyloid Burden in Cerebral Amyloid Angiopathy. *Stroke*. <https://doi.org/10.1161/STROKEAHA.125.051482>
- Charidimou, A., Boulouis, G., Pasi, M., Gokcal, E., Martinez-Ramirez, S., Warren, A. B., Guroi, M. E., Viswanathan, A., Greenberg, S. M., Albuchoer, J. F., Charidimou, A., Boulouis, G., Frosch, M. P., Baron, J.-C., Pasi, M., Francois Albuchoer, J., Banerjee, G., Barbato, C., Bonneville, F., ... Greenberg, S. M. (2022). The Boston criteria version 2.0 for cerebral amyloid angiopathy: a multicentre, retrospective, MRI-neuropathology diagnostic accuracy study. *Lancet Neurol*, 21, 714–739. www.thelancet.com/neurology
- Duering, M., Biessels, G. J., Brodtmann, A., Chen, C., Cordonnier, C., de Leeuw, F.-E., Debette, S., Frayne, R., Jouvent, E., Rost, N. S., ter Telgte, A., Al-Shahi Salman, R., Backes, W. H., Bae, H.-J., Brown, R., Chabriat, H., De Luca, A., DeCarli, C., Dewenter, A., ... Wardlaw, J. M. (2023). Neuroimaging standards for research into small vessel disease—advances since 2013. *The Lancet Neurology*, 22(7), 602–618. [https://doi.org/10.1016/S1474-4422\(23\)00131-X](https://doi.org/10.1016/S1474-4422(23)00131-X)
- Fazekas, F., Chawluk, J. B., Alavi, A., Hurtig, H. I., & Zimmerman, R. A. (1987). MR Signal Abnormalities at 1.5 T in Alzheimer's Dementia and Normal Aging. *American Journal of Roentgenology*, 149(2), 223–441. www.ajronline.org
- Figley, C. R., Uddin, M. N., Wong, K., Kornelsen, J., Puig, J., & Figley, T. D. (2022). Potential Pitfalls of Using Fractional Anisotropy, Axial Diffusivity, and Radial Diffusivity as Biomarkers of Cerebral White Matter Microstructure. In *Frontiers in Neuroscience* (Vol. 15). Frontiers Media S.A. <https://doi.org/10.3389/fnins.2021.799576>
- Kim, K. W., MacFall, J. R., & Payne, M. E. (2008). Classification of White Matter Lesions on Magnetic Resonance Imaging in Elderly Persons. *Biological Psychiatry*, 64(4), 273–280. <https://doi.org/10.1016/j.biopsych.2008.03.024>
- Lawrence, K. E., Nabulsi, L., Santhalingam, V., Abaryan, Z., Villalon-Reina, J. E., Nir, T. M., Ba Gari, I., Zhu, A. H., Haddad, E., Muir, A. M., Laltoo, E., Jahanshad, N., & Thompson, P. M. (2021). Age and sex effects on advanced white matter microstructure measures in 15,628 older adults: A UK biobank study. *Brain Imaging and Behavior*, 15(6), 2813–2823. <https://doi.org/10.1007/s11682-021-00548-y>
- Phuah, C., Chen, Y., Strain, J. F., & Yechoor, N. (2022). Association of Data-Driven White Matter Hyperintensity Spatial Signatures With Distinct Cerebral Small Vessel Disease Etiologies. 0, 2535–2548. <https://doi.org/10.1212/WNL.0000000000201186>
- Rosbergen, M. T., Wolters, F. J., Vinke, E. J., Mattace-Raso, F. U. S., Roshchupkin, G. V., Ikram, M. A., & Vernooij, M. W. (2024). Cluster-Based White Matter Signatures and the Risk of Dementia, Stroke, and Mortality in Community-Dwelling Adults. *Neurology*, 103(7). <https://doi.org/10.1212/WNL.0000000000209864>
- Ter Telgte, A., Van Leijsen, E. M. C., Wiegertjes, K., Klijn, C. J. M., Tuladhar, A. M., & De Leeuw, F. E. (2018). Cerebral small vessel disease: From a focal to a global perspective. *Nature Reviews Neurology*, 14(7), 387–398. <https://doi.org/10.1038/s41582-018-0014-y>
- Wardlaw, J. M., Smith, E. E., Biessels, G. J., Cordonnier, C., Fazekas, F., Frayne, R., Lindley, R. I., O'Brien, J. T., Barkhof, F., Benavente, O. R., Black, S. E., Brayne, C., Breteler, M., Chabriat, H., DeCarli, C., de Leeuw, F. E., Doubal, F., Duering, M., Fox, N. C., ... Dichgans, M. (2013). Neuroimaging standards for research into small vessel disease and its contribution to ageing and neurodegeneration. *The Lancet Neurology*, 12(8), 822–838. [https://doi.org/10.1016/S1474-4422\(13\)70124-8](https://doi.org/10.1016/S1474-4422(13)70124-8)

(Remarks on code availability)

Reviewer #3

(Remarks to the Author)

The study aims to disentangle vascular and neurodegenerative contributions to white matter hyperintensities (WMHs), which are commonly interpreted as markers of cerebral small vessel disease but may also reflect neurodegenerative processes. Traditional FLAIR imaging lacks specificity to distinguish these mechanisms in vivo. To address this, the authors propose a multimodal MRI framework combining seven microstructural markers with voxel-wise normative modelling to generate subject-level pathophysiology maps.

Main comments:

Introduction:

1. Lines 95–98: The authors appear to misinterpret the meaning of the nomenclature “of presumed vascular origins.” This terminology was adopted because the primary underlying pathology—cerebral small vessel disease—cannot be directly observed in vivo, rather than to imply variability in the pathophysiological mechanisms of WMHs. This fact is another crucial point, that the authors also should acknowledge.
2. Lines 115–117: the authors claim that no study has attempted to investigate spatial pattern of WMH pathophysiology. This is only partially true, as there are indeed studies finding different spatial WMH pathology related to pathophysiological underpinnings and underlying pathology, as the authors acknowledge in the discussion section. A more complete contextualisation at this stage would probably be more valuable.

Methods and Results:

3. The central construct—"pathophysiology" defined as the z scored deviation from normative NAWM—relies on the assumption that NAWM is sufficiently disease free to serve as a reference. This is unlikely to hold, as cerebral small vessel disease is diffuse and not confined to WMHs. Failing to account for this risks biasing estimates toward zero, particularly in participants with high WMH burden, and the magnitude of this bias is difficult to predict given the unknown contribution of each metric. A sensitivity analysis excluding individuals with high WMH burden could strengthen confidence in the approach and this should be acknowledged in the discussion and/or the limitations.
4. Clusters 2 (posterior) and 3 (anterior) show very similar pathophysiological profiles, differing mainly in the magnitude of abnormality rather than in distinct marker patterns. This similarity is further reinforced by the SuStaln modeling, which shows nearly identical pseudo-time progression of biomarkers for these two clusters. This raises concerns about whether the clustering truly captures biologically meaningful heterogeneity or simply partitions WMHs by spatial location.
5. The clustering approach is based on a population-averaged pathophysiology map restricted to voxels with high WMH prevalence. While this ensures stability, it assumes a single canonical pattern of WMH pathophysiology across individuals and does not allow for subject-level heterogeneity or co-pathology. This design choice may also explain why the posterior and anterior clusters exhibit very similar pathophysiological signatures, differing mainly in magnitude and location rather than in distinct biological profiles.
6. For both UK Biobank and ADNI, neurodegenerative dementia diagnoses appear to be based on clinical criteria rather than biomarker-confirmed status (e.g., amyloid or tau PET, CSF). This distinction is important because clinical diagnosis alone can misclassify cases, especially in mixed or atypical presentations, and may dilute disease-specific WMH signatures. The potential for misclassification is particularly relevant when interpreting the posterior WMH pattern as Alzheimer's-related. Please clarify.
7. The reported correlation between the posterior WMH cluster's cortical connectivity profile and tau PET distribution is intriguing, but its biological specificity warrants caution. Occipital and posterior association cortices—regions contributing the largest number of fibres to posterior white matter—are also heavily connected hubs in the brain. If these regions naturally exhibit high streamline density and are simultaneously among the earliest or most prominent sites of tau deposition in Alzheimer's disease, the observed association could reflect network topology rather than a mechanistic link between WMHs and tau pathology. Please clarify: how did the analysis account for baseline connectivity density or hubness? Could the correlation be driven by the fact that posterior WMH clusters inherently connect to high-degree nodes (e.g., visual association cortices)? Would the association remain significant after normalising for streamline count or using a null model that preserves degree distribution (beyond the spin test)? Consider tempering the interpretation: the result may indicate spatial co-localization of two processes in highly connected regions, rather than direct evidence of Wallerian degeneration from tau-positive cortex. Including these caveats or performing a sensitivity analysis would strengthen the claim and avoid overinterpretation.

Discussion:

1. The statement on lines 415-417 claims that the data-driven approach "revealed that deep-anterior WMHs had more severe pathophysiology than periventricular-posterior WMHs, which does not seem to be caused by spreading patterns." While this is technically accurate in terms of severity, the interpretation appears to overstate novelty. Based on the presented results: a) the most salient axis of heterogeneity still aligns with the traditional periventricular vs deep WMH distinction, as the periventricular cluster clearly differs from the other two in both magnitude and pattern, b) the additional anterior/posterior split adds spatial nuance but does not fundamentally overturn the established classification; rather, it refines it, c) furthermore, the posterior and anterior clusters share highly similar pathophysiological signatures and nearly identical SuStaln trajectories, suggesting that the main biological contrast remains between periventricular and deep WMHs. We recommend tempering the claim and clarifying that the findings largely confirm the periventricular vs deep dichotomy, while providing additional granularity within the deep WMH category. A brief discussion of how this refinement compares to prior heuristic classifications would strengthen the interpretation.
2. Lines 438-442: while the authors interpret the correlation between posterior WMHs and tau-prone cortical regions as evidence of a mechanistic link, this association could equally reflect the inherent topology of white matter pathways (see previous comments). Without tract-specific degeneration evidence or longitudinal data, the claim of a causal chain from tau accumulation to fibre loss and WMH formation remains speculative.
3. Although the authors argue that spatiotemporal pathophysiological signatures can disentangle vascular and neurodegenerative WMHs, they also acknowledge that anterior WMHs are associated with both small vessel disease and neurodegeneration (lines 450-454), which "cannot purely be distinguished by neuroanatomical location." This admission appears to contrast with their spatial clustering approach, as their primary evidence for differentiation relies heavily on location (posterior vs anterior) combined with microstructural profiles that are largely similar in pattern and differ mainly in magnitude.
4. The discussion section ventures into clinical implications (lines 455-464; e.g., antihypertensive treatment decisions and eligibility for amyloid-clearing drugs) that are not directly supported by the presented data. Given the cross-sectional design and the modest, partially overlapping pathophysiological patterns, these statements risk overstating the immediate clinical applicability of the findings. While we concur that a more nuanced approach may have potential clinical relevance, it should be framed more cautiously as a future perspective rather than as a practical implication.

Minor comments:

- Voxel inclusion thresholds (Methods, Spatial clustering section, lines 710, 724-725): please justify the fixed cut-offs used for voxel inclusions (WMH prevalence < 30, NAWM < 5,000). In a sample of 32,526, these map to 0.09% and 15.4% of participants, respectively, which appears arbitrary. We recommend adopting either (a) percentage-based thresholds (e.g., WMH \geq 0.5%, NAWM \geq 15%) or, preferably, (b) precision/estimate-stability based thresholds (e.g. thresholds that bind the SEM of voxelwise mean z-scores).
- We suggest rephrasing lines 323-324 for clarity. The current sentence reads "Control subjects had no endocrine,

circulatory, behavioural, nervous, or metabolic-disease related lifetime diagnoses, with some exceptions (n = 10,629).” The phrase “with some exceptions” is ambiguous and does not align well with the explicit sample size. Instead, we recommend specifying the rationale for these exceptions, for example: “Control subjects had no endocrine, circulatory, behavioural, nervous, or metabolic disease-related lifetime diagnoses, except for common conditions considered unlikely to impact WMH burden (e.g., migraine, depressive episode).” Additionally, please note that the assumption that migraine has no influence on WMH burden is debatable and should be acknowledged or justified with supporting evidence.

- The analysis reports standardised betas for volume differences by sex (age adjusted), and for pathophysiology (additionally volume adjusted). These are small to modest effects but consistent. Consider adding absolute volume differences (mL) alongside betas for interpretability.

Summing up, the study demonstrates spatial heterogeneity (posterior vs anterior predominance linked to AD vs vascular risk), but the underlying microstructural profiles differ mainly in magnitude rather than composition, and temporal divergence is not supported (SuStaln found no sub-trajectories, with nearly identical progression curves). Disease-specific contrasts and connectivity analyses are suggestive but remain correlational and limited by cross-sectional design, diagnostic granularity, and metric specificity. Overall, the work advances spatial characterization and provides preliminary etiologic insights, yet falls short of delivering robust, diverging spatiotemporal signatures as originally hypothesised.

(Remarks on code availability)

Reviewer #4

(Remarks to the Author)

The manuscript by Parent et al. called ‘Characterizing spatiotemporal white matter hyperintensity pathophysiology in vivo to disentangle vascular and neurodegenerative contributions’ describes a study in which based on multi-modal brain MRI the authors have identified different spatial patterns of white matter hyperintensity pathology using a data driven and voxel-wise approach on a large population-based dataset (UKBB). To assess the tissue properties of white matter hyperintensity pathology, the following microstructural markers were used: MD, FA, ISOVF, ICFV, OD, T2* and QSM. The extensive analyses performed in this study broadly consist of the following steps:

1) Using voxel-based normative modelling of each of these markers based on only the NAWM voxels, resulted in age- and sex- specific maps of expected healthy microstructure. Using these maps for the normalization, individual-level pathophysiology maps are created.

2) To assess patterns of shared pathophysiological properties, spectral clustering was performed. Averaging the individual-level pathophysiology maps, only using the WMH voxels, creates seven between-subject averaged WMH pathophysiology maps, where the value of the voxel in a map represents the average deviation from healthy NAWM (expressed in SD) at that voxel location. These between subject maps are the input of the spectral clustering, which essentially leads to clustering the voxels based on the values of the 7 microstructural markers in those voxels. This resulted in the following spatial patterns: Periventricular (Cluster 1), Posterior (Cluster 2) and Anterior (Cluster 3). There was no consistent agreement of these spatial patterns and often-used WMH parcellations (periventricular vs deep, lobar regions, cerebral artery territories).

3) temporal pathological cascades per spatial cluster were determined using SuStaln (disease progression model which performs subtyping as well as staging) and provides information on the order at which changes per microstructural marker occurs within each cluster. In general the progression patterns was similar across the clusters, except for the amount of ‘events’ that take place (corresponds with the amount of stages) in each cluster.

The authors conclude that these spatial patterns show that periventricular-anterior WMHs have more severe pathophysiology than periventricular-posterior WMHs. Temporal progression pattern of WMHs were similar across the different clusters and is in general in agreement with the hypothesis that water infiltration and inflammation are the early events in the progression pattern of WMH. Based on the dominance of WMHs in the posterior cluster in patients with dementia/AD and differences in OD, they conclude that the WMHs in individuals with dementia/AD are neurodegeneration-related.

This paper is an important contribution to the field, where the advanced data-driven approach provides relevant new insights in the pathophysiology behind WMH. The authors did not stop when spatial clusters with similar pathophysiological signatures were identified, but also assessed temporal progression of the microstructural markers and evaluated in which WMH clusters the WMH volume occurred in vascular and neurodegenerative disorders and which microstructural markers deviated most compared to controls. Putting all this information together the authors suggest that WMH in patients with AD/dementia is the result of selective degeneration of fibers.

I believe it is an excellent paper and it was a pleasure to read it. I only have a few minor concerns/questions for the authors:

- Regarding the exclusion of voxels for the between-subject averaged pathophysiology maps, used as input for the spectral clustering, how are the cut off values of 30 (WMH) and 5000 (NAWM) chosen? A prevalence of only 30 individuals with WMH at a certain voxel in the large UKBB dataset seems still very low to me.

- Did the authors explore the relationship between stage and WMH volume? Since there is not a very strong relationship between WMH volume and Z-score values of the 7 markers within the periventricular cluster (as shown in Extended Figure 4), perhaps comparing the staging with the volumes provides a clearer picture of the meaning of WMH volume in this location.

- In the results section where spatial clustering is described, I'd suggest to change the sentence: “Subject-level maps of WMH pathophysiology were first collapsed into an average for each of the seven markers,” to something like:

“Between-subject averaged WMH pathophysiology maps were retrieved by averaging all subject-level maps, resulting in one between-subject map per marker”. I found the description in the methods much clearer and the term ‘collapsed’ and then only stating ‘average of each of the seven markers’ in the results section are confusing in my opinion.

- I struggle a bit with the interpretation or the way the authors describe the level of pathophysiology, especially in the results

section of the temporal pathophysiological cascades. For example It is stated that: 'Periventricular WMHs showed low pathophysiology across temporal stages'. I am not sure whether 'low pathophysiology' makes sense here, I don't understand what the authors meant by this. The pathological events are defined as Z-scores, or in this case steps of 0.5, 1, 1.5 SD. Using SuStaln you could also choose to label 0.25 SD as an event, so the number of events does not perse reflect severity or level of pathology, in case that is wat the authors meant with low pathophysiology. Furthermore, Although in this region it only goes until 1.5 SD, it could in terms of 'pathology' perhaps be as severe as reaching 5SD in another region? Later on in the paragraph the authors see the Z-score as level of abnormality, however assigning the biological meaning of a certain deviation from the mean is not something SuStaln does, as user of the algorithm you have to choose what you define as an event. To me it makes sense here to use the Z-scores as the authors did (with the addition of 0.5 and 1.5 SD), but in my opinion the authors should be careful with the interpretation and formulation.

- Did the authors assess whether the staging within each of the spatial clusters correlated with WMH volume in that region? You'd expect that individuals with low WMH in a spatial cluster would be staged early in the temporal cascade. It would also be interesting to see whether individuals with no visible WMH yet are always ending up in stage=0, or that they already show signs of deviations in some of the markers. In that case the staging could be an interesting predictive marker of WMH in the future potentially?

(Remarks on code availability)

Version 1:

Reviewer comments:

Reviewer #1

(Remarks to the Author)

The authors have adequately addressed all comments.

(Remarks on code availability)

Reviewer #2

(Remarks to the Author)

Apologies if I am stating the obvious, but part of the revision process is to improve the quality of the paper through feedback from other researchers and by thoughtfully accepting or rebutting the points raised during review. The keyword here is "improve". While some aspects are indeed clearer in this second version of the manuscript, some of the authors' responses honestly diminish the significance of the work.

Reviewer #1 (Comment 3), Reviewer #2 (Comment 8), and Reviewer #3 (Comment 4 and 10): Since clusters 2 and 3 differ primarily in overall severity, an important question is whether they actually represent distinct aetiologies or underlying pathophysiological mechanisms. The response provided—namely, that aetiological differences cannot be inferred solely from this finding and that further characterisation of specific pathophysiological alterations is required—raises two concerns. 1. Given that the stated aim of the manuscript was to "disentangle vascular and neurodegenerative contributions" and to "parse the pathophysiological, spatial, temporal, and ultimately aetiological heterogeneity of WMHs", I find it difficult to conclude that the current version fully meets these objectives. The authors appear to address this discrepancy primarily by noting in the limitations that the study's aims were not entirely achieved. While this is an important acknowledgement, it also suggests that key elements of the intended contribution remain underdeveloped, leaving the work feeling incomplete in its present form. This is not merely a matter of wording choices, but rather a concern that relates to the conceptualisation, study design, and interpretation of the findings. The inability to disentangle aetiology also implies that the discussion requires reconsideration. For example, the authors state that "While our findings will require further validation before being integrated into clinical settings, using in vivo MRI-based pathophysiological markers to disentangle neurodegenerative and vascular contributions to WMHs could have broad implications for clinical management and for the interpretation of research studies and clinical trials". This may be true in principle, but it does not sound fitting given the earlier acknowledgement that such disentanglement was not actually achieved in the study.

2. If the authors—who have access to what is likely the largest multimodal dataset for analysing WMH patterns—are unable to characterise these clusters deeply, disentangle vascular from neurodegenerative contributions, or describe (not speculate) aetiologies, then it is difficult to see who would be able to do so. The literature is already full of papers offering speculative interpretations of WMH and their patterns, so producing yet another one—even if it uses a larger sample—would not be sufficient. The authors need to move beyond speculation and clearly articulate the concrete findings that their analyses have revealed. What do these massive datasets allow us to learn that has not already been demonstrated in smaller studies, many of which have reported similar patterns? In other words, what is genuinely new here—clinically, medically, or neuroscientifically?

Reviewer # 3 (Comment 8): Reviewer 3 argues that the anterior-posterior split adds nuance but does not fundamentally alter the established classification; it merely refines it. The authors' response appears to concur with this viewpoint, which, from my perspective, implies that undertaking such a large and complex study may not have been necessary if existing classifications were already adequate. This can happen in scientific research, of course, but in my view simply agreeing with this point substantially diminishes the impact of the work. A closer inspection of the parcellation shows that the anterior–

posterior wording is not entirely accurate. Several juxtacortical regions appear to align with cluster 2 (posterior), and, conversely, describing cluster 3 as “deep-anterior” is not particularly precise, as most white-matter regions—aside from those in the occipital lobe—are assigned to this cluster. Even if one assumes that the main contribution of the work is to add nuance to the anterior–posterior distinction (which, in my view, is not the case), the subsequent question is whether this added nuance actually holds any benefits. To address this, the authors need to compare the performance of their parcellation with the established anterior–posterior split, as well as with the commonly used periventricular versus deep white-matter distinction, across the relevant tasks (those already conducted in their work, for instance).

One of the points I was not able to articulate clearly during the first round of revisions concerns the region-growing strategy used by the authors to incorporate voxels with greater sparsity (that is, voxels observed in relatively few participants). I am uncertain about the implications of this step for the clustering results. Specifically, I wonder whether the inclusion of very rare voxels may be diluting—or even obscuring—the genuine differences between clusters, thereby making the separation of clusters appear less biologically meaningful than it might otherwise be. Given this possibility, it would be valuable to assess the sensitivity of the findings to the region-growing approach. Presenting the clustering results both with and without region growth would allow readers to evaluate whether this methodological choice meaningfully alters the spatial patterns, the cluster characteristics, or the downstream associations.

(Remarks on code availability)

I commend the authors for sharing both the patterns and the accompanying code. This is valuable, and I can imagine such an analysis being useful to replicate in the future. However, the current version of the code does not appear to be designed or documented to the extent that an external user could readily run it. For instance, the README describes the data included in the repository but does not explain how to execute the code. To make the code genuinely usable, additional documentation is needed. There is Python code in the repository, but it is unclear which environment or software versions are required to run the application.

Reviewer #3

(Remarks to the Author)

I thank the authors for their thorough work on the manuscript and I would like to congratulate them for successfully addressing most of the comments from me and the other reviewers. I believe the manuscript has strongly improved in clarity and robustness of the methods. Overall, this is an important work with very extensive analysis. There are few points that I would still like the author to consider for their next formulation of the paper.

Estimates of WMH Pathophysiology: This whole paragraph has been greatly improved and it reads very clearly. One question regarding this is how the authors handled voxels that have more commonly WMH, and therefore less “points” for the normative modelling. For example, some periventricular voxels will often have lesions in individual >60 and this could bias the normative modelling estimation, and thus the WMH pathophysiology estimation. I understand that these were excluded from the clustering analysis (correct?) and then re-inserted as being part of the closest cluster. However, biased estimates would remain. The authors should address this limitation and maybe give a supplementary figure showing the number of used points (individuals with usable data) for the normative model for each voxel.

Spatial regions of pathophysiological similar WMHs: I have three comments regarding this part.

- 1) I still find it reductive that the authors, after having computed all this very informative WMH pathophysiological maps, decide to go for group averages to find clusters. While I understand why and how this is done, looking for patterns at the individual level (and whether the observed clusters are actually reproducible at the subject level, or their expression changes based on some variables) would still have a very strong impact I believe. The authors should note this limitation, and I hope that they will consider this in future studies.
- 2) Most importantly: The authors find 3 clusters and characterize them on several indices and spatial similarity. Have the authors considered that the observed cluster could also represent underlying fiber-types and their specific microstructural characteristics? For example, cluster 2 could represent lesions in commissural fibers, while cluster three in projections or associative fibers. This might help and change the interpretation of the observed alterations, also in relationship to the temporal sequence of events.
- 3) It is unclear how were zero values used here, participants with very little lesions will have sparse maps, with zeros in most voxels. Was this information considered in the group averages? Was there any weighting for nr. of voxels?

Defining Temporal Pathophysiological cascades: In line with my previous comment, some participants will have all zeros in the 8 metrics for some of the clusters (or even for all of them). It is unclear how this sparsity was considered for the sustain analysis.

(Remarks on code availability)

Reviewer #4

(Remarks to the Author)

I have carefully reviewed the authors’ rebuttal and the revised manuscript. I am fully satisfied with the responses provided to my previous comments. The authors have addressed all concerns in a thorough and convincing manner. I have no further comments or requests.

(Remarks on code availability)

Version 2:

Reviewer comments:

Reviewer #2

(Remarks to the Author)

This revised version highlights the importance of the work much more clearly, and I am happy with it. I again thank the authors for their substantial efforts in addressing my concerns, their patience throughout the review process, and for this excellent contribution.

(Remarks on code availability)

Reviewer #3

(Remarks to the Author)

I thank the authors for their work, I have no further comment.

(Remarks on code availability)

Summary of the main changes

Many of the changes requested by reviewers were regarding some specific wording choices. To address these issues raised, we added caveats to the biological interpretation of MRI microstructural markers, clarified the specific novelty of our spatial clustering approach within the context of the existing literature, indicated that our results were observational and remain to be causally established, changed “healthy” to “normal-appearing” white matter, clarified our interpretation of the clustering solution, and softened our wording comparing our novel parcellation to the PV/deep parcellation scheme.

Some changes to existing analyses were made, which did not alter our previous interpretations. Mainly, for the UKB diagnosis analysis (Fig 4A), we excluded participants with comorbidities among the diseases analyzed (e.g., people with a stroke or a ischemic heart disease diagnosis were excluded from the dementia group). This reduced the group sample sizes but all interpreted significant effects remained.

Additional analyses were performed to investigate if our chosen WMH and NAWM prevalence threshold were appropriate (Supp. Fig. 2), see which microstructural measures were driving the clustering solution (Supp. Fig. 4), compare our results to a clustering solution done without excluding any voxels based on low WMH prevalence (Supp Fig. 7B), investigate different PRS cutoffs (Supp. Fig. 13), analyze sex-by-group interactions for disease effects (Supp. Fig. 14), and derive the WMH pathophysiological signature of ADNI amyloid positive MCI and AD participants (Supp. Fig. 15).

We further deepened our discussion regarding the uncovered spatial clustering, SuStaIn limitations which could explain why no temporal WMH pathophysiology subtrajectories were detected, how non-vascular WMHs could result in FLAIR hyperintensities, how our results can be used by different groups, and we now interpret the observed sex differences. Responses to individual comments are detailed below.

Reviewer 1

Summary: Parent and colleagues assessed multiple MRI parameters within areas of WMH in a large sample of participants from the UK Biobank ($n > 300,000$) to test whether WMH cluster into spatially and microstructurally heterogeneous subtypes. The rationale was that combining multiple quantitative MRI measures (diffusion and susceptibility measures) may reveal the underlying biological substrates of WMH and thus detect different subtypes, such as those more characteristic of vascular vs. neurodegenerative etiology. The authors employed spectral clustering to derive WMH clusters and plotted the z-values of abnormality for each MRI measure. Pseudotemporal evolution was estimated using event-based modeling. MRI measures were subsequently compared between control subjects and different diagnostic groups in the UK Biobank, as well as across levels of PRS scores for cardiovascular/stroke

and AD. Furthermore, MR measures were compared between AD and control groups in an external dataset from the ADNI study. Spatial associations between WMH areas of each cluster and typical tau deposition were assessed based on group-level connectivity analyses. In addition, sex differences were investigated. Results showed three clusters of WMH: periventricular, posterior periventricular, and anterior-deep white matter types. Each cluster was characterized by distinct z-score profiles across MRI parameters, suggesting microstructural alterations specific to different WMH types. Event-based modeling with SuStaIn indicated a similar temporal ordering of the development of abnormalities among the different MR measures, with mean diffusivity and intracellular volume fraction (ICVF) appearing first. Comparisons of MR parameters across groups for each MRI measure revealed disease-dependent differences. Overall, this is a comprehensive study involving multi-parameter characterization of microstructural alterations in WMH. The approach is innovative as a step forward from simply mapping the spatial locations to different diseases, given that the underlying microstructural changes can be heterogeneous as revealed in prior post-mortem studies. Yet the conclusiveness of the findings remains somewhat unclear.

Comment 1: A concern is that the MRI measures assessed in the current study cannot be unambiguously linked to the pathophysiological events underlying WMH, and the chosen approach may not resolve these uncertainties. In the introduction, the authors state that “fluid infiltration from ventricles and blood vessels triggers inflammation, leading to tissue damage such as demyelination, oligodendrocyte loss, and, in more severe cases, axonal degeneration,” which may lead to WMH. They also suggest that multiparametric MRI provides homologs of post-mortem brain stains and can therefore reveal WMH pathophysiology. However, MR measures such as FA, MD, OD, ICVF, ISOVF, T2*, and QSM are ambiguous in terms of the underlying substrate when considered individually. It remains unclear how combining these measures resolves such ambiguities, especially when nearly all indices show alterations, as observed in this study. For example, free water (edema), fiber loss, macromolecules (inflammation), or myelin loss may all affect MD, FA, or OD. The authors are advised to critically discuss these issues and moderate their conclusions.

Response: *We thank the reviewer for their time and thorough and comprehensive review of our paper. We acknowledge and agree that microstructural MRI lacks biological specificity. While this fact was pointed out in some places throughout the manuscript, we have added the following text to further highlight and emphasize this point in the Introduction section (p. 4, lines 112-115):*

“Multi-contrast microstructural MRI, while lacking the biological specificity of these techniques, has established sensitivities to various tissue properties,^{22,23} providing a window into biophysical tissue alterations in the brain and allowing for the non-invasive, large-scale investigation of WMH pathophysiology.”

as well as in the Results section (p. 6, lines 153-162):

“For example, diffusion tensor imaging (DTI) indices such as FA and MD are particularly difficult to interpret since they assume a main axis of diffusion within voxels, while most voxels instead contain multiple fiber populations with different orientations.²⁷ Different multi-compartment modeling schemes of the diffusion signal have been developed to handle the crossing-fiber problem and increase the interpretability of microstructural metrics, one of which is the Neurite Orientation Dispersion and Density Imaging (NODDI) model used here.²⁸ While no metric perfectly captures individual biological processes (e.g., inflammation, demyelination, edema), interpreting these markers together can help generate hypotheses about the potential biological substrates of the observed effects.²³”

and in the Discussion section (p. 23, lines 481-483):

“Of note, these biological interpretations remain speculative and will need to be confirmed with gold standard ex vivo methods such as histology.”

However, we argue that combining these metrics can give us hints regarding the underlying tissue alterations. For example, the lack of QSM abnormality combined with positive T2 abnormality points to myelin AND iron loss, as is discussed in the Discussion section (p. 22-23, lines 476-481):*

“The combination of positive T2* abnormality, consistent with our previous work,⁴⁹ and very low QSM positive abnormality potentially indicates myelin and iron loss, as these two processes would have an additive positive impact on T2* but would cancel each other out on QSM.^{34,35} While iron accumulation occurs in aging, especially in subcortical grey matter regions,⁵⁰ iron loss in white matter could be caused by the death of oligodendrocytes, which contain large amounts of paramagnetic iron.⁵¹”

As a follow-up study, we aim to quantify the biological sensitivities of these 7 markers specifically in WMHs using a rich dataset combining MRI and histology, which will ultimately bridge the gap between gold-standard ex vivo histology and in vivo MRI markers used here. However, we believe this is well outside the scope of the current study.

Comment 2: Visual inspection of cluster 1 suggests that it covers a small rim around the ventricles, an area prone to partial volume effects (PVE), which may result from inadequate segmentation and spatial normalization. Additionally, for this cluster, z-scores for most MR measures are close to zero, making interpretation difficult. Did the authors exclude voxels directly adjacent to ventricles to minimize PVE or took other measures to reduce PVE?

Response: *We did not exclude voxels around the ventricles, and we agree with the reviewer that PVE effects likely drive the observed effects of minimal microstructural abnormality. However, we argue that our spatial clustering step essentially removes these noisy voxels, which include PVE-contaminated voxels and very small WMH rims which are very likely benign, as now discussed in the Results section (p. 10, lines 218-219):*

“These voxels may be contaminated by partial volume effects (i.e., ventricle signal) and include small caps WMHs that are likely benign.³⁹”

Comment 3: Clusters two and three show the same pattern of alterations in MRI measures but differ quantitatively in z-scores and spatial location. Do the authors conclude that these two signatures reflect similar etiology or underlying pathophysiology?

Response: *We agree with the reviewer that the pathophysiological patterns mostly differ in overall severity. We don't believe etiological differences can be determined purely from this finding. We added this clarification in the Results section (p. 11, lines 230-233):*

“The posterior and anterior clusters mostly differed in the severity of pathophysiological markers, but still revealed distinct spatial signatures. Further characterization of specific pathophysiological alterations is needed to assess potential etiological signatures.”,

and in the Discussion section (p. 24, lines 523-530):

“This is supported by our finding that WMH volume was significantly elevated relative to controls across all three spatial regions in both cerebrovascular disorders and dementia, highlighting the need for a deeper phenotyping of WMH pathophysiology to parse etiologies. We still encourage other groups to use our publicly available parcellation scheme (https://github.com/CoBrALab/WMH_patho_UKB), but note that this is informative but not sufficient to disentangle vascular from neurodegenerative contributions. Based on our observations, metrics derived from multi-shell diffusion MRI acquisitions in particular show promise for identifying WMH pathogenesis.”

Comment 4: The analysis of group differences and associations with PRS scores remains unclear.

Comment 4a: Conceptually, what was the rationale for comparing groups across each MRI measure separately after identifying cluster-specific MRI signatures? Why not test whether patients with a particular disease are more strongly associated with specific WMH clusters? Such an analysis could clarify whether WMH clusters and their microstructural alterations are disease-specific.

Response: *From our understanding, the analysis proposed by the reviewer is already included in our diagnosis analyses in Figure 4. We used linear models to relate diseases to WMH measures within each cluster, as is proposed (e.g., WMH FA in cluster 1 ~ Group (DX vs HC) + Age + Sex). We apologize for this ambiguity. To clarify, the clustering analysis leveraged microstructural MRI measures to derive a parcellation at the group level, but information regarding the pathophysiological alterations present for each individual subject is not represented in the clusters. We hope that our analysis scheme is clearer following our answer to Comment 4b.*

Comment 4b: Technically, it is unclear how statistical analyses and corresponding FDR corrections were performed. For example, for ischemic heart disease the authors report: “WMH alterations were small but significant, with stronger effects in anterior regions (average beta = 0.19; p (FDR) < 0.05 for all markers) than posterior regions (average beta = 0.13; p (FDR) < 0.05 for all markers except OD and T2*).” Does that mean a voxel-based analysis was performed for each MRI marker, with beta coefficients subsequently averaged across all markers within each cluster mask? What exactly does the FDR correction refer to?

Response: *All analyses performed after the section “Spatial regions of pathophysiologically similar WMHs” are ROI-based, and not voxel-based. To do so, we sampled the median microstructural value across all voxels within the WMH mask within each spatial cluster mask for each subject. This is now clarified in the Results section (p.13, lines 280-282):*

“This reduced the complex voxel information into a set of 24 values (8 measures * 3 regions) for each subject. All further analyses are made at the ROI level and not at the voxel level.”.

*FDR correction is performed across all effects calculated in one analysis scheme. For example, Figure 4A shows 81 effects (9 measures * 3 regions * 3 diagnoses) and FDR correction is calculated across these 81 p-values. Now clarified in the Methods section (p. 30, lines 651-652):*

“For example, in Figure 4A, FDR correction is applied on 81 p-values (9 measures * 3 regions * 3 diagnoses).”.

Comment 4c: For controls, several diseases were defined as exclusion criteria. Were the same criteria applied to disease cases (apart from the disease of interest)?

Response: *The reviewer raises an important point and limitation of our current work, and more broadly speaks to the difficulty in dealing with complex comorbidity profiles in population-based cohort studies like UK Biobank. We could not use the strict exclusion criteria that we used to design the control group (excluding 247 unique ICD-10 diagnoses), as very few “clean” participants would be left. However, we have now rerun the diagnosis analysis, excluding individuals who had comorbidities among the diagnoses analyzed (e.g., for dementia, we now excluded individuals with a diagnosis of stroke or ischemic heart disease). Results are now updated in the Results section “Pathophysiology of WMHs across vascular and neurodegenerative disorders” (p. 19-20), in Figure 4A, in Supplementary Figure 8, and in the Methods section (p. 33, lines 727-729):*

“We excluded subjects with comorbidities on the studied diagnoses (e.g., for dementia, we excluded individuals with a diagnosis of stroke or ischemic heart disease).”.

No substantial changes in our interpretation of the results were necessary following this update.

Comment 5: Related to point 4: Figure 4: What determined the location of the asterisks?

Response: *The location of the asterisks is determined by which WMH marker and cluster showed a significant effect after FDR correction.*

Comment 6: The authors highlight OD differences in MCI/AD, yet alterations in other indices were observed as well. Furthermore, OD differences were seen in MCI but not in AD dementia. How do the authors interpret these findings?

Response: *While the posterior OD effect reaches significance in MCI and not AD in the ADNI analysis, the effect sizes are moderate and of similar amplitude (MCI $\beta = 0.37$; AD $\beta = 0.38$). It is likely that the AD effect does not reach significance because of the small sample size ($n=21$); however we still believe that the almost identical effect sizes between MCI and AD still demonstrates a consistent effect, which was also observed in the UKB diagnosis and PRS analyses. This is now clarified in the Results section (p. 19, lines 396-399):*

“For example, the posterior WMH OD effect size was almost identical in the MCI group ($\beta = 0.37$; $p = 0.01$) and dementia group ($\beta = 0.38$; $p = 0.08$), hence the difference in significance likely stems from differences in sample sizes.”

Comment 7: Why dichotomize PRS when it is inherently a continuous measure?

Dichotomization introduces arbitrary cut-offs, reduces statistical power, and, as in the present study, results in exclusion of a substantial amount of data.

Response: *Dichotomization of PRS scores is an often-used method to identify high-risk individuals (eg. Lennon et al., 2024, Nature Medicine, <https://doi.org/10.1038/s41591-024-02796-z>). We hypothesize that this is likely because PRS scores are noisy in the middle of the distribution, and may show complex non-linear associations with disease outcomes. Attempting to model these more complex associations would complicate the interpretation of findings and limit the quantitative comparison between effect size patterns we performed in Figure 4D. Instead, we have now replicated our findings using different PRS cutoffs (top 5% and top 3%), now available in Supplementary Figure 9 and in the Results section (p. 18, lines 385-386):*

“This analysis was replicated using different cut-off values to determine high genetic risk individuals (top 3% and top 5%; Supplementary Figure 13).”

Comment 8: Why were no covariates controlled for when comparing AD vs. controls in the ADNI sample? Were the MCI subjects in ADNI amyloid-positive?

Response: *As in the UKB diagnosis analysis, covariates of age and sex were included in the linear models (see Methods p. 34, lines 748-749: “Groups were compared with linear models controlling for age and sex (Fig. 4C)”). We clarified the number of amyloid positive*

individuals in each group in the main text, although it is important to note that not all included subjects had amyloid PET data available. Results section (p. 18, lines 390-392):

“Among individuals with amyloid positron emission tomography (PET) data, 28% of cognitively normal individuals, 53% of people with mild cognitive impairment, and 94% of people with dementia were amyloid-positive (demographic statistics detailed in Supplementary Table 2).”

Lastly, we added a sensitivity analysis where we compared controls to amyloid-positive MCI and AD subjects in ADNI, as discussed in the Results section (p. 19, lines 406-411):

“We additionally compared amyloid-positive MCI and dementia subjects (n=42) to amyloid-negative cognitively normal subjects from ADNI (Supplementary Figure 15A) in order to isolate biomarker-confirmed cases. Effects were similar to the other ADNI patterns and showed higher correlations with effect size patterns in the neurodegenerative grouping (average $r = 0.63$) compared to the vascular grouping (average $r = 0.32$; Supplementary Figure 15B).”

Comment 9: Overall, the number of AD/dementia cases is small, and in the UK Biobank includes mixed dementia types. The results should be interpreted with caution.

Response: *We agree with the reviewer that this is an important limitation of the UK Biobank diagnosis analysis, as we mention in the Results section (p. 17, lines 361-362):*

“However, this analysis is limited by the uncertainty in dementia type (n=23 were diagnosed with unspecified dementia).”

This is what prompted the replication of the signatures using genetic risk and the ADNI data. The replicability of the UK Biobank dementia signature is exemplified with a very high correlation of the effect size patterns with the ADNI dementia pattern ($r=0.81$; Figure 4D), which we now added in the Results section (p. 19, lines 405-406):

“For example, the UK Biobank dementia signature was highly correlated with the ADNI dementia signature ($r=0.81$).”

Comment 10: How do the authors interpret the observed sex effects? Were sex differences in MRI measures evident across diagnostic groups?

Response: *We have now added a paragraph in the Discussion interpreting the observed sex differences (p. 25, lines 531-547):*

“Investigating sex differences, we observed that females had lower WMH volumes but higher pathophysiological effects in the posterior region, and higher WMH volume but similar pathophysiological effects in the anterior region relative to males. This novel finding adds

nuance to the reported higher WMH load in women,⁶² further highlighting the relevance of differentiating posterior and anterior WMHs. While this pattern could be due to the menopausal transition and the reduction in estrogen's neuroprotective effects,⁴³ a recent investigation from our group found no differences in WMH volumes in women before and after the menopausal transition when accounting for the effect of age in UK Biobank,⁶³ although examining region-specific effects using our novel parcellation would be an interesting follow-up. The observed higher WMH volumes in females in the more vascular-related anterior region has been previously reported by other groups.⁶⁴ While this is counterintuitive, given that cerebrovascular diseases are more prevalent in males, this could be due to the higher frequency of arterial stiffness in females.⁶⁵ The pattern of higher WMH volumes in males but worse pathophysiology in females in the more neurodegeneration-related posterior region is intriguing, and may be related to female-specific AD dynamics. For example, inflammatory microglial activation has been reported to play a more important role in the AD pathological cascade in females versus males, which could also impact WMH pathophysiology.⁶⁶

We also investigated sex by group interactions and found no significant effects (p. 19, lines 400-402):

“We investigated sex by group interactions to investigate if some effects were higher in males or females, but no effect was significant at the p (FDR) < 0.05 level (Supplementary Figure 14).”

Reviewer 2

Summary: I thank the authors for the considerable effort invested in this manuscript. From the very first read, it was clear to me that this work is the product of years of dedicated research. I am thus sincerely grateful to both the authors and the editors for the opportunity and privilege to review this wonderful piece of work. To match the authors' dedication, I have sought to provide feedback that was both thorough and (hopefully) constructively critical.

Comment 1: Line of argumentation. The authors argue that attributing white matter hyperintensities (WMH) to a vascular origin, as the expression “of presumed vascular origin” does, neglects their heterogeneous origins. While I partly agree with the authors' line of argumentation, I would also contend that the qualifier itself is not the core problem. The expression “of presumed vascular origin” does not categorically assert a vascular cause for all WMH, but rather implies that some—though not necessarily all—may arise from vascular mechanisms. The authors of the STRIVE criteria themselves acknowledged that, although the vascular hypothesis provided a plausible explanation for the origin of WMH, WMH were possibly multifactorial and emphasised that the qualifier “of presumed vascular origin” was a way to distinguish CSVD-related WMH from similar lesions in disorders such as multiple sclerosis and leukodystrophies (see page 826 of Wardlaw et al. (2013) for exact phrasing).

Instead, I would argue that the core limitation of STRIVE's definition lies in its broad characterisation of WMH of presumed vascular origin as T2-weighted MRI hyperintense white matter regions, without providing additional criteria to reliably differentiate vascular from non-vascular aetiologies. The leap from "WMH of presumed vascular origin" to assuming that all WMH fall under this vascular category is another issue entirely. My main recommendation for the manuscript is to move from this initial line of argumentation and place greater emphasis on the subtyping of WMH per se. Sure, WMH are multifactorial, but how can we meaningfully differentiate between their underlying causes in vivo and develop robust methods to quantify them accordingly? This is exactly what the authors aimed to accomplish: to define WMH subtypes using multimodal data, assess their clinical relevance, and try to determine how the underlying processes unfold—and whether these differ across the identified subtypes.

Response: We thank the reviewer for their kind words, their recognition of our scientific effort, and their dedication to a detailed review that will undoubtedly improve the manuscript. We acknowledge that the wording used may have unjustly criticized the STRIVE criteria, which was not our intent, and have thus removed references to the wording "of presumed vascular origins" in the introduction, as this was also pointed out by Reviewer 3.

Comment 2: Acknowledging previous work. The second paragraph of the introduction provides context by highlighting recent findings suggesting that the observed heterogeneity may, to some extent, be connected to the anatomical location of WMH within the brain. There might be a limit on the number of references, but several claims in this section would really benefit from a more comprehensive acknowledgment of prior work and accurate phrasing. To the best of my knowledge, the earliest work to distinguish WMH based on their anatomical location is the same study that introduced what is now known as the Fazekas scale (Fazekas et al., 1987). Although based on a small sample, this visual rating offered some of the earliest neuroimaging evidence suggesting that anatomical location relates to the heterogeneity of WMH. However, as later discussed by Kim et al. (Kim et al., 2008), this distinction requires "complementary rules" to effectively classify periventricular and deep WMH, as both the terminology, definitions, and implementations used were inconsistently—the limitation the authors are also highlighting. While I agree with this point, I would like to point out that this is not the only approach to study WMH; other approaches include, for instance, the increasingly popular assessment of the so-called multispot WMH pattern for CAA diagnosis (Charidimou et al., 2022; Charidimou & Baron, 2025) as well as lobe-wise and voxel-wise analyses (see (Botz et al., 2023) for a systematic revision of the literature up to 2023). To ensure accuracy, I kindly encourage the authors to broaden their current literature review to acknowledge prior work in the field.

Response: We have now updated the paragraph addressing prior literature in WMH spatial analysis in the Introduction (p. 4, lines 97-104):

"Consequently, periventricular and deep WMHs are often assessed separately using heuristic methods with criteria that can vary significantly between studies.¹⁹ Other parcellation

schemes include cortical lobes, vascular territories, or white matter tracts.²⁰ In contrast to these hypothesis-driven parcellations, data-driven methods can uncover more complex patterns of WMH spatial heterogeneity, and while such examinations have been done based on the co-occurrence of WMHs,^{12,21} these have not taken into account the underlying pathophysiological alterations. Given the sparsity of WMH prevalence at the voxel-level, such an analysis requires a very large dataset.”

Comment 3: Novelty claims. I find it difficult to understand why the authors claim that “no study has used unbiased, data-driven methods to systematically investigate spatial patterns of WMH pathophysiology”. In recent years, we have seen a couple of papers using data-driven clustering methods for examining spatial WMH patterns, e.g., (Bachmann et al., 2024; Phuah et al., 2022; Rosbergen et al., 2024). I understand that spatial patterns and “pathophysiology” are not synonymous—the former relates to spatial distribution, while the latter concerns microstructural integrity. Nonetheless, there is already a paper published last year that incorporates microstructural measurements into the analysis (Rosbergen et al., 2024). One can say that this one has more metrics, but I wonder whether that would be enough to claim novelty. The term “unbiased” should not be used lightly here—it represents a substantial claim that, to the best of my knowledge (and apologies if I have overlooked something), has not been adequately verified. Since the methods are trained on specific datasets acquired using particular protocols and targeting certain populations, inherent biases are inevitable. Using a large cohort for training does not, in itself, guarantee the absence of bias. One thing is to show that patterns generalise and a completely different one that they are unbiased. The use of the term “systematic” is also problematic and may work against the authors, as it requires strong experimental justification. In this context, “systematic” would imply that the authors, for instance, investigated the minimal set of sequences needed to achieve the observed subtyping (i.e., whether all sequences are essential or if some can be omitted) and assessed the sensitivity of the subtyping to each subset of images (e.g., number of groups). However, I could not find any evidence supporting this “systematic” aspect. Apologies if I have overlooked it. Overall, I believe the current claims of novelty are not sufficiently substantiated and would benefit from revision. To be clear, I am not questioning the quality of the work—it is undoubtedly a strong contribution—but rather pointing out that the way it is currently described lacks precision and does not fully reflect the actual work presented.

Response: *We addressed the following points from the reviewer's comment. First, we now more comprehensively compare our results with other data-driven WMH clustering approaches in the literature, as suggested. This was done briefly in the Introduction (p. 7), as in the previous reviewer comment's response, and now also in the Discussion (p. 22, lines 463-470):*

“The investigation by Phuah et al. (2022) used spectral clustering on WMH probability maps from FLAIR images, resulting in a 5-cluster solution and found that deep frontal WMHs were associated with hypertension and diabetes, while parietal WMHs were associated with amyloid SUVR.¹² Habes et al. (2018) used non-negative matrix factorization to uncover clusters of covarying WMH volumes and found that high blood pressure and female sex were

associated with higher frontal WMH load, while AD genetic risk was associated with higher dorsal WMH load.²¹ Our pathophysiology-based approach to parsing WMH spatial heterogeneity is thus consistent with this emerging data-driven literature.”.

We also note that the paper by Rosbergen et al. (2024) clustered subjects and not WMH regions, thus did not address the spatial heterogeneity of WMHs as we have done here. Second, we have now removed the term “unbiased” from our claims in the introduction. We specify that we only meant “unbiased” in the context of our sample (data-driven approach without any assumptions), not as a guarantee of the generalizability of our results. Third, we also removed the term “systematic”. We hope that the new wording and motivation of our spatial clustering analysis is now more precise and representative of the work done and its novelty.

Comment 4: MRI vs post-mortem. In the introduction, the authors suggest that multi-contrast microstructural MRI sequences can serve as “homologs” of post-mortem stains. I’m not entirely convinced by this claim. While microstructural measurements undoubtedly bring us closer to understanding the underlying nature of these lesions, but they are not true homologs in the strict biological or pathological sense. MRI signals are indirect, sensitive to a mix of tissue properties (e.g., water content, diffusion, susceptibility), and their interpretation is often ambiguous and context-dependent (Both table 1 and the discussion actually talk about this; see also (Figley et al., 2022) for more references and further discussion on the topic). It would be more accurate (and cautious) for authors to say that multi-contrast microstructural MRI can approximate or reflect some aspects of histopathology—rather than calling them “homologs” which implies a much stronger equivalence than is currently supported by evidence—which is not the aim of the study.

Response: This was also pointed out by Reviewer 1 (Comment 1). We have added the following statements to emphasize the lack of biological specificity of microstructural MRI in the manuscript in the Introduction section (p. 4, lines 112-115):

“Multi-contrast microstructural MRI, while lacking the biological specificity of these techniques, has established sensitivities to various tissue properties,^{22,23} providing a window into biophysical tissue alterations in the brain and allowing for the non-invasive, large-scale investigation of WMH pathophysiology. ”,

in the Results section (p. 6, lines 153-162):

“For example, diffusion tensor imaging (DTI) indices such as FA and MD are particularly difficult to interpret since they assume a main axis of diffusion within voxels, while most voxels instead contain multiple fiber populations with different orientations.²⁷ Different multi-compartment modeling schemes of the diffusion signal have been developed to handle the crossing-fiber problem and increase the interpretability of microstructural metrics, one of which is the Neurite Orientation Dispersion and Density Imaging (NODDI) model used here.²⁸ While no metric perfectly captures individual biological processes (e.g., inflammation,

demyelination, edema), interpreting these markers together can help generate hypotheses about the potential biological substrates of the observed effects.²³”,

and in the Discussion section (p. 23, lines 481-483):

“Of note, these biological interpretations remain speculative and will need to be confirmed with gold standard ex vivo methods such as histology.”.

Comment 5: Estimation of WMH pathophysiology. The authors generate age- and sex-specific normative maps and compare each individual image against them to produce personalised abnormality maps. I find this approach interesting, as it directs the clustering towards individual abnormalities rather than shared features that might be more closely linked to ageing. My main concern, however, lies in how this is operationalised: the authors appear to assume that voxels labelled as normal-appearing white matter (NAWM) represent truly healthy measurements. This is a strong assumption, considering that the distinction between normal and abnormal is highly context-dependent and often lacks a clear boundary. In segmentation algorithms, for instance, the choice of threshold has effects on the segmentation of WMH and, by extension, on NAWM. A more stringent threshold tends to result in undersegmentation of WMH, while a more lenient one leads to oversegmentation. As a consequence, the definition of NAWM can vary depending on the selected threshold. At the same time, I believe the authors would agree that the abnormality associated with WMH extends beyond the visibly affected tissue, esp. in microstructural sequences (Duering et al., 2023; Ter Telgte et al., 2018). I thus truly wonder whether the assumption is correct. Could the authors please elaborate further on the correctness of their assumptions?

Response: *The reviewer is right to point out this limitation of our work. We now removed wording related to “healthy” white matter, replacing it with “normal appearing”, and acknowledge the limitation in the Results section (p. 8, lines 174-177):*

“We note that NAWM may still be pathological to a certain extent, as has been previously demonstrated.¹⁶ However, pathological NAWM remains ill-defined, and there is currently no established framework to identify it. We thus focus on contrasting WMH from non-WMH tissue, considering that some tissue alterations are part of the NAWM variations.”.

We argue that the framework implemented here does not rely on the assumption of healthy white matter to be efficient. Given the sample size of the dataset (n=32,000), the presence of some pathological NAWM in some subjects included in the normative modeling should have a relatively small effect. This could still result in the z-score metrics being a little bit higher or lower relative to the hypothetical scenario of excluding these pathological voxels. However, the effect of age and sex would still be accurately removed. Furthermore, the most important feature of this voxel-wise normative modeling step is removing the spatial contrast induced by the anatomy, leaving spatial patterns that are caused by the presence of WMHs and not by the anatomy. This crucial aspect should not be significantly affected by the

presence of some pathological values in the normative modeling, particularly given that even when present, they would be at mild (sub-threshold) severity levels.

Comment 6: Assumptions in spectral clustering. The spectral clustering results reveal regions that align with previous findings in the literature, with anterior WMH more closely associated with vascular problems and posterior WMH more linked to neurodegenerative processes. This is a valuable contribution to the literature, as it provides large-scale validation of previously observed patterns. Interestingly, even the authors appeared somewhat surprised that, despite using multiple imaging modalities, the clusters emerged in just three specific brain regions. Given that these patterns largely mirror those identified in earlier unimodal studies, I wonder whether the inclusion of multiple modalities—which is one of the key points of the work—offered any substantial value. This ties back to my earlier point about the need for a systematic evaluation of the proposed approach. It would be highly beneficial for the authors to thoroughly assess the individual contribution of each modality to the clustering itself. The spatial layout of the clusters is to some extent unsurprising. First, the authors selected a three-cluster solution rather than four, prioritising a pattern in which the clusters remained more spatially coherent. Second, it is well known that white-matter microstructure differs across brain regions (Brander et al., 2010; Lawrence et al., 2021), because local fibre architecture varies—from highly aligned, single-direction bundles to regions with dense fibre crossings. Could this very anatomical structure, by design, influence how the clustering unfolds? It is also interesting to see that apparently there are no differences in the “temporal dynamics”. If they unfold in the same manner, what distinguishes them from each other?

Response: *We added a supplementary analysis investigating the variance explained in the average pathophysiology maps by the spatial clusters for each marker, from which we can infer the contribution of each marker to the clustering solution. Details are available in the Results section (p. 10-11, lines 226-230):*

“We also observed that the clustering solution explained most of the variance in the average pathophysiology spatial maps of MD, ICVF, and ISOVF (R^2 of 0.81, 0.78, and 0.63, respectively; Supplementary Figure 4), a moderate amount for FA and T2* (R^2 of 0.39 and 0.29, respectively), and almost none for QSM and OD (R^2 of 0.04 and 0.00, respectively).”

in the Discussion section (p. 22, lines 458-459):

“These spatial patterns were mostly influenced by the MD, ICVF, and ISOVF markers (...)”

and in the Methods section (p. 30, lines 668-670):

“We calculated the variance explained (R^2) in the maps of between-subject averaged WMH pathophysiology (Figure 2A) by the spatial clustering solution (Figure 2B) for each microstructural metric using ANOVAs (Supplementary Figure 4).”

Second, the variations in white matter microstructure (what we refer to as spatial anatomical contrast in the text) are removed by the voxel-wise normalization approach, which minimizes the impact on the clustering. This clarification was added in the Results section (p. 8, lines 186-187):

“In other words, regional variations in normal white matter microstructure are removed.”.

Third, while SuStaIn did not find temporal subtrajectories, this doesn't exclude the possibility that differences exist at the more granular level in terms of smaller differences in the amplitude of abnormality for different markers, while the general ordering between markers remains relatively stable, which we find in our disorders analyses. We now discuss other possibilities as to why SuStaIn did not detect subtrajectories in the Discussion section (p. 23, lines 483-491):

“Furthermore, we hypothesize that SuStaIn did not uncover subtrajectories because the subgroup of individuals with neurodegeneration-related WMHs could be very small and vastly over-weighted by the subgroup with vascular-related WMHs (e.g., a few hundred versus 10s of thousands). Another limitation of SuStaIn is that it depends on the sensitivity of the markers. It is possible that small deviations in fiber-sensitive metrics like OD are biologically meaningful (as evidenced by the differential OD effects in vascular and neurodegenerative diseases) but were overshadowed by less disease-specific markers that show very high abnormality (like the MD marker). Further analyses are needed to establish if our results are due to modeling limitations.”

Comment 7: Note: While I talk about many points raised in the introduction, it is clear that many of them apply equally to the discussion.

Response: *Modifications to the Discussion are specified in the other comment's responses, where appropriate.*

Comment 8: Recommendations. Assuming these patterns are indeed valid, the pertinent issue becomes their practical implementation. Must we acquire every imaging modality used in the study, or would a carefully selected subset—or simply applying the resulting cluster masks—be adequate? The authors argue that the prevailing misinterpretation of WMH as being of vascular origin poses a significant obstacle to implementing precision medicine in both diagnosis and treatment. I share this view—if a patient's WMH do not stem from a vascular cause, prioritising the management of cardiovascular risk factors may not be a rational approach. Since this is a recurring theme throughout the manuscript, I would encourage the authors to verbalise it further. What are the concrete benefits of understanding the aetiology of WMH?

Response: *We added a passage describing how our findings can be used by other groups who want to parse WMH etiological heterogeneity in the Discussion section (p. 24, lines 523-530):*

“This is supported by our finding that WMH volume was significantly elevated relative to controls across all three spatial regions in both cerebrovascular disorders and dementia, highlighting the need for a deeper phenotyping of WMH pathophysiology to parse etiologies. We still encourage other groups to use our publicly available parcellation scheme (https://github.com/CoBrALab/WMH_patho_UKB), but note that this is informative but not sufficient to disentangle vascular from neurodegenerative contributions. Based on our observations, metrics derived from multi-shell diffusion MRI acquisitions in particular show promise for identifying WMH pathogenesis.”

We also clarified our amyloid-clearing drug use case and added two specific use cases in the Discussion section (p. 25-26, lines 554-564):

“Second, the use of amyloid-clearing drugs like lecanemab is currently restricted to individuals without comorbid cerebrovascular pathology, including a criteria of “no significant white matter hyperintensities”,⁶⁸ since they are at higher risk of developing potentially severe amyloid-related imaging abnormalities (ARIA).⁶⁸ Clarifying WMH pathogenesis could improve the selection of patients, potentially including people with neurodegeneration-related WMHs who could benefit most from these treatments. Third, identifying neurodegenerative WMHs could complement other AD-specific biomarkers, such as amyloid and tau protein markers, and help detect the disease at earlier stages, when treatments are more likely to be effective. Lastly, our framework could help with the differential diagnosis of vascular and Alzheimer’s disease dementia.”

Comment 9: Introduction – “[...] For example, ... Wallerian degeneration [...]”: it is clear that not all WMH are a product of hypoperfusion, they are multifactorial. Could the authors please elaborate on the importance of this example?

Response: *We now elaborate in the following sentence in the Introduction (p. 3, lines 86-89):*

“This pathological heterogeneity is currently invisible to conventional in vivo imaging and could result in misinterpretation or misdiagnosis of cerebral pathology, hindering the application of precision medicine in diagnosis and treatment as well as complicating the interpretation of research findings”

Comment 10: I wonder whether it is necessary to reference some of the hypotheses mentioned in lines 119 to 121, as it is already clear at that point that these represent only one of several possible pathways through which WMH may develop.

Response: *We added the requested changes in the Introduction (p. 4, lines 107-110):*

“The prevailing hypothesis suggests that fluid infiltration from ventricles and blood vessels triggers inflammation, leading to tissue damage such as demyelination, oligodendrocyte loss,

and, in more severe cases, axonal degeneration;³ however, this is one of several pathways through which WMHs can emerge.¹¹”

Comment 11: Introduction - “[...] most common brain radiological findings [...]”:
brain radiological -> neuroradiological. Introduction - “[...] They appear in most [...]”:
perhaps “are present” instead. Introduction - “[...] are typically detected [...]”:
“appear” perhaps

Response: *The requested changes were made.*

Comment 12: Discussion – The account of how the process unfolds still needs sharper focus, subtler reasoning, and a well-curated thesis. With so much work already completed, what new understanding can we add about the breakdown of white-matter integrity that has not yet been articulated?

Response: *We thank the reviewer for challenging us in going further in our interpretation of our results. We have added the following paragraph in the Discussion section (p. 23-24, lines 504-512):*

“The question remains as to why neurodegeneration would cause hyperintense signals on FLAIR images, which are mostly sensitive to increased fluid content. In vascular-related WMHs, hypoxia/ischemia, potentially leading to blood-brain barrier dysfunction, may result in fluid leakage from the ventricles or the blood vessels, as evidenced by recent works.^{15,56} In neurodegeneration-related WMHs, the death of axons and myelin (captured by the altered OD marker) could leave more space to be filled by extracellular fluid content or inflammatory cells, but without important fluid leakage from ventricles or blood vessels, which could explain the lower MD abnormality in posterior WMHs versus anterior WMHs. More hypotheses are discussed in a recent review.⁵⁷”

Reviewer 3

Summary: The study aims to disentangle vascular and neurodegenerative contributions to white matter hyperintensities (WMHs), which are commonly interpreted as markers of cerebral small vessel disease but may also reflect neurodegenerative processes. Traditional FLAIR imaging lacks specificity to distinguish these mechanisms in vivo. To address this, the authors propose a multimodal MRI framework combining seven microstructural markers with voxel-wise normative modelling to generate subject-level pathophysiology maps. Summing up, the study demonstrates spatial heterogeneity (posterior vs anterior predominance linked to AD vs vascular risk), but the underlying microstructural profiles differ mainly in magnitude rather than composition, and temporal divergence is not supported (SuStaIn found no sub-trajectories, with nearly identical progression curves). Disease-specific contrasts and connectivity analyses are suggestive but remain correlational and limited by cross-sectional design, diagnostic granularity, and metric specificity. Overall, the

work advances spatial characterization and provides preliminary etiologic insights, yet falls short of delivering robust, diverging spatiotemporal signatures as originally hypothesised.

Comment 1: Lines 95–98: The authors appear to misinterpret the meaning of the nomenclature “of presumed vascular origins.” This terminology was adopted because the primary underlying pathology—cerebral small vessel disease—cannot be directly observed in vivo, rather than to imply variability in the pathophysiological mechanisms of WMHs. This fact is another crucial point, that the authors also should acknowledge.

Response: *We thank the reviewer for their time and thorough and comprehensive review of our paper. This issue was also pointed out by Reviewer 2 (Comment 1) and we now have removed all references to the wording “of presumed vascular origins” in the introduction. We also now note that direct investigation of the brain’s small vessels in vivo is currently not feasible in the Introduction section (p. 3, lines 75-78):*

“The presence of WMHs is widely considered a downstream consequence of cerebral small vessel disease and is typically used to detect this disorder, as neuroimaging methods able to directly investigate small blood vessels (<500 µm in diameter) are lacking.^{4”}

Comment 2: Lines 115-117: the authors claim that no study has attempted to investigate spatial pattern of WMH pathophysiology. This is only partially true, as there are indeed studies finding different spatial WMH pathology related to pathophysiological underpinnings and underlying pathology, as the authors acknowledge in the discussion section. A more complete contextualisation at this stage would probably be more valuable.

Response: *This comment is similar to Reviewer 2’s comments 2 and 3. Accordingly, we now discuss prior work in that field in more detail and clarify the specific value added of our set of analyses. The details are added in the Introduction section (p. 4, lines 99-104):*

“Other parcellation schemes include cortical lobes, vascular territories, or white matter tracts.²⁰ In contrast to these hypothesis-driven parcellations, data-driven methods can uncover more complex patterns of WMH spatial heterogeneity, and while such examinations have been done based on the co-occurrence of WMHs,^{12,21} these have not taken into account the underlying pathophysiological alterations. Given the sparsity of WMH prevalence at the voxel-level, such an analysis requires a very large dataset.”

and Discussion section (p. 22, lines 463-470):

“The investigation by Phuah et al. (2022) used spectral clustering on WMH probability maps from FLAIR images, resulting in a 5-cluster solution and found that deep frontal WMHs were associated with hypertension and diabetes, while parietal WMHs were associated with amyloid SUVR.¹² Habes et al. (2018) used non-negative matrix factorization to uncover clusters of covarying WMH volumes and found that high blood pressure and female sex were associated with higher frontal WMH load, while AD genetic risk was associated with higher

dorsal WMH load.²¹ Our pathophysiology-based approach to parsing WMH spatial heterogeneity is thus consistent with this emerging data-driven literature.”

Comment 3: The central construct—“pathophysiology” defined as the z scored deviation from normative NAWM—relies on the assumption that NAWM is sufficiently disease free to serve as a reference. This is unlikely to hold, as cerebral small vessel disease is diffuse and not confined to WMHs. Failing to account for this risks biasing estimates toward zero, particularly in participants with high WMH burden, and the magnitude of this bias is difficult to predict given the unknown contribution of each metric. A sensitivity analysis excluding individuals with high WMH burden could strengthen confidence in the approach and this should issue should be acknowledged in the discussion and/or the limitations.

Response: *This issue was also pointed out by Reviewer 2 (comment 5). We now removed wording related to “healthy” white matter, replacing it with “normal appearing”, and acknowledge the limitation in the Results (p. 8, lines 174-177):*

“We note that NAWM may still be pathological to a certain extent, as has been previously demonstrated.¹⁶ However, pathological NAWM remains ill-defined, and there is currently no established framework to identify it. We thus focus on contrasting WMH from non-WMH tissue, considering that some tissue alterations are part of the NAWM variations.”.

We argue that even excluding individuals with high WMH burden does not guarantee that NAWM is healthy, hence would only partly mitigate the issue raised. Furthermore, we argue that the framework implemented here does not rely on the assumption of healthy white matter to be efficient. Given the sample size of the dataset (n=32,000), the presence of some pathological NAWM in some subjects included in the normative modeling should have a relatively small effect. This could still result in the z-score metrics being a little bit higher or lower relative to the hypothetical scenario of excluding these pathological voxels. However, the effect of age and sex would still be accurately removed. Furthermore, the most important feature of this voxel-wise normative modeling step is removing the spatial contrast induced by the anatomy, leaving spatial patterns that are caused by the presence of WMHs and not by the anatomy. This crucial aspect should not be significantly affected by the presence of some pathological values in the normative modeling, particularly given that even when present, they would be at mild (sub-threshold) severity levels.

Comment 4: Clusters 2 (posterior) and 3 (anterior) show very similar pathophysiological profiles, differing mainly in the magnitude of abnormality rather than in distinct marker patterns. This similarity is further reinforced by the SuStaIn modeling, which shows nearly identical pseudo-time progression of biomarkers for these two clusters. This raises concerns about whether the clustering truly captures biologically meaningful heterogeneity or simply partitions WMHs by spatial location.

Response: *We added clarifications regarding our interpretation of the spatial clustering results in the Results section (p. 11, lines 230-233):*

“The posterior and anterior clusters mostly differed in the severity of pathophysiological markers, but still revealed distinct spatial signatures. Further characterization of specific pathophysiological alterations is needed to assess potential etiological signatures.”

and our hypotheses as to why SuStaIn did not find any subtrajectories in the Discussion section (p. 23, lines 483-491):

“Furthermore, we hypothesize that SuStaIn did not uncover subtrajectories because the subgroup of individuals with neurodegeneration-related WMHs could be very small and vastly over-weighted by the subgroup with vascular-related WMHs (e.g., a few hundred versus 10s of thousands). Another limitation of SuStaIn is that it depends on the sensitivity of the markers. It is possible that small deviations in fiber-sensitive metrics like OD are biologically meaningful (as evidenced by the differential OD effects in vascular and neurodegenerative diseases) but were overshadowed by less disease-specific markers that show very high abnormality (like the MD marker). Further analyses are needed to establish if our results are due to modeling limitations.”

Despite these more negative SuStaIn results, we still demonstrate the usefulness of the spatial clusters to capture biologically-meaningful heterogeneity with our observations of spatially distinct sex effects (Figure 3B-C), spatially distinct and reproducible effects across neurodegenerative and cerebrovascular disorders (Figure 4), and differential connectivity profiles with grey matter regions and their associated vulnerability to Alzheimer’s pathology (Figure 5). We also now clarify our claim of differential etiology by spatial region in the Discussion section (p. 24, lines 523-530):

“This is supported by our finding that WMH volume was significantly elevated relative to controls across all three spatial regions in both cerebrovascular disorders and dementia, highlighting the need for a deeper phenotyping of WMH pathophysiology to parse etiologies. We still encourage other groups to use our publicly available parcellation scheme (https://github.com/CoBrALab/WMH_patho_UKB), but note that this is informative but not sufficient to disentangle vascular from neurodegenerative contributions. Based on our observations, metrics derived from multi-shell diffusion MRI acquisitions in particular show promise for identifying WMH pathogenesis.”

Comment 5: The clustering approach is based on a population-averaged pathophysiology map restricted to voxels with high WMH prevalence. While this ensures stability, it assumes a single canonical pattern of WMH pathophysiology across individuals and does not allow for subject-level heterogeneity or co-pathology. This design choice may also explain why the posterior and anterior clusters exhibit very similar pathophysiological signatures, differing mainly in magnitude and location rather than in distinct biological profiles.

Response: *This is true and is a limitation of our approach. We explored a few other strategies for including subject variability in the spatial clustering (i.e., directly clustering 7*

*voxel*subject matrices) but encountered the problem of sparsity because most of these matrices were empty since, for each subject, WMHs cover a small amount of the total white matter that is highly variable across subjects. In each of these matrices, the non-WMH voxels had zero value and WMH voxels had non-zero values (the pathophysiology z-score). This resulted in clusters that were simply driven by the presence or absence of WMHs and neglected the pathophysiology variance (i.e., driven by the patterns of non-zero voxels, not by the variance within the non-zero voxels). Since our goal was to explore the spatial patterning based on pathophysiological estimates, we instead opted to make a dense matrix by calculating the average pathophysiology maps across subjects, which solved this issue. Our strategy was then to investigate spatially-overlapping subtypes using SuStaIn; however, this experiment did not reveal any subtrajectories. We added hypotheses as to why SuStaIn did not find any subtypes in the Discussion section (p. 23, lines 483-491):*

“Furthermore, we hypothesize that SuStaIn did not uncover subtrajectories because the subgroup of individuals with neurodegeneration-related WMHs could be very small and vastly over-weighted by the subgroup with vascular-related WMHs (e.g., a few hundred versus 10s of thousands). Another limitation of SuStaIn is that it depends on the sensitivity of the markers. It is possible that small deviations in fiber-sensitive metrics like OD are biologically meaningful (as evidenced by the differential OD effects in vascular and neurodegenerative diseases) but were overshadowed by less disease-specific markers that show very high abnormality (like the MD marker). Further analyses are needed to establish if our results are due to modeling limitations.”

Comment 6: For both UK Biobank and ADNI, neurodegenerative dementia diagnoses appear to be based on clinical criteria rather than biomarker-confirmed status (e.g., amyloid or tau PET, CSF). This distinction is important because clinical diagnosis alone can misclassify cases, especially in mixed or atypical presentations, and may dilute disease-specific WMH signatures. The potential for misclassification is particularly relevant when interpreting the posterior WMH pattern as Alzheimer’s-related. Please clarify.

Response: *We now added a supplementary analysis comparing WMH pathophysiology between amyloid-positive MCI and AD subjects (combined) and amyloid-negative CN subjects. Details are available in the Results section (p. 19, lines 406-411):*

“We additionally compared amyloid-positive MCI and dementia subjects (n=42) to amyloid-negative cognitively normal subjects from ADNI (Supplementary Figure 15A) in order to isolate biomarker-confirmed cases. Effects were similar to the other ADNI patterns and showed higher correlations with effect size patterns in the neurodegenerative grouping (average $r = 0.63$) compared to the vascular grouping (average $r = 0.32$; Supplementary Figure 15B).”

and the Methods section (p. 34, lines 752-756):

“However, we compared amyloid-positive MCI and dementia subjects combined (n=42) to amyloid-negative cognitively normal subjects from ADNI (n=73; Supplementary Figure 15). Amyloid positivity was determined from PET data ([18F] florbetaben (FBB) or [18F] florbetapir (FBP)) using tracer-specific cerebellum-normalized cortical uptake SUVR thresholds.⁸⁵”

Comment 7: The reported correlation between the posterior WMH cluster's cortical connectivity profile and tau PET distribution is intriguing, but its biological specificity warrants caution. Occipital and posterior association cortices-regions contributing the largest number of fibres to posterior white matter-are also heavily connected hubs in the brain. If these regions naturally exhibit high streamline density and are simultaneously among the earliest or most prominent sites of tau deposition in Alzheimer's disease, the observed association could reflect network topology rather than a mechanistic link between WMHs and tau pathology. Please clarify: how did the analysis account for baseline connectivity density or hubness? Could the correlation be driven by the fact that posterior WMH clusters inherently connect to high-degree nodes (e.g., visual association cortices)? Would the association remain significant after normalising for streamline count or using a null model that preserves degree distribution (beyond the spin test)? Consider tempering the interpretation: the result may indicate spatial co-localization of two processes in highly connected regions, rather than direct evidence of Wallerian degeneration from tau-positive cortex. Including these caveats or performing a sensitivity analysis would strengthen the claim and avoid overinterpretation.

Response: *The connectivity maps are indeed normalized by dividing by the total streamline counts across all clusters. This results in the relative streamline density for each spatial clusters relative to the other clusters, thus normalizing for hubness. We added this clarification in the main text in the Results section (p. 20, lines 422-424):*

“Since larger ROIs will have more streamlines, and thus higher raw connectivity metrics, we further normalized the values by dividing by the number of streamlines of each ROI across all three clusters.”.

We also now temper our interpretation of this result in the Discussion section (p. 23, lines 498-504):

“This interpretation is further supported by our finding that posterior WMH regions structurally connect cortical regions prone to early tau accumulation. We speculate that our observations point to a link between tau pathology in inferior temporal regions, degeneration of the associated fiber populations, and the appearance of posterior FLAIR hyperintensities. Cortical regions connecting through WMHs have been shown to be disproportionately affected relative to unconnected regions.^{54,55} Further research is needed to test the causality of this hypothesis. ”

Comment 8: The statement on lines 415-417 claims that the data-driven approach "revealed that deep-anterior WMHs had more severe pathophysiology than periventricular-posterior WMHs, which does not seem to be caused by spreading patterns." While this is technically accurate in terms of severity, the interpretation appears to overstate novelty. Based on the presented results: a) the most salient axis of heterogeneity still aligns with the traditional periventricular vs deep WMH distinction, as the periventricular cluster clearly differs from the other two in both magnitude and pattern, b) the additional anterior/posterior split adds spatial nuance but does not fundamentally overturn the established classification; rather, it refines it, c) furthermore, the posterior and anterior clusters share highly similar pathophysiological signatures and nearly identical SuStaIn trajectories, suggesting that the main biological contrast remains between periventricular and deep WMHs. We recommend tempering the claim and clarifying that the findings largely confirm the periventricular vs deep dichotomy, while providing additional granularity within the deep WMH category. A brief discussion of how this refinement compares to prior heuristic classifications would strengthen the interpretation.

Response: *We agree with the reviewer that the new clustering methods does not contradict the periventricular/deep parcellation and instead refines it. Wording was adapted as such in the Discussion section (p. 22, line 456):*

"Our data-driven approach refined the often-used periventricular/deep parcellation (...)"

Comparison with prior heuristic parcelations is now discussed in the Discussion section (p. 22, lines 458-461):

"These spatial patterns were mostly influenced by the MD, ICVF, and ISOVF markers, and were a mix of different heuristic parcellations since they incorporate both the medial-to-lateral axis (similar to periventricular/deep) and the anterior-to-posterior axis (represented in the lobar parcellation)."

We also clarify that the data-driven "periventricular" cluster is much more narrow in coverage than the a priori region, as now discussed in the Results section (p. 12, lines 250-252):

"This limited overlap between cluster 1 and the "periventricular" a priori region is due to the much broader coverage of the a priori region (<8mm from the ventricle, as is typically done),¹⁹ as compared to the data-driven cluster."

Comment 9: Lines 438-442: while the authors interpret the correlation between posterior WMHs and tau-prone cortical regions as evidence of a mechanistic link, this association could equally reflect the inherent topology of white matter pathways (see previous comments). Without tract-specific degeneration evidence or longitudinal data, the claim of a causal chain from tau accumulation to fibre loss and WMH formation remains speculative.

Response: *We agree with the reviewer that the evidence shown remains correlational and not causal, and that further research is needed to test the hypothesis that we proposed based on our observational results. We now specify that this study was observational/correlational and not experimental/causal in various parts of the manuscript, including the Introduction section (p. 4, lines 118-119):*

“Our observational study aimed to parse the pathophysiological, spatial, temporal, and ultimately etiological heterogeneity of WMHs.”,

the Results section (p. 19, lines 413-414):

“However, these results remain correlational and require validation using gold-standard ex vivo methods.”,

and Discussion, specifically addressing the issue raised by the reviewer regarding the lack of tract-specific degeneration evidence (p. 23, lines 500-504):

“We speculate that our observations point to a link between tau pathology in inferior temporal regions, degeneration of the associated fiber populations, and the appearance of posterior FLAIR hyperintensities. Cortical regions connecting through WMHs have been shown to be disproportionately affected relative to unconnected regions.^{54,55} Further research is needed to test the causality of this hypothesis. ”.

Comment 10: Although the authors argue that spatiotemporal pathophysiological signatures can disentangle vascular and neurodegenerative WMHs, they also acknowledge that anterior WMHs are associated with both small vessel disease and neurodegeneration (lines 450–454), which “cannot purely be distinguished by neuroanatomical location.” This admission appears to contrast with their spatial clustering approach, as their primary evidence for differentiation relies heavily on location (posterior vs anterior) combined with microstructural profiles that are largely similar in pattern and differ mainly in magnitude.

Response: *This issue raised is similar to Comment 8 by Reviewer 2. We now clarify the limitations of a purely spatial approach in differentiating etiologies and make more specific comments on how our work should be used by others in the Discussion section (p. 24, lines 523-530):*

“This is supported by our finding that WMH volume was significantly elevated relative to controls across all three spatial regions in both cerebrovascular disorders and dementia, highlighting the need for a deeper phenotyping of WMH pathophysiology to parse etiologies. We still encourage other groups to use our publicly available parcellation scheme (https://github.com/CoBrALab/WMH_patho_UKB), but note that this is informative but not sufficient to disentangle vascular from neurodegenerative contributions. Based on our observations, metrics derived from multi-shell diffusion MRI acquisitions in particular show promise for identifying WMH pathogenesis.”

Comment 11: The discussion section ventures into clinical implications (lines 455-464; e.g., antihypertensive treatment decisions and eligibility for amyloid-clearing drugs) that are not directly supported by the presented data. Given the cross-sectional design and the modest, partially overlapping pathophysiological patterns, these statements risk overstating the immediate clinical applicability of the findings. While we concur that a more nuanced approach may have potential clinical relevance, it should be framed more cautiously as a future perspective rather than as a practical implication.

Response: *We have now nuanced our paragraph on potential use cases in the Discussion (p. 25, lines 548-551):*

“While our findings will require further validation before being integrated into clinical settings, using in vivo MRI-based pathophysiological markers to disentangle neurodegenerative and vascular contributions to WMHs could have broad implications for the clinical management of patients and the interpretation of research findings and clinical trials.”

Comment 12: Voxel inclusion thresholds (Methods, Spatial clustering section, lines 710, 724–725): please justify the fixed cut-offs used for voxel inclusions (WMH prevalence < 30, NAWM < 5,000). In a sample of 32,526, these map to 0.09% and 15.4% of participants, respectively, which appears arbitrary. We recommend adopting either (a) percentage-based thresholds (e.g., WMH \geq 0.5%, NAWM \geq 15%) or, preferably, (b) precision/estimate–stability based thresholds (e.g. thresholds that bind the SEM of voxelwise mean z-scores).

Response: *Since we used these thresholds only to chose which voxels will be included in the averaged pathophysiology maps for clustering, our choice of a fixed-value cutoff (as opposed to percentage-based) was based in the rationale that we only needed enough values for what we considered to be a stable and robust estimate of WMH pathophysiology. We chose a WMH prevalence value of 30 to balance achieving both a stable estimate of the average value while still including enough voxels to achieve a broad coverage of the white matter. We also chose a NAWM prevalence threshold of 5000 in order to have enough observations so that our Bayesian linear regression normative model was stable and because voxels where almost everybody has a WMH are less interesting. We now added two analyses: 1) plotting the SEM relative to NAWM and WMH prevalence in Supplementary Figure 2, detailed in the Methods section (p. 30, lines 657-661):*

“This was necessary to 1) accurately model the NAWM microstructure trends using our Bayesian linear regression approach, and 2) obtain robust estimates of the WMH average

pathophysiology. These thresholds were informed by prevalence by standardized mean error (SEM) plots (Supplementary Figure 2), since we observed SEM plateaus starting at these cutoff values.”

and 2) calculating clusters with WMH prevalence > 1 and NAWM prevalence > 1000 shown in Supplementary Figure 5B, detailed in the Results section (p. 13, lines 274-265):

“This yielded a parcellation that was comparable but more spatially contiguous relative to a clustering solution where voxels were not excluded based on WMH prevalence (Supplementary Fig. 7B).”

Comment 13: We suggest rephrasing lines 323–324 for clarity. The current sentence reads “Control subjects had no endocrine, circulatory, behavioural, nervous, or metabolic-disease related lifetime diagnoses, with some exceptions (n = 10,629).” The phrase “with some exceptions” is ambiguous and does not align well with the explicit sample size. Instead, we recommend specifying the rationale for these exceptions, for example: “Control subjects had no endocrine, circulatory, behavioural, nervous, or metabolic disease-related lifetime diagnoses, except for common conditions considered unlikely to impact WMH burden (e.g., migraine, depressive episode).” Additionally, please note that the assumption that migraine has no influence on WMH burden is debatable and should be acknowledged or justified with supporting evidence.

Response: *In order to remove this ambiguity, we have rerun the analysis without making any exceptions for specific diagnoses to include in the control group, which reduced the number of control participants from 10,629 to 8,528. The results are virtually identical and all interpretations remained unchanged. To satisfy the reviewer’s curiosity, we note that individuals with a Migraine diagnosis did not have more WMHs relative to individuals without a diagnosis (average beta = -0.008), while the depression diagnosis (average beta = 0.04) and anxiety diagnosis (average beta = 0.04) had significantly more WMHs relative to individuals without a diagnosis, but with very low effect sizes.*

Comment 14: The analysis reports standardised betas for volume differences by sex (age adjusted), and for pathophysiology (additionally volume adjusted). These are small to modest effects but consistent. Consider adding absolute volume differences (mL) alongside betas for interpretability.

Response: *While we understand that effect sizes expressed as absolute volume differences are useful and easily interpretable to clinicians, we note that it was necessary to log-transform WMH volumes to respect the linear model assumptions of normality (since distributions very heavily left-skewed, as is usually the case for WMH volume distributions). This is standard when analyzing WMHs and is recommended (Roseborough et al., 2022, Alzheimer’s and Dementia). Thus, the unstandardized betas would still not be expressed in mL. On the other hand, not log-transforming WMH volumes would render the fit of linear models less representative of the true trends present in the data.*

Reviewer 4

Summary: The manuscript by Parent et al. called ‘Characterizing spatiotemporal white matter hyperintensity pathophysiology in vivo to disentangle vascular and neurodegenerative contributions’ describes a study in which based on multi-modal brain MRI the authors have identified different spatial patterns of white matter hyperintensity pathology using a data driven and voxel-wise approach on a large population-based dataset (UKBB). To assess the tissue properties of white matter hyperintensity pathology, the following microstructural markers were used: MD, FA, ISOVF, ICVF, OD, T2* and QSM. The extensive analyses performed in this study broadly consist of the following steps: 1) Using voxel-based normative modelling of each of these markers based on only the NAWM voxels, resulted in age- and sex- specific maps of expected healthy microstructure. Using these maps for the normalization, individual-level pathophysiology maps are created. 2) To assess patterns of shared pathophysiological properties, spectral clustering was performed. Averaging the individual-level pathophysiology maps, only using the WMH voxels, creates seven between-subject averaged WMH pathophysiology maps, where the value of the voxel in a map represents the average deviation from healthy NAWM (expressed in SD) at that voxel location. These between subject maps are the input of the spectral clustering, which essentially leads to clustering the voxels based on the values of the 7 microstructural markers in those voxels. This resulted in the following spatial patterns: Periventricular (Cluster 1), Posterior (Cluster 2) and Anterior (Cluster 3). There was no consistent agreement of these spatial patterns and often-used WMH parcellations (periventricular vs deep, lobar regions, cerebral artery territories). 3) temporal pathological cascades per spatial cluster were determined using SuStaIn (disease progression model which performs subtyping as well as staging) and provides information on the order at which changes per microstructural marker occurs within each cluster. In general the progression patterns was similar across the clusters, except for the amount of ‘events’ that take place (corresponds with the amount of stages) in each cluster. The authors conclude that these spatial patterns show that periventricular-anterior WMHs have more severe pathophysiology than periventricular-posterior WMHs. Temporal progression pattern of WMHs were similar across the different clusters and is in general in agreement with the hypothesis that water infiltration and inflammation are the early events in the progression pattern of WMH. Based on the dominance of WMHs in the posterior cluster in patients with dementia/AD and differences in OD, they conclude that the WMHs in individuals with dementia/AD are neurodegeneration-related. This paper is an important contribution to the field, where the advanced data-driven approach provides relevant new insights in the pathophysiology behind WMH. The authors did not stop when spatial clusters with similar pathophysiological signatures were identified, but also assessed temporal progression of the microstructural markers and evaluated in which WMH clusters the WMH volume occurred in vascular and neurodegenerative disorders and which microstructural markers deviated most compared to controls. Putting all this information together the authors suggest that WMH in patients with AD/dementia is the result of selective degeneration of fibers. I believe it is an excellent paper and it was a pleasure to read it. I only have a few minor concerns/questions for the authors:

Comment 1: Regarding the exclusion of voxels for the between-subject averaged pathophysiology maps, used as input for the spectral clustering, how are the cut off values of 30 (WMH) and 5000 (NAWM) chosen? A prevalence of only 30 individuals with WMH at a certain voxel in the large UKBB dataset seems still very low to me.

Response: *We thank the reviewer for their time and thorough and comprehensive review of our paper. The issue raised is similar to Comment 12 by Reviewer 3. While these thresholds were chosen subjectively, we chose a WMH prevalence value of 30 to balance achieving both a stable estimate of the average value while still including enough voxels to achieve a broad coverage of the white matter. We also chose a NAWM prevalence threshold of 5000 in order to have enough observations so that our Bayesian linear regression normative model was stable and because voxels where almost everybody has a WMH are less interesting. While we agree that a prevalence of 30 in the context of UKBB is low, this speaks to the sparsity of voxel-wise WMH data.. We now added two analyses: 1) plotting the SEM relative to NAWM and WMH prevalence in Supplementary Figure 2, detailed in the Methods section (p. 30, lines 657-661):*

“This was necessary to 1) accurately model the NAWM microstructure trends using our Bayesian linear regression approach, and 2) obtain robust estimates of the WMH average pathophysiology. These thresholds were informed by prevalence by standardized mean error (SEM) plots (Supplementary Figure 2), since we observed SEM plateaus starting at these cutoff values.”

and 2) calculating clusters with WMH prevalence > 1 and NAWM prevalence > 1000 shown in Supplementary Figure 5B, detailed in the Results section (p. 13, lines 274-265):

“This yielded a parcellation that was comparable but more spatially contiguous relative to a clustering solution where voxels were not excluded based on WMH prevalence (Supplementary Fig. 7B).”

Comment 2: Did the authors explore the relationship between stage and WMH volume? Since there is not a very strong relationship between WMH volume and Z-score values of the 7 markers within the periventricular cluster (as shown in Extended Figure 4), perhaps comparing the staging with the volumes provides a clearer picture of the meaning of WMH volume in this location.

Response: *We now investigated associations between SuStaIn stages and WMH volumes in Supplementary Fig. 11B, and interpret these new results in the Results section (p. 14, lines 309-312):*

“SuStaIn stages were correlated with regional WMH volumes (Spearman ρ : Periventricular = 0.41, Posterior = 0.66, Anterior = 0.85; all $p < 0.001$; Supplementary Fig. 11B) but still showed substantial variation not explained by WMH volumes.”

Comment 3: In the results section where spatial clustering is described, I'd suggest to change the sentence: "Subject-level maps of WMH pathophysiology were first collapsed into an average for each of the seven markers," to something like: "Between-subject averaged WMH pathophysiology maps were retrieved by averaging all subject-level maps, resulting in one between-subject map per marker". I found the description in the methods much clearer and the term 'collapsed' and then only stating 'average of each of the seven markers' in the results section are confusing in my opinion.

Response: *We agree that the proposed wording is clearer and have adopted it in the Results section (p. 10, lines 206-208):*

"Mean WMH pathophysiology maps were calculated by averaging all subject-level maps, resulting in one between-subject map per marker."

Comment 4: I struggle a bit with the interpretation or the way the authors describe the level of pathophysiology, especially in the results section of the temporal pathophysiological cascades. For example It is stated that: 'Periventricular WMHs showed low pathophysiology across temporal stages'. I am not sure whether 'low pathophysiology' makes sense here, I don't understand what the authors meant by this. The pathological events are defined as Z-scores, or in this case steps of 0.5, 1, 1.5 SD. Using SuStaIn you could also choose to label 0.25 SD as an event, so the number of events does not necessarily reflect severity or level of pathology, in case that is what the authors meant with low pathophysiology. Furthermore, Although in this region it only goes until 1.5 SD, it could in terms of 'pathology' perhaps be as severe as reaching 5SD in another region? Later on in the paragraph the authors see the Z-score as level of abnormality, however assigning the biological meaning of a certain deviation from the mean is not something SuStaIn does, as user of the algorithm you have to choose what you define as an event. To me it makes sense here to use the Z-scores as the authors did (with the addition of 0.5 and 1.5 SD), but in my opinion the authors should be careful with the interpretation and formulation.

Response: *We appreciate the need for a more nuanced interpretation of the SuStaIn z-score thresholds and interpretations. To clarify, we meant to interpret the absolute z-score thresholds reached and not the number of events. In the text, we have added a clarification to how the z-score thresholds are determined empirically (p. 14, lines 291-293):*

"Pathophysiological events were determined empirically on a per-marker basis (at least 1% of participants needed to reach the threshold; Methods; Supplementary Fig. 8)."

and we changed our interpretation to not infer biological meaning from SuStain stages (p. 14, lines 300-302):

"In periventricular WMHs, we observed low abnormality in most markers, with the maximum z-score thresholds being set to 1.5 SD (for MD and ICVF) according to our heuristic. ICVF was the earliest marker to be altered."

Our interpretation of the z-score levels as an indication of abnormality does not stem from SuStaIn modeling, but from our normative modeling step. We thus keep the wording “abnormality” instead of “pathology” in the context of SuStaIn. We also discuss the dependence of SuStaIn on the sensitivity of the underlying metrics in the Discussion section (p. 23, lines 486-491):

“Another limitation of SuStaIn is that it depends on the sensitivity of the markers. It is possible that small deviations in fiber-sensitive metrics like OD are biologically meaningful (as evidenced by the differential OD effects in vascular and neurodegenerative diseases) but were overshadowed by less disease-specific markers that show very high abnormality (like the MD marker). Further analyses are needed to establish if our results are due to modeling limitations.”

Comment 5: Did the authors assess whether the staging within each of the spatial clusters correlated with WMH volume in that region? You’d expect that individuals with low WMH in a spatial cluster would be staged early in the temporal cascade. It would also be interesting to see whether individuals with no visible WMH yet are always ending up in stage=0, or that they already show signs of deviations in some of the markers. In that case the staging could be an interesting predictive marker of WMH in the future potentially?

Response: *In line with the reviewer’s Comment 2, we now investigated associations between SuStaIn stages and WMH volumes in the Extended Figure 4B, and interpret these new results in the Results section (p. 14, lines 309-312):*

“SuStaIn stages were correlated with regional WMH volumes (Spearman ρ : Periventricular = 0.41, Posterior = 0.66, Anterior = 0.85; all $p < 0.001$; Supplementary Fig. 11B) but still showed substantial variation not explained by WMH volumes.”.

These results indeed showed that individuals with low WMH volumes tend to be at earlier pathophysiological stages. Regarding individuals with no visible WMHs, this data is not included in our analyses, as our experiment is centered around WMHs that were detected by our segmentation algorithm, so by definition visible on FLAIR or T1w images. When there is no WMH voxels in the spatial regions defined in our clustering step, the SuStaIn stage is defined as 0. In future work, we aim to explore pathophysiological alterations to the tissue before any WMHs are visible using UK Biobank longitudinal data.

Other changes

- Grammar
- Put Extended Data figures into the Supplementary Data
- Abbreviated Alzheimer’s disease with AD throughout the manuscript
- Adapted figure numbers

Summary of the main changes

We were encouraged that two of the four reviewers indicated that the manuscript was ready for publication. Both Reviewers 1 and 4 remarked that they found we had “adequately” and “convincingly” addressed “all” previous comments. Reviewer 3 provided similarly positive feedback, and we found that their new comments and suggestions were very helpful and improved the clarity and completeness of our manuscript. We have also carefully considered the remaining comments from Reviewer 2. Some of these concerns revisited issues previously raised by other reviewers that had already been addressed to their satisfaction, while others were broader in nature and did not always lend themselves to direct methodological or analytic resolution. In response, we focused on clarifying the scope of our aims, sharpening the distinction between empirically supported findings and biological interpretation, and adding new quantitative analyses that directly assess the extent to which WMH spatial and microstructural features differentiate cerebrovascular from neurodegenerative disease contexts.

In this revision, we made both analytical additions and language clarifications to improve the clarity, robustness, and impact of the manuscript. Most importantly, we added an analysis demonstrating that MRI-based WMH pathophysiological data and our customized data-driven parcellation lead to significant improvements in the separation of cerebrovascular and neurodegenerative disease contexts compared to traditional approaches based on periventricular/deep WMH volumes.

The Discussion was also revised to more clearly distinguish empirically supported findings from biological interpretation, reducing speculative tone while preserving statements that were added in response to previous reviewer requests. To further clarify the manuscript’s contribution, we added a concluding paragraph summarizing the main findings and their relevance relative to the existing WMH literature.

A sensitivity analysis was added, where we reproduced our disease-specific patterns (Figure 4) using a parcellation that did not include rare WMH voxels using our region-growing approach. We also added a figure showing the NAWM and WMH prevalence maps, overlaps between our clusters and an atlas of fiber types, and more details in the README of our code repository to facilitate reproducibility and replication that could be performed in the UK Biobank or other cohorts.

Reviewer 1

Summary: The authors have adequately addressed all comments.

Response: *We thank the reviewer for their efforts in reviewing our manuscript.*

Reviewer 2

Summary: Apologies if I am stating the obvious, but part of the revision process is to improve the quality of the paper through feedback from other researchers and by thoughtfully accepting or rebutting the points raised during review. The keyword here is “improve”. While some aspects are indeed clearer in this second version of the manuscript, some of the authors’ responses honestly diminish the significance of the work.

Comment 1: Reviewer #1 (Comment 3), Reviewer #2 (Comment 8), and Reviewer #3 (Comment 4 and 10): Since clusters 2 and 3 differ primarily in overall severity, an important question is whether they actually represent distinct aetiologies or underlying pathophysiological mechanisms. The response provided—namely, that aetiological differences cannot be inferred solely from this finding and that further characterisation of specific pathophysiological alterations is required—raises two concerns.

Response: *We thank the reviewer for their careful reading and the opportunity to clarify the scope, aims, and empirical contributions of this work. Several concerns appear to stem from differing expectations regarding the level of mechanistic inference that can be supported by in vivo MRI-based analyses, which we address explicitly below. Regarding the question of whether etiological differences could be distinguished purely from spatial patterns, we wish to point out that post-mortem studies tend to agree with our assessment and findings (e.g., McAleese et al., 2021: “To conclude, this pilot data suggest that frontal WML in AD may result from both increased arteriolosclerosis and AD-associated degenerative changes”). We therefore believe that this is not a failure of our approach, but a general feature of the problem at hand, and that expecting spatial patterns alone to yield discrete aetiologies is not realistic. This is also clearer in the new analysis that we describe in our response to Comment 2.*

Comment 2: Given that the stated aim of the manuscript was to “disentangle vascular and neurodegenerative contributions” and to “parse the pathophysiological, spatial, temporal, and ultimately aetiological heterogeneity of WMHs”, I find it difficult to conclude that the current version fully meets these objectives. The authors appear to address this discrepancy primarily by noting in the limitations that the study’s aims were not entirely achieved. While this is an important acknowledgement, it also suggests that key elements of the intended contribution remain underdeveloped, leaving the work feeling incomplete in its present form. This is not merely a matter of wording choices, but rather a concern that relates to the conceptualisation, study design, and interpretation of the findings. The inability to disentangle aetiology also implies that the discussion requires reconsideration. For example, the authors state that “While our findings will require further validation before being integrated into clinical settings, using in vivo MRI-based pathophysiological markers to disentangle neurodegenerative and vascular contributions to WMHs could have broad implications for clinical management and for the interpretation of research studies and clinical trials”. This may be true in principle, but it does not sound fitting given the earlier acknowledgement that such disentanglement was not actually achieved in the study.

Response: *We have added an analysis to answer the reviewer's current issues with our manuscript, where we quantify the extent to which our approach is able to disentangle cerebrovascular from neurodegenerative pathologies by training a machine learning model to classify stroke from cognitive impairment (i.e., MCI/dementia) purely based on different combinations of WMH predictors. This analysis demonstrates that 1) WMH microstructure provides substantial independent discriminatory information beyond volume alone, and 2) our custom spatial parcellation consistently outperforms the conventional periventricular/deep scheme on this task. This is described in the Results section (p. 20-21, lines 418-437):*

“As a proof of concept, we investigated whether a machine learning model could differentiate vascular from neurodegenerative pathologies based on our derived spatial and pathophysiological WMH data. We trained Lasso logistic regression models to classify individuals with stroke diagnoses ($n=642$) from individuals with AD-related dementia or MCI diagnoses ($n=127$), combining data from UKB and ADNI. WMH variables were additionally harmonized across datasets using ComBat (see Methods).⁴⁶ Models were trained using 5-fold cross-validation repeated 100 times. We used 6 sets of predictors: WMH volumes only or WMH volumes and microstructure, either unparcellated, following the periventricular/deep parcellation, or using our custom clusters (distributions of area under the receiver operating characteristic curve (AUROC), balanced accuracy, and F1 score are available in Supplementary Fig. 18). Performance was compared using paired Wilcoxon tests. Unparcellated WMH volumes performed poorly in differentiating stroke from cognitive impairment (AUROC mean[SD]: 0.562[0.001]) and improved using clustered volumes (periventricular/deep: 0.630[0.005]; custom clusters: 0.666[0.006]). Performance was noticeably elevated when adding microstructural data (unparcellated: 0.653[0.007]; periventricular/deep: 0.697[0.009]; custom clusters: 0.706[0.011]). All performance differences were significant at $p(\text{Bonferroni}) < 0.001$. Results were similar for balanced accuracy and F1 score. Thus, we observed that 1) WMH microstructural information adds meaningful predictive value for disentangling pathologies and 2) our custom clusters performed better than the traditional periventricular/deep parcellation on this task.”

the Discussion section (p. 25-26, lines 531-545):

“Our prediction framework, where we classified individuals with MCI and dementia from individuals with stroke, demonstrated that our approach using customized spatial regions and microstructural WMH information meaningfully outperformed the traditional approach of only using WMH volumes following the periventricular/deep parcellation (mean AUROC of 0.706 for the former and 0.630 for the latter). Still, the highest accuracy remained moderate, highlighting the inherent difficulty of the task, driven in part by the challenge of obtaining “pure” groups (i.e., a group with only AD pathology and no cerebrovascular pathology, and vice-versa) given the high degree of co-presence of the two disorders.⁵⁷ However, this analysis should be considered as a proof-of-concept since its primary goal was to demonstrate that WMH characteristics between the two groups systematically differ to the degree that this

information can be used to classify the two disorders. In other words, we wanted to investigate this classification purely based on WMH characteristics, and therefore did not use more disease-specific markers (e.g., blood pressure, amyloid PET, whole brain features) that would undoubtedly improve the accuracy for practical applications.”

as well as the Methods section (p. 37-38, lines 829-848):

“We then used a machine learning classification framework to classify individuals with stroke from individuals with MCI or dementia purely based on their WMH characteristics (Supplementary Fig. 18). The input data was region-of-interest measures following three parcellations (unparcellated, periventricular/deep, custom clusters) derived as specified in section Spatial clustering, with either volumes only or volumes and microstructural variables. From UKB, we selected individuals with stroke or dementia diagnoses, excluding vascular dementia as in the previous analyses, and removed overlapping participants. From ADNI, we selected participants with a clinical diagnosis of MCI or dementia, excluding participants where the cause was determined to be due to non-AD factors as in the previous analysis. The WMH data (volumes and microstructure) were harmonized across ADNI and UKB using the ComBat method with the neuroHarmonize *python* package.⁸⁹ Specifically, controls from UKB and ADNI were used to calculate site effects with the covariates of age and sex, and corrections were applied to the diseased groups. Lasso logistic regression models were fit using the LogisticRegression function from sklearn in *python* (parameters: L1 penalty, C = 1, saga solver, balanced class weights (to address class imbalances), and max_iter = 1000). We performed 5-fold cross-validation, calculated performance metrics using all out-of-fold predictions, and repeated the process 100 times. Input data was normalized using a scaler fit on the train set and predicted on the test set at each fold. We did six runs across six sets of predictors as specified above and compared their performance using paired Wilcoxon tests using the rstatix *R* package, adjusting *p*-values with Bonferroni.”

Comment 3: If the authors—who have access to what is likely the largest multimodal dataset for analysing WMH patterns—are unable to characterise these clusters deeply, disentangle vascular from neurodegenerative contributions, or describe (not speculate) aetiologies, then it is difficult to see who would be able to do so. The literature is already full of papers offering speculative interpretations of WMH and their patterns, so producing yet another one—even if it uses a larger sample—would not be sufficient. The authors need to move beyond speculation and clearly articulate the concrete findings that their analyses have revealed. What do these massive datasets allow us to learn that has not already been demonstrated in smaller studies, many of which have reported similar patterns? In other words, what is genuinely new here—clinically, medically, or neuroscientifically?

Response: *We understand the reviewer’s request for greater clarity regarding what our analyses demonstrate, as distinct from how these findings may be interpreted biologically. First, our new analysis demonstrates that we can disentangle vascular from neurodegenerative contributions significantly better than the traditional approach of using periventricular/deep WMH volumes. Second, our work demonstrates that microstructural*

MRI metrics can be used to characterise within-lesion WMH properties across diseases. The primary advancement enabled by large multimodal datasets, therefore, is not merely replication at scale, but the ability to link spatial WMH patterns to distinct tissue-level processes and disorders. We acknowledge that, in responding to requests for nuance and caution, some of the Discussion language may have obscured these concrete advances. We have therefore revised the Discussion to more clearly separate empirically supported findings from hypothesis-generating biological interpretations. Specifically, we updated the wording in several key areas to emphasise what is directly supported by the data:

p. 23, lines 479-480: “Our data-driven approach adds important nuance to the often-used periventricular/deep parcellation (...)”

p. 23-24, lines 492-496: “Our novel approach to parsing WMH spatial heterogeneity extends this emerging data-driven literature by anchoring these spatial patterns in the underlying WMH pathophysiology.”

p. 24, lines 499-502: “While the early reduction in ICVF could point to axonal loss,^{28,32} other observations are most consistent with inflammatory swelling of cells which could increase the extracellular signal fraction and thus decrease the intracellular signal fraction, resulting in the observed effect.”

p. 24, lines 508-509: “As with all in vivo MRI studies, biological interpretations remain tentative and will need to be confirmed with gold standard ex vivo methods such as histology.”

p. 25, lines 522-523: “The pathophysiological signature suggests selective degeneration of fibers (...)”

p. 26, lines 546-548: “The question remains as to why neurodegeneration would cause hyperintense signals on FLAIR images, which are mostly sensitive to increased fluid content. Our observations allow for a more informed hypothesis.”

We added a conclusion paragraph where we summarize the main contributions in the Discussion section (p. 39, lines 621-630):

“In conclusion, this work demonstrates that moving beyond volumetric summaries of WMHs and incorporating microstructural MRI markers enables more specific separation of cerebrovascular and neurodegenerative pathologies. By modeling spatial WMH heterogeneity in a data-driven manner based on pathophysiology, we identified a posterior WMH signature in individuals on the AD spectrum, characterized by selective fiber alterations and anatomical connectivity to cortical regions vulnerable to early tau accumulation. While *in vivo* MRI cannot provide a definitive biological ground truth, the present results establish concrete, reproducible pathophysiological distinctions that extend beyond existing periventricular/deep classifications. This framework for measuring WMH

pathophysiology using clinically translatable imaging tools provides a principled foundation for future mechanistic and clinical studies.”

Comment 4: Reviewer # 3 (Comment 8): Reviewer 3 argues that the anterior-posterior split adds nuance but does not fundamentally alter the established classification; it merely refines it. The authors’ response appears to concur with this viewpoint, which, from my perspective, implies that undertaking such a large and complex study may not have been necessary if existing classifications were already adequate. This can happen in scientific research, of course, but in my view simply agreeing with this point substantially diminishes the impact of the work. A closer inspection of the parcellation shows that the anterior–posterior wording is not entirely accurate. Several juxtacortical regions appear to align with cluster 2 (posterior), and, conversely, describing cluster 3 as “deep-anterior” is not particularly precise, as most white-matter regions—aside from those in the occipital lobe—are assigned to this cluster. Even if one assumes that the main contribution of the work is to add nuance to the anterior–posterior distinction (which, in my view, is not the case), the subsequent question is whether this added nuance actually holds any benefits. To address this, the authors need to compare the performance of their parcellation with the established anterior–posterior split, as well as with the commonly used periventricular versus deep white-matter distinction, across the relevant tasks (those already conducted in their work, for instance).

Response: *First, we want to clarify that we believe our custom parcellation is very valuable and the refinements it adds to the PV/deep parcellation are crucial for distinguishing etiologies. We have modified the wording used in the Discussion section (p. 23, lines 479-481):*

“Our data-driven approach adds important nuance to the often-used periventricular/deep parcellation and revealed that deep-anterior WMHs had more severe pathophysiology than periventricular-posterior WMHs, which does not seem to be caused by spreading patterns.”

Second, qualitative descriptions of complex spatial information will always struggle to capture this complexity. We present quantitative overlaps of our parcellation with different white matter regions (Figure 2C) and show that the “deep-anterior” cluster mostly overlaps with the frontal white matter (DSC = 0.66) and not with the occipital or temporal white matter (DSC of 0.01 and 0.06, respectively), which does not support that the deep anterior cluster encompasses most white matter regions. Third, our new analysis now demonstrates that our customized parcellation outperforms the PV/deep scheme on the most crucial test (distinguishing pathologies), both when only using volumes (mean AUROC of 0.630 for PV/deep and 0.666 for custom parcellation) and when integrating microstructural information (mean AUROC of 0.697 for PV/deep and 0.706 for custom parcellation). However, we could not find an established parcellation of anterior-posterior in the literature with specific criteria (i.e., where to put the boundary between anterior and posterior). Therefore, we compared only with PV/deep, which is by far the most common parcellation scheme in the field.

Comment 5: One of the points I was not able to articulate clearly during the first round of revisions concerns the region-growing strategy used by the authors to incorporate voxels with greater sparsity (that is, voxels observed in relatively few participants). I am uncertain about the implications of this step for the clustering results. Specifically, I wonder whether the inclusion of very rare voxels may be diluting—or even obscuring—the genuine differences between clusters, thereby making the separation of clusters appear less biologically meaningful than it might otherwise be. Given this possibility, it would be valuable to assess the sensitivity of the findings to the region-growing approach. Presenting the clustering results both with and without region growth would allow readers to evaluate whether this methodological choice meaningfully alters the spatial patterns, the cluster characteristics, or the downstream associations.

Response: To clarify, all clustering analyses were performed before the region growing approach, including the described overlaps (i.e., all of Figure 2, Supplementary Figures 4 to 8). It is only when calculating ROI medians that the region-growing approach is used to include all segmented WMH voxels. In order to assess the sensitivity of our results to this approach, we replicated our central result of comparing WMH features across disorders (Figure 4) without the region-growing approach (now available in Supplementary Figure 17), and we find very few differences between the results, which do not alter any of the conclusions. This indicates that our results and conclusions are robust to our methodological choices. This is described in the Results section (p. 20, lines 412-414):

“These findings were also reproduced using a parcellation that was made without including rare WMH voxels using our search area strategy (Supplementary Fig. 17), with minimal differences that do not alter our conclusions.”

Comment 6: Remarks on code availability. I commend the authors for sharing both the patterns and the accompanying code. This is valuable, and I can imagine such an analysis being useful to replicate in the future. However, the current version of the code does not appear to be designed or documented to the extent that an external user could readily run it. For instance, the README describes the data included in the repository but does not explain how to execute the code. To make the code genuinely usable, additional documentation is needed. There is Python code in the repository, but it is unclear which environment or software versions are required to run the application.

Response: We clarified how the code should be run and the required dependencies in the README:

“To run the code, the user needs to assign the working directory to the directory of the script (e.g., analyses/1_norm_mod), load the required dependencies (see below), and then run each script in order.”

“# Dependencies

- R/4.1.2
- python/3.9.8
- PCNtoolkit/0.35 (python package; <https://github.com/amarquand/PCNtoolkit>)
- Spectrum/v1.1 (R package; <https://cran.r-project.org/package=Spectrum>)
- pySuStaIn (python package; <https://github.com/ucl-pond/pySuStaIn>)
- neuromaps (python package; <https://github.com/netneurolab/neuromaps>)”

Reviewer 3

Summary: I thank the authors for their thorough work on the manuscript and I would like to congratulate them for successfully addressing most of the comments from me and the other reviewers. I believe the manuscript has strongly improved in clarity and robustness of the methods. Overall, this is an important work with very extensive analysis. There are few points that I would still like the author to consider for their next formulation of the paper.

Comment 1: Estimates of WMH Pathophysiology: This whole paragraph has been greatly improved and it reads very clearly. One question regarding this is how the authors handled voxels that have more commonly WMH, and therefore less “points” for the normative modelling. For example, some periventricular voxels will often have lesions in individual >60 and this could bias the normative modelling estimation, and thus the WMH pathophysiology estimation. I understand that these were excluded from the clustering analysis (correct?) and then re-inserted as being part of the closest cluster. However, biased estimates would remain. The authors should address this limitation and maybe give a supplementary figure showing the number of used points (individuals with usable data) for the normative model for each voxel.

Response: *We thank the reviewer for this positive appraisal and, more generally, for their sustained and diligent attention to critiquing our work and commitment to the highest scientific standards. The reviewer is correct that voxels labelled as WMHs in individuals were excluded from the normative modelling, and voxels that were almost always labelled as WMHs (NAWM prevalence < 5000) were further excluded from the spatial clustering step. Regarding the bias, we believe that these voxels are less informative as it pertains to our goal of disentangling etiologies, as their ubiquity in healthy older adults likely indicates that they are benign. We generated a figure of NAWM and WMH prevalence and referenced it in the Results section (p. 11, lines 208-209):*

“We excluded voxels with low WMH or NAWM prevalence, which have higher noise levels (Fig. 2A; Methods; Supplementary Fig. 2 and 3) (...).”

And we discuss the potential bias in the Methods section (p. 33, lines 713-716):

“While excluding voxels with low NAWM prevalence caused by almost ubiquitous presence of WMH in older adults could induce a bias, WMHs in these areas (which are primarily in

immediately periventricular regions, as seen in Supplementary Fig. 2) are likely benign and would not be informative for our goal of parsing heterogeneity.”

Furthermore, to address reviewer 2’s comment 5, we replicated our central result of comparing WMH features across disorders (Figure 4) without re-including voxels with low NAWM of WMH into the final parcellation (now available in Supplementary Figure 17), and we find very few differences between the results, which do not alter any of the conclusions. This is described in the Results section (p. 20, lines 412-414):

“These findings were also reproduced using a parcellation that was made without including rare WMH voxels using our search area strategy (Supplementary Fig. 17), with minimal differences that do not alter our conclusions.”

Comment 2: Spatial regions of pathophysiological similar WMHs: I have three comments regarding this part. 1) I still find it reductive that the authors, after having computed all this very informative WMH pathophysiological maps, decide to go for group averages to find clusters. While I understand why and how this is done, looking for patterns at the individual level (and whether the observed clusters are actually reproducible at the subject level, or their expression changes based on some variables) would still have a very strong impact I believe. The authors should note this limitation, and I hope that they will consider this in future studies.

Response: *We agree with the reviewer that such an analysis, if designed correctly, could be very powerful, and we will attempt to find a way to do so in the future. We now discuss this limitation in the Discussion section (p. 24, lines 494-496):*

“We note that our clustering solution was based on group averages to increase robustness, but future studies should explore clustering methods taking into account subject-level variance.”

Comment 3: 2) Most importantly: The authors find 3 clusters and characterize them on several indices and spatial similarity. Have the authors considered that the observed cluster could also represent underlying fiber-types and their specific microstructural characteristics? For example, cluster 2 could represent lesions in commissural fibers, while cluster three in projections or associative fibers. This might help and change the interpretation of the observed alterations, also in relationship to the temporal sequence of events.

Response: *We have now calculated the overlaps between our clusters and commissural, association, and projection fibers. While we find some notable overlaps, these remain moderate (max DSC = 0.56), showing that our customized clusters capture unique spatial variance. Figure 2C was updated with the new results, which are discussed in the text in the Results section (p. 13, lines 245-259):*

“To quantify and contextualize the localization of each spatial cluster, we compared the overlap of our novel parcellation with other heuristic parcellations used in the field: periventricular/deep, lobar regions, cerebral artery territories as estimated from a probabilistic atlas,⁴⁰ and fiber type (Fig. 2C). We observed that the first cluster was more periventricular (Dice similarity coefficient [DSC] = 0.37), located in temporal white matter (DSC = 0.31) and in commissural fibers (DSC = 0.21), and supplied by the posterior cerebral artery (DSC = 0.27). This limited overlap between cluster 1 and the “periventricular” *a priori* region is due to the much broader coverage of the *a priori* region (<8mm from the ventricle, as is typically done),¹⁹ as compared to the data-driven cluster. The second cluster was more periventricular (DSC = 0.54), located in parietal white matter (DSC = 0.37) and commissural fibers (DSC = 0.45), and supplied by the middle cerebral artery (DSC = 0.37), although it also showed overlap with the posterior and anterior cerebral artery territories (DSC of 0.30 and 0.26, respectively). The third cluster was mostly located in deep and frontal white matter (DSC of 0.76 and 0.66, respectively) as well as association and projection fibers (DSC of 0.56 and 0.41, respectively), and largely supplied by the middle cerebral artery (DSC = 0.68).”

We clarify how the fiber atlas was derived in the Methods section (p. 33-34, lines 729-738):

“The fiber atlas was derived using the high-resolution population-averaged streamline tractography obtained from the Fibre Data Hub, comprising 34 association, 2 commissural, and 34 projection fibre bundles derived from 1,065 Human Connectome Project participants and processed using DSI Studio.^{83,84} For each fibre bundle, a voxelwise tract density map was computed by calculating the tract density index (TDI) as the total number of streamlines in the bundle passing through each voxel. Within each fibre class (association, commissural, projection), voxelwise TDIs were aggregated by taking the maximum TDI across all constituent fibre bundles, yielding one class-specific density map per class. A three-class fibre atlas was then generated using a winner-take-all voting approach.”

Comment 4: 3) It is unclear how were zero values used here, participants with very little lesions will have sparse maps, with zeros in most voxels. Was this information considered in the group averages? Was there any weighting for nr. of voxels?

Response: *Regarding the group averages (Figure 2A), only WMH values were included, thus participants without many WMHs will indeed have a lower impact on the average. To be clear, NAWM values do not have a z-score calculated and are not included in any analyses. We clarified this in the Results section (p. 9, line 179):*

“Voxels labeled as NAWM were not included in any other analyses.”

and in (p. 11, lines 206-208):

“Mean WMH pathophysiology maps were calculated by averaging all subject-level maps (excluding all voxels labelled as NAWM), resulting in one between-subject map per marker.”

NAWM voxels are not treated as 0s, but as missing values that have no impact on any result. Regarding ROI values, for individuals where <5 voxels were labelled as WMH within a cluster, we assigned the ROI value 0, which adequately represents the phenomenon of no WMH pathophysiology. We do not do any weighting for number of voxels, but we do calculate the median instead of the mean to increase robustness and limit the impact of noise. This is noted in the Methods section (p. 34, lines 755-757):

“We then calculated region-of-interest measures of WMH pathophysiology by computing the median pathophysiological value for each metric within WMHs in each spatial cluster. In cases with fewer than 5 WMH voxels within a spatial cluster, data was imputed as 0, indicating no WMH-related microstructural abnormality.”

Comment 5: Defining Temporal Pathophysiological cascades: In line with my previous comment, some participants will have all zeros in the 8 metrics for some of the clusters (or even for all of them). It is unclear how this sparsity was considered for the sustain analysis.

Response: The ROI data was not modified prior to the SuStaIn runs. Essentially, SuStaIn will treat 0s (or, more specifically, values below our first “abnormality event” of $Z > 0.5$) as evidence that there isn’t any pathology. This results in subjects having 0s on all metrics (if they didn’t have any WMH voxels in this cluster) being assigned stage and subtype 0. The variance in SuStaIn stages is thus driven by subjects with some WMHs. This is an accurate representation of reality and not a bias, since individuals with no WMH voxels in a region have logically not started the pathophysiological process of WMHs deteriorating in terms of the underlying tissue. This is now clarified in the Methods section (p. 35, lines 769-771):

“We note that subjects without WMHs in a cluster (< 5 voxels) will have all 0 values and will be assigned the stage and subtype of 0, thereby not contributing to the variance in SuStaIn stages used in subsequent analyses.”

Reviewer 4

Summary: I have carefully reviewed the authors’ rebuttal and the revised manuscript. I am fully satisfied with the responses provided to my previous comments. The authors have addressed all concerns in a thorough and convincing manner. I have no further comments or requests.

Response: We thank the reviewer for their efforts in reviewing our manuscript.